# Citizen science flow - an assessment of simple streamflow measurement methods

Jeffrey C. Davids[1,2], Martine M. Rutten[3], Anusha Pandey[4], Nischal Devkota[4], Wessel David van Oyen[3], Rajaram Prajapati[4], and Nick van de Giesen[1]

[1]Water Management, Civil Engineering and Geosciences, Delft University of Technology, Building 23, Stevinweg 1, 2628 CN, Delft, Netherlands
[2]SmartPhones4Water, 3881 Benatar Way, Suite G, Chico, California, 95928, USA
[3]Engineering and Applied Sciences, Rotterdam University, G.J. de Jonghweg 4-6, 3015 GG, Rotterdam, Netherlands
[4]SmartPhones4Water-Nepal, Damodar Marg, Thusikhel, 44600, Lalitpur, Nepal

*Correspondence to*: Jeffrey C. Davids (j.c.davids@tudelft.nl)

**Abstract.** Wise management of water resources requires data. Nevertheless, the amount of streamflow data being collected globally continues to decline. Generating hydrologic data together with citizen scientists can help fill this growing hydrological data gap. Our aim herein was to (1) perform an initial evaluation of three simple streamflow measurement methods (i.e. float, salt dilution, and Bernoulli run-up), (2) evaluate the same three methods with citizen scientists, and (3) apply the preferred method at more sites with more people. For computing errors, we used mid-section measurements from an acoustic Doppler velocimeter as reference flows. First, we (three authors) performed 20 evaluation measurements in headwater catchments of the Kathmandu Valley, Nepal. Reference flows ranged from 6.4 to 240 L s$^{-1}$. Absolute errors averaged 23, 15, and 37 % with average biases of 8, 6, and 26 % for float, salt dilution, and Bernoulli methods, respectively. Second, we evaluated the same three methods at 15 sites in two watersheds within the Kathmandu Valley with 10 groups of citizen scientists (three to four members each) and one "expert" group (three authors). At each site, each group performed three simple methods; "experts" also performed SonTek FlowTracker mid-section reference measurements (ranging from 4.2 to 896 L s$^{-1}$). For float, salt dilution, and Bernoulli methods, absolute errors averaged 41, 21, and 43 % for "experts" and 63, 28, and 131 % for citizen scientists, while biases averaged 41, 19, and 40 % for "experts" and 52, 7, and 127 % for citizen scientists, respectively. Based on these results, we selected salt dilution as the preferred method. Finally, we performed larger scale pilot testing in week-long pre- and post-monsoon Citizen Science Flow campaigns involving 25 and 37 citizen scientists, respectively. Observed flows (n = 131 pre-monsoon; n = 133 post-monsoon) were distributed among the 10 headwater catchments of the Kathmandu Valley, and ranged from 0.4 to 425 L s$^{-1}$ and 1.1 to 1804 L s$^{-1}$ in pre- and post-monsoon, respectively. Future work should further evaluate uncertainties of citizen science salt dilution measurements, the feasibility of their application to larger regions, and the information content of additional streamflow data.

# 1    Introduction

## 1.1    Background

The importance of measuring streamflow is underpinned by the reality that it is the only truly integrated representation of the entire catchment that we can plainly observe (McCulloch, 1996). Traditional streamflow measurement approaches relying on

sophisticated sensors (e.g. pressure transducers, acoustic doppler devices, etc.), site improvements (e.g. installation of weirs or stable cross-sections, etc.), and discharge measurements performed by specialists are often necessary at key observation points. However, these approaches require significant funding, equipment, and expertise, and are often difficult to maintain, and even more so to scale (Davids et al., 2017). Consequently, despite growing demand, the amount of streamflow data being collected continues to decline in several parts of the world, especially in Africa, Latin America, Asia, and even North America

(Hannah et al., 2011; Van de Giesen et al., 2014; Feki et al., 2016; Tauro et al., 2018). Specifically, there is an acute shortage of streamflow data in headwater catchments (Kirchner, 2006) and developing regions (Mulligan, 2013). This data gap is perpetuated by a lack of understanding among policy makers and citizens alike regarding the importance of streamflow data, which leads to persistent funding challenges (Kundzewicz, 1997; Pearson, 1998). This is further compounded by the reality that the hydrological sciences research community has focused much of its efforts in recent decades on advancing modeling

techniques, while innovation in methods for generating the data these models depend on has been relegated to a lower priority (Mishra and Coulibaly, 2009; Burt and McDonnell, 2015), even though these data form the foundation of hydrology (Tetzlaff et al., 2017).

Considering these challenges, alternative methods for generating streamflow and other hydrological data are being explored

(Tauro et al., 2018). For example, developments in using remote sensing to estimate streamflow are being made (Tourian et al., 2013; Durand et al., 2014), but applications in small headwater streams are expected to remain problematic (Tauro et al., 2018). Utilizing cameras for measuring streamflow is also a growing field of research (Muste et al., 2008; Le Coz et al., 2010; Dramais et al., 2011; Le Boursicaud et al., 2016), but it is doubtful that these methods will be broadly applied in headwater catchments in developing regions soon because of high costs, lacking technical capacity, and potential for vandalism. In these

cases, however, involving citizen scientists to generate hydrologic data can potentially help fill the growing global hydrological data gap (Lowry and Fienen, 2012; Buytaert et al., 2014; Sanz et al., 2014; Davids et al., 2017; van Meerveld et al., 2017; Assumpção et al., 2018).

Kruger and Shannon (2000) define citizen science as the process of involving citizens in the scientific process as researchers.

Citizen science often uses mobile technology (e.g. smartphones) to obtain georeferenced digital data at many sites, in a manner that has the potential to be easily scaled (O'Grady et al., 2016). Turner and Richter (2011) partnered with citizen scientists to map the presence or absence of flow in ephemeral streams. Fienen and Lowry (2012) showed that water level measurements from fixed staff gauges reported by passing citizens via a text message system can have acceptable errors. Mazzoleni et al.

(2017) showed that flood predictions can be improved by assimilating citizen science water level observations into hydrological models. Le Coz et al. (2016) used citizen scientist photographs to improve the understanding and modeling of flood hazards. Davids et al. (2017) showed that lower frequency observations of water level and discharge like those produced by citizen scientists can provide meaningful hydrologic information. Van Meerveld et al. (2017) showed that citizen science
observations of stream level class can be informative for deriving model-based streamflow time series of ungauged basins.

While the previously referenced studies focus mainly on involving citizen scientists for observing stream levels, we were primarily concerned with the possibility of enabling citizen scientists to take direct measurements of streamflow. Using keyword searches using combinations of "citizen science", "citizen hydrology", "community monitoring", "streamflow
monitoring", "streamflow measurements", "smartphone streamflow measurement", and "discharge measurements," we found that research on using smartphone video processing methods for streamflow measurement has been ongoing for nearly five years (Lüthi et al., 2014; Peña-Haro et al., 2018). Despite the promising nature of these technologies, we could not find any specific studies evaluating the strengths and weaknesses of citizen scientists applying these technologies directly in the field themselves.

Etter et al. (2018) evaluated the error structure of simple "stick method" streamflow estimates (similar to what we later refer to as the float method) from 136 participants from four streams in Switzerland. Participants estimated cross-sectional area with visual estimates of stream width and depth. Floating sticks were used to measure surface velocity, which was scaled by 0.8 to estimate average velocity. Besides this study, we could not find other evaluations of simple streamflow measurement
techniques that citizen scientists could possibly use. Therefore, in addition to the "stick method," we turned to the vast body of general knowledge about observing streamflow to develop a list of potential simple citizen science streamflow measurement methods to evaluate further (see Sect. 2.1 for details).

## 1.2    Research questions

Our aims in this paper were to (1) perform an initial evaluation of selected potential simple streamflow measurement methods,
(2) evaluate these potential methods with actual citizen scientists, and (3) apply the preferred method at a larger scale. Our research questions were:

- *Which simple streamflow measurement method provides the most accurate results when performed by "experts"?*
- *Which simple streamflow measurement method provides the most accurate results when performed by citizen*
*scientists?*
- *What are citizen scientists' perceptions of the required training, cost, accuracy, etc. of the evaluated simple streamflow measurement methods?*
- *Can citizen scientists apply the selected streamflow measurement method at a larger scale?*

## 1.3 Context and limitations

This research was performed in the context of a larger citizen science project called SmartPhones4Water or S4W (Davids et al., 2017; Davids et al., 2018; www.SmartPhones4Water.org). S4W leverages young researchers, citizen science, and mobile technology to improve lives by strengthening our understanding and management of water. S4W focuses on developing simple field data collection methods and low-cost sensors that young researchers and citizen scientists can use to fill data gaps in data scarce regions. Our aim is to partner with young researchers, local schools, and communities to use these openly available data to improve the quality and applicability of their water related research. S4W's first pilot project, S4W-Nepal, initially concentrated on the Kathmandu Valley, and is now expanding into other regions of the country. S4W-Nepal facilitates ongoing monitoring of precipitation, stream and groundwater levels and quality, freshwater biodiversity, and several short-term measurement campaigns focused on monsoon precipitation, land use changes, stone spout (Nepali: dhunge dhara) flow and quality, and now streamflow. One immediate application in the Kathmandu Valley is to improve estimates of water balance fluxes, including net groundwater pumping.

While identifying and refining methods for citizen scientists to measure streamflow may be an important step towards generating more streamflow data, these types of citizen science applications are not without challenges of their own. For example, citizen science often struggles with the perception (and possible reality) of poor data quality (Dickinson et al., 2010) and the intermittent nature of data collection (Lukyanenko et al., 2016). Additionally, there are other non-citizen science based streamflow measurement methods (e.g. permanently installed cameras) that may undergo rapid development and transfer of technology, and thus make a significant contribution towards closing the streamflow data gap.

Additionally, the use of "citizen scientist" herein is restricted to only student citizen scientists, which is a narrow but important subset of potential citizen scientists. Our vision was to partner with student citizen scientists first to develop and evaluate streamflow measurement methodologies. Once methodologies are refined in coordination with students, we aim to partner with community members and students in the rural hills of Nepal to improve the availability of quantitative stream and spring flow data.

## 2 Materials and methods

### 2.1 Simple streamflow measurement methods considered

Streamflow measurement techniques suggested in the United States Bureau of Reclamation Water Measurement Manual (USBR, 2001) that seemed potentially applicable for citizen scientists included: deflection velocity meters; the Manning-Strickler slope area method; and pitot tubes for measuring velocity heads. The float, current meter, and salt dilution methods described by several authors also seemed applicable (British Standards Institute, 1964; Rantz, 1982; Fleming and Henkel,

2001; Escurra, 2004; Moore, 2004a, 2004b, 2005; Herschy, 2014).  Finally, Church and Kellerhals (1970) introduced the velocity head rod, or what we later refer to as the Bernoulli run-up (or just Bernoulli) method.  Table 1 provides a summary of these eight simple measurement methods.  For the categories of (1) inapplicability in Nepal (specifically to headwater catchments), (2) cost, (3) required training, and (4) complexity of the measurement procedure, a rank of either 1, 2, or 3 was

5   given by the authors, with 1 being most favorable and 3 being least favorable.  Theses ranks were then summed, and the three methods with the lowest ranks (i.e. Bernoulli, float, and salt dilution (slug)) were selected for additional evaluation in the field.

**Table 1.  Summary of simple streamflow measurement methods considered for further evaluation.  Integer ranks of 1, 2, or 3 for inapplicability in Nepal (especially for smaller headwater catchments), cost, required training, and complexity were given to each**
10  **method, with 1 being most favorable and 3 being least favorable.  The three methods with the lowest rank were selected for further evaluation.  Smartphones are not included in equipment needs because it was assumed that citizen scientists would provide these themselves.**

| # | Method | Brief Description | Equipment Needs | Inappli-cability in Nepal | Cost | Req-uired Training | Com-plexity | Total Rank (4 to 12) | Selected for Evalu-ation (yes / no) |
|---|--------|------------------|-----------------|--------------------------|------|-------------------|-------------|---------------------|------------------------------------|
| 1 | Bernoulli | Velocity-area method.  Thin flat plate (e.g. measuring scale) used to measure velocity head.  Repeated at multiple stations. | Measuring scale | 1 | 1 | 2 | 1 | 5 | yes |
| 2 | Current Meter | Velocity-area method.  Current meter (e.g. bucket wheel, propeller, acoustic, etc.) used to measure velocity.  Repeated at multiple stations. | Current meter, measuring scale | 2 | 3 | 3 | 2 | 10 | no |
| 3 | Deflectio n Rod | Velocity-area method.  Shaped vanes projecting into the flow along with a method to measure deflection, and thereby computing velocity.  Repeated at multiple stations. | Deflection rod, measuring scale | 3 | 2 | 2 | 2 | 9 | no |
| 4 | Float | Velocity-area method.  Time for floating object to travel known distance used to determine water velocity.  Repeated at multiple stations. | Measuring scale, timer | 2 | 1 | 2 | 1 | 6 | yes |
| 5 | Manning-Strickler | Slope area method.  Slope of the water surface elevation combined with estimates of channel roughness and channel geometry to determine flow using the Manning-Strickler equation. | Auto level (or water level), measuring scale | 2 | 2 | 2 | 3 | 9 | no |

| | | | | | | | | | |
|---|---|---|---|---|---|---|---|---|---|
| 6 | Pitot Tube | Velocity-area method. Pitot tube used to measure velocity. Repeated at multiple stations. | Pitot tube, measuring scale | 2 | 2 | 2 | 2 | 8 | no |
| 7 | Salt Dilution (Constant-rate Injection) | Constant rate of known concentration of salt injected into stream. Background and steady state electrical conductivity values measured after full mixing. Flow is proportional to rate of salt injection and change in EC. | EC meter, mixing containers | 1 | 2 | 3 | 3 | 9 | no |
| 8 | Salt Dilution (Slug) | Known volume and concentration of salt injected as a single slug. EC of breakthrough curve measured. Flow is proportional to integration of breakthrough curve and volume of tracer introduced. | EC meter, mixing containers | 1 | 2 | 2 | 2 | 7 | yes |

## 2.2 Expanded description of selected simple streamflow measurement methods

### 2.2.1 Float method

The float method is based on the velocity-area principle, whereby the channel cross-section is defined by measuring depth and

width of n sub-sections, and the velocity is found by the time it takes a floating object to travel a known distance which is then corrected for friction losses. In some cases, a single float near the middle of the channel (often repeated to obtain an average value) is used to determine surface velocity (Harrelson et al., 1994). In this study, surface velocity was measured at each of the n sub-sections. Total streamflow (Q) in liters per second (L s$^{-1}$) is calculated with Eq. (1):

$$Q = 1000 * \sum_{i=1}^{n} C * V_{F_i} * d_i * w_i \qquad (1)$$

where 1000 is a conversion factor from m$^3$ s$^{-1}$ to L s$^{-1}$, C is a unitless coefficient to account for the fact that surface velocity is typically higher than average velocity (typically in the range of 0.66 to 0.80 depending on depth; USBR, 2001) due to friction from the channel bed and banks, $V_{F_i}$ is surface velocity from float in meters per second (m s$^{-1}$), $d_i$ is depth (m), and $w_i$ is width

(m) of each sub-section (i = 1 to n, where n is the number of stations). A coefficient of 0.8 was used for all float method measurements in this study. Surface velocity for each sub-section was determined by measuring the amount of time it takes for a floating object to move a certain distance. For floats we used sticks found on site. Sticks are widely available (i.e. easiest for citizen scientists), generally float (except for the densest varieties of wood), and depending on their density, are between 40 and 80% submerged, which minimizes wind effects. An additional challenge with floats is that they can get stuck in eddies,

pools, or overhanging vegetation.

Float method streamflow measurements involve the following steps:

1. Select stream reach with straight and uniform flow
2. Divide cross-section into several sub-sections (n, typically between 5 and 20)
3. For each sub-section, measure and record
   a. The depth in the middle of the sub-section
   b. The width of the sub-section
   c. The time it takes a floating object to move a known distance downstream (typically 1 or 2 m) in the middle of the sub-section
4. Solve for streamflow (Q) with Eq. (1)

Distances of 1 or 2 m were necessary to measure surface velocity for each sub-section since it was unlikely that a float would stay in a single sub-section for 10 or 20 m. These shorter distances ensured that surface velocity measurements were representative of their respective sub-sections and associated areas. One benefit of this approach was that the measured surface velocities were cross-sectional area weighted. This area weighting was more important as surface velocity differences between the center and the sides of the channel increased. Since these velocity differences vary from site to site, using a single float with a single coefficient (e.g. 0.8) would have ignored these differences among sites.

### 2.2.2    Salt dilution method

There are two basic types of salt dilution flow measurements: slug (previously known as instantaneous) and continuous rate (Moore, 2004a). Salt dilution measurements are based on the principle of the conservation of mass. In the case of the slug method, a single known volume of high concentration salt solution is introduced to a stream and the electrical conductivity (EC) is measured over time at a location sufficiently downstream to allow good mixing (Moore, 2005). An approximation of the integral of EC as a function of time is combined with the volume of tracer and a calibration constant (Eq. 2) to determine discharge. In contrast, continuous rate salt dilution method involves introducing a known flow rate of salt solution into a stream (Moore, 2004b). Slug method salt dilution measurements are broadly applicable in streams with flows up to 10 $m^3$ $s^{-1}$ with steep gradients and low background EC levels (Moore, 2005). For the sake of citizen scientist repeatability, we chose to only investigate the slug method, because of the added complexity of measuring the flow rate of the salt solution for the continuous rate method. Some limitations of the salt dilution method include: (1) inadequate vertical and horizontal mixing of the tracer in the stream, (2) trapping of the tracer in slow moving pools of the stream, and (3) incomplete dilution of salt within the stream water prior to injection. The first two limitations can be addressed with proper site selection (i.e. well mixed reach with little slow-moving bank storage), while incomplete dilution can be avoided by proper training of the personnel performing the measurement.

Streamflow (Q; L s$^{-1}$) is solved for using Eq. (2) (Rantz, 1982; Moore, 2005):

$$Q = \frac{V}{k \sum_{i=1}^{n}(\sigma(t) - \sigma_{BG})\Delta t} \qquad (2)$$

where V is the total volume of tracer introduced into the stream (L), k is the calibration constant in centimeters per microsiemens (cm µS$^{-1}$), n is the number of measurements taken during the breakthrough curve (unitless), $\sigma(t)$ is the EC at time t (µS cm$^{-1}$), $\sigma_{BG}$ is the background EC (µS cm$^{-1}$), and $\Delta t$ is the change in time between EC measurements (s).

Salt dilution method streamflow measurements involve the following steps:

1. Select stream reach with turbulence to facilitate vertical and horizontal mixing
2. Determine upstream point for introducing the salt solution and a downstream point for measuring EC
   a. A rule of thumb in the literature is to separate these locations roughly 25 stream widths apart (Day, 1977; Butterworth et al., 2000; Moore, 2005)
3. Estimate flow either performing a "simplified float measurement (i.e. only a few sub-sections)" or by visually estimating width, average depth, and average velocity
4. Prepare salt solution based on the following guidelines (approximate average of dosage recommendations from previous studies cited by Moore (2005))
   a. 10000 ml of stream water for every 1 m$^3$ s$^{-1}$ of estimated streamflow
   b. 1667 g of salt for every 1 m$^3$ s$^{-1}$ of estimated streamflow
   c. Thoroughly mix salt and water until all salt is dissolved
   d. Following these guidelines ensure a homogenous salt solution with 1 to 6 salt to water ratio by mass
5. Establish the calibration curve relating EC values to actual salt concentrations (Moore, 2004b) to determine calibration constant (k) relating changes in EC values in micro Siemens per centimeter (µS cm$^{-1}$) in the stream to relative concentration of introduced salt solution (RC; see Sect. 2.3.3 for details)
6. Dump salt solution at upstream location
7. Measure EC at downstream location during salinity breakthrough until values return to background EC
   a. Record a video of the EC meter screen at the downstream location and later digitize the values using the time from the video and the EC values from the meter
8. Solve for streamflow (Q) with Eq. (2)

### 2.2.3    Bernoulli run-up method

Like the float method, Bernoulli run-up (or Bernoulli) is based on the velocity-area principle.  The basic principle is that "run-up" on a flat plate inserted perpendicular to flow is proportional to velocity based on the solution to Bernoulli's equation.  Bernoulli run-up is also referred to as the "velocity head rod" by Church and Kellerhals (1970), Carufel (1980), and Fonstad et al. (2005), and is similar to the "weir stick" discussed by USBR (2001).  The velocity measurement theory of Bernoulli is similar to using a pitot tube (Almeida and Souza, 2017), without the associated challenges of (1) using and transporting potentially bulky and fragile equipment and (2) clogging from sediment or trash (WMO, 2010).  However, the accuracy and precision of Bernoulli method velocity head measurements are likely lower than pitot measurements.  Total streamflow (Q; L s$^{-1}$) is calculated with Eq. (3):

$$Q = 1000 * \sum_{i=1}^{n} V_{B_i} * d_{1_i} * w_i \qquad (3)$$

where 1000 is a conversion factor from m$^3$ s$^{-1}$ to L s$^{-1}$, $V_{B_i}$ is velocity from Bernoulli run-up (m s$^{-1}$), $d_{1_i}$ is depth (m), and w$_i$ is width (m) of each sub-section (i = 1 to n).  Area for each sub-section is the product of the width and the depth in the middle of each sub-section.  Velocity for each sub-section ($V_{B_i}$) was determined by measuring the "run-up" or change in water level on a thin meter stick (or "flat plate;" dimensions: 1 meter long, by 34 mm wide, by 1.5 mm thick used in this study) from when the flat plate was inserted parallel and then perpendicular to the direction of flow.  The parallel depth measurement represents static head, while the perpendicular represents total head.  Velocity ($V_{B_i}$; m s$^{-1}$) is calculated from Bernoulli's principle with Eq. (4):

$$V_{B_i} = \sqrt{2g * (d_{2_i} - d_{1_i})} \qquad (4)$$

where g is the gravitational constant (m s$^{-2}$) and $d_{2_i}$ and $d_{1_i}$ are the water depths (m) when the flat plate was perpendicular and parallel to the direction of flow, respectively.

Bernoulli method streamflow measurements involve the following steps:

1. Select constricted stream section with elevated velocity to increase the difference between $d_{1_i}$ and $d_{2_i}$
2. Divide cross-section into several sub-sections (n, typically between 5 and 20)
3. For each sub-section, measure and record
   a. The depth with a flat plate held perpendicular to flow ($d_{2_i}$ or the "Run-up" depth)
   b. The depth with a flat plate held parallel to flow ($d_{1_i}$ or the actual water depth)

          c.   The width of the sub-section

   4.   Solve for streamflow (Q) with Eq. (3) and Eq. (4)

## 2.3     General items

### 2.3.1    Types of streams evaluated

Streams evaluated during this investigation (Phases 1, 2, and 3) were a mixture of pool and riffle, pool and drop, and run stream types. Streamflows ranged from 0.4 to 1804 L s$^{-1}$. Stream widths and average depths ranged from 0.1 to 6.0 m and 0.0040 and 0.97 m, respectively. Streambed materials ranged from cobles, gravels, and sands in the upper portions of watershed to sands, silts, and sometimes man-made concrete streambeds and side retaining walls in the lower portions. During pre-monsoon, sediment loads were generally low, while during post-monsoon, increased water velocities led to increased sediment

loads (both suspended and bed). Slopes (based on Phase 2 data) ranged from 0.020 to 0.148 m m$^{-1}$. Additional details about the measurement sites are provided in Tables 4 and 5. Since roughly 80 % of Nepal's precipitation occurs during the summer monsoon (Nayava, 1974), pre- and post-monsoon represent periods of relatively low and high streamflows, respectively. Therefore, we consistently use pre-monsoon and post-monsoon to refer to the general seasons that Phase 1, 2, and 3 activities were performed in.

### 2.3.2    Reference flows

To evaluate different simple citizen science flow measurement methods, reference (or actual) flows for each site were needed. We used a SonTek FlowTracker acoustic Doppler velocimeter (ADV) to determine reference flows. The United States Geological Survey (USGS) mid-section method was used, following guidelines from USGS Water Supply Paper 2175 (Rantz, 1982), along with instrument specific recommendations from SonTek's FlowTracker manual (SonTek, 2009). Stream depths

were shallow enough that a single vertical 0.6 depth velocity measurement (i.e. 40% up from the channel bottom) was used to measure average velocity for each sub-section (Rantz, 1982). While there is uncertainty in using the 0.6 depth as representative of average velocity, Rantz (1982) states that "actual observation and mathematical theory have shown that the 0.6 depth method gives reliable results" for depths less than 0.76 m; multipoint methods are not recommended for depths less than 0.76 m, so this is the recommended USGS approach. Depending on the total width of the channel, the number of sub-sections ranged

from 8 to 30. The FlowTracker ADV has a stated velocity measurement accuracy of within one percent (SonTek, 2009). Based on an ISO discharge uncertainty calculation within the SonTek FlowTracker software, the uncertainties in reference flows for Phase 1 and 2 ranged from 2.5 to 8.2 %, with a mean of 4.2 %. Based on the literature (Rantz, 1982; Harmel, 2006; Herschy, 2014), these uncertainties in reference flows are towards the lower end of the expected range for field measurements of streamflow. Therefore, we do not think that any systematic biases or uncertainties in our data change the results of this

paper. A compilation of the measurement reports generated by the FlowTracker ADV, including summaries of measurement uncertainty, are included as supplementary material.

### 2.3.3    Salt dilution calibration coefficient (k)

Our experience was that the most complicated portion of a salt dilution measurement was performing the dilution test to determine the calibration coefficient k.  The calibration coefficient k relates changes in EC values in micro Siemens per centimeter ($\mu$S cm$^{-1}$) in the stream to relative concentrations of introduced salt solution (RC).  During Phases 1 and 2, we determined k using a calibrated GHM 3431 [GHM-Greisinger] EC meter with the procedure recommended by Moore (2004b; additional details included as supplementary materials).

Due to the challenges of measuring k in the field, especially for citizen scientists who are the ultimate target for performing these streamflow measurements, average k values were used to determine salt dilution streamflows.  For Phase 1, an average k of 2.79E-06 (n = 10) was used for all 20 measurement sites (Table 4).  For Phase 2, an average k of 2.95E-06 (n = 15) was used for all 15 sites (Table 5).  For Phase 3, the Phase 2 average k of 2.95E-06 was used to calculate streamflows for all salt dilution measurements.  The impact of using average k values on salt dilution measurements is discussed in Sect. 4.1.  Moore (2005) suggests that k is a function of (1) the ratio of salt and water in the tracer solution and (2) the chemical composition of the stream water.  To minimize variability in k due to changes in salt concentration, a fixed ratio of salt to water (i.e. 1 to 6 by mass) was used to prepare tracer solutions for all phases of this investigation.

### 2.3.4    Inexpensive EC meters

For Phases 2 and 3, ten inexpensive (i.e. $15) Water Quality Testers [HoneForest] were used to measure EC for salt dilution measurements.  To evaluate the accuracy of these meters, we performed a six-point comparison test with reference EC values of 20, 107, 224, 542, 1003, and 1517 $\mu$S cm$^{-1}$, as determined by a calibrated GHM 3431 [GHM-Greisinger] EC meter.  EC measurements were performed from low EC to high EC (for all six points) and were repeated three times for each meter.  Because EC is used to compute the integral of the breakthrough curve (Eq. 2), the percent difference (i.e. error) in EC changes between the six points (i.e. five intervals) from the inexpensive meters were compared to reference EC intervals (Fig. 1).  Based on this analysis, the inexpensive meters had a positive median bias of roughly 5 % (ranging from -14 to 21 %) for EC value changes between 20 and 542 $\mu$S cm$^{-1}$ (i.e. D1, D2, and D3).  A nearly zero median bias (ranging from -5 to 5 %) for EC value changes between 542 and 1003 $\mu$S cm$^{-1}$ (i.e. D4) was present.  Finally, there was a negative median bias of roughly -9 % (ranging from -18 to 6 %) for EC value changes between 1003 and 1517 $\mu$S cm$^{-1}$ (i.e. D5).  No corrections were made to EC measurements collected with inexpensive [HoneForest] EC meters.

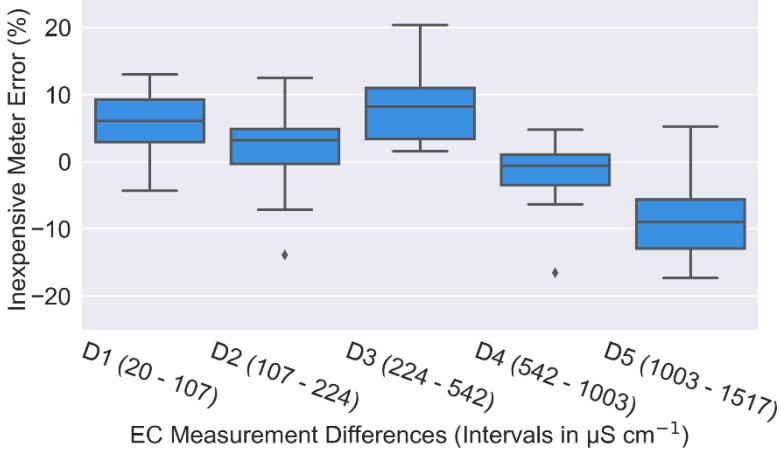

**Figure 1: Box plots of inexpensive Water Quality Testers [HoneForest] errors for five different intervals (i.e. D1 to D5). The range of EC values from reference EC measurements (determined by a calibrated GHM 3431 [GHM-Greisinger] EC meter) are shown in parentheses in µS cm⁻¹. Boxes show the interquartile range between the first and third quartiles of the dataset, while whiskers extend to show minimum and maximum values of the distribution, except for points that are determined to be "outliers" (shown as diamonds), which are more than 1.5 times the interquartile range away from the first or third quartiles.**

## 2.4    Phases of the investigation

This investigation was carried out in three distinct phases including: Phase 1 - initial evaluation; Phase 2 - citizen scientist evaluation; and Phase 3 - citizen scientist application (Table 2).

**Table 2.  Brief descriptions of three data collection phases including who performed the field data collection, and what period and season the data were collected in.**

| # | Phase | Description | Performed by | Period | Season |
|---|-------|-------------|--------------|--------|--------|
| 1 | Initial Evaluation | Initial evaluation of three simple flow measurement methods (i.e. float, salt dilution, and Bernoulli) along with FlowTracker ADV reference flow measurements at 20 sites within the Kathmandu Valley. Reference flows ranged from 6.4 to 240 L s⁻¹. | Authors | March/ April 2017 | Pre-monsoon |
| 2 | Citizen Scientist Evaluation | Citizen Scientist evaluation of three simple flow measurement methods (i.e. float, salt dilution, and Bernoulli) along "expert" and FlowTracker ADV reference flow measurements at 15 sites within the Kathmandu Valley. Reference flows ranged from 4.2 to 896 L s⁻¹. | Authors for "expert" and reference flows plus 10 Citizen Science Flow groups for simple methods | September 2018 | Post-monsoon |

| | | Salt dilution measurements at roughly 130 sites in the 10 perennial watersheds of the Kathmandu Valley. Float measurements with a small number of sub-sections (e.g. 3 to 5) performed at each site to determine salt dosage. Observed flows ranged between 0.4 to 425 and 1.1 to 1804 L s$^{-1}$ in pre and post-monsoon, respectively. | 18 Citizen Science Flow groups (8 from April and 10 from September) | April and September 2018 | Pre- and Post-Monsoon |
|---|---|---|---|---|---|
| 3 | Citizen Scientist Application | | | | |

### 2.4.1     Initial evaluation (Phase 1)

For Phase 1 evaluation of the three simple streamflow measurement methods, we (three authors) performed sets of measurements at 20 sites within the Kathmandu Valley, Nepal (Fig. 2.a and 2.b). The Kathmandu Valley is a small
5    intermontane basin roughly 25 km in diameter with a total area of 587 km$^2$ in the Central Region of Nepal, and encompasses most of Kathmandu, Bhaktapur, and Lalitpur districts. Figure 2.c is a photograph of the typical types of relatively steep pool and drop stream systems included in Phase 1. Sites were chosen to represent a typical range of stream types, slopes, and flow rates. At each site, we performed float, salt dilution, and Bernoulli measurements, in addition to reference flow measurements with the FlowTracker ADV per the descriptions in Sect. 2.2 and 2.3.2, respectively. All Phase 1 salt dilution EC measurements
10    were taken with a calibrated GHM 3431 [GHM-Greisinger] EC meter.

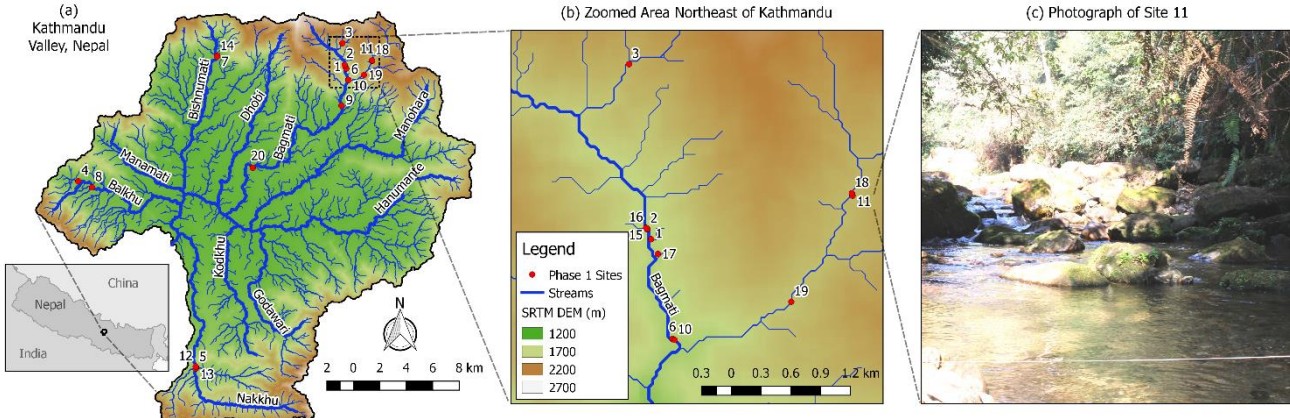

**Figure 2: Map showing topography of the Kathmandu Valley from a Shuttle Research Telemetry Mission (SRTM, 2000) Digital Elevation Model (DEM), resulting stream network (Davids et al., 2018), and locations of phase 1 measurement sites (a). Names of**
15    **the ten historically perennial tributaries are shown. Panel (b) shows an enlarged view of the area where 11 of the 20 measurements were taken. Panel (c) is a photograph of site 11, a pool and riffle sequence flowing at roughly 100 L s$^{-1}$. Measurement sites are labelled with Phase 1 Site IDs.**

At each site, measurements were performed consecutively, and took roughly one to two hours to perform, depending on the size of the stream and the resulting number of sub-sections for float, Bernoulli, and reference flow measurements. Measurements were performed during steady state conditions in the stream; if runoff generating precipitation occurred during measurements at a site, the measurements were stopped, and repeated after streamflows stabilized at pre-event levels. As

5    previously described, salt dilution calibration coefficient k was determined at 10 of the 20 sites. Field notes for float, salt dilution, and Bernoulli were taken manually and later digitized into a spreadsheet (included as supplementary material). Results from Phase 1 are summarized in tabular form (Table 4). To understand relative (normalized) errors, we calculated percent differences in relation to reference flow for each method. Averages of absolute value percent differences (absolute errors), average errors (bias), and standard deviations of errors were used as metrics to compare results among methods and

10    between Phase 1 and 2.

### 2.4.2    Citizen scientist evaluation (Phase 2)

To evaluate the same three streamflow measurement methods with actual citizen scientists, we recruited 37 student volunteers from Khwopa College of Engineering in Bhaktapur, Nepal for our Citizen Science Flow (CS Flow) evaluation. 10 CS Flow evaluation groups of either three or four members were formed. Citizen scientists were second and third-year civil engineering

15    Bachelors' students ranging in age from 21 to 25; 12 were female and 25 were male. Phase 2 citizen scientist evaluations (Fig. 3) were performed at seven sites in the Dhobi watershed in the north (Fig. 3.b; D1 to D7) and eight sites in the Nakkhu watershed in the south (Fig. 3.c; N1 to N8). Sites were chosen to represent a typical range of stream types, slopes, and flow rates found within the headwater catchments of the Kathmandu Valley, and to minimize travel time between locations.

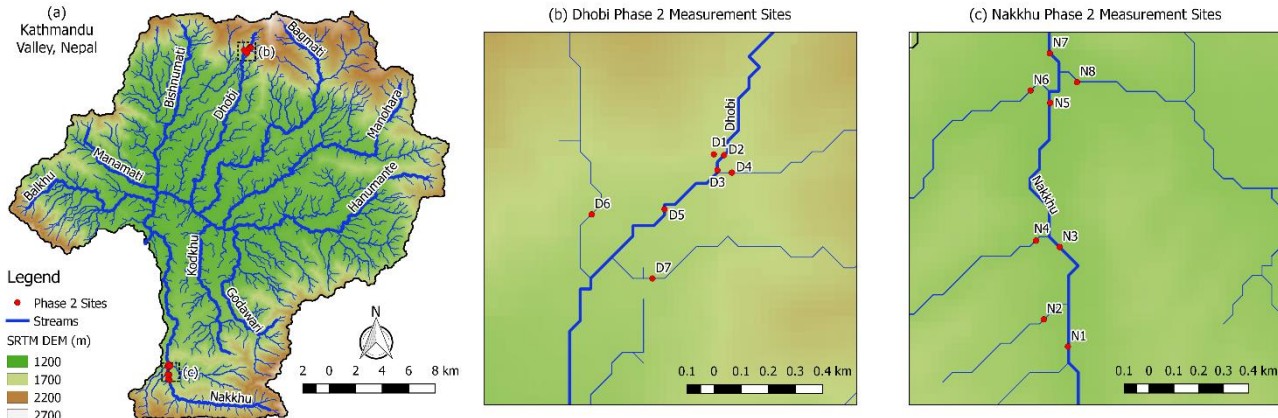

**Figure 3: Map showing topography of the Kathmandu Valley, stream network, and locations of phase 2 measurement sites (a). Names of the ten historically perennial tributaries are shown. Panel (b) shows an enlarged view of the upper Dhobi watershed where Phase 2 measurements D1 through D7 were performed. Panel (c) shows an enlarged view of the middle Nakkhu watershed where Phase 2 measurements N1 through N8 were performed. Measurement sites are labelled with Phase 2 Site IDs.**

Phase 2 started on 17 September (2018) with a four-hour theoretical training on the float, salt dilution, and Bernoulli streamflow measurement methods per Sect. 2.2. The theoretical training also introduced citizen scientists to Open Data Kit (ODK; Anokwa et al., 2009), a freely available open-source software for collecting and managing data in low-resource settings. ODK was used with the specific streamflow measurement workflow described below.

Based on our initial experiences and results from Phase 1, we developed an ODK form to facilitate the collection of float, salt dilution, Bernoulli, and reference streamflow measurement data. After installing ODK on an Android smartphone, and downloading the necessary form from S4W-Nepal's ODK Aggregate server on the Google Cloud App Engine, the general workflow is included as supplementary material.

Training was continued on 18 September with a two-hour field demonstration session in the Dhobi watershed located in the north of the Kathmandu Valley. During this field training, we worked with three to four groups at a time, and together performed float, salt dilution, and Bernoulli measurements at site D3.

15 Following the field training, a Google My Map with the 15 sites was provided to the citizen scientists. Groups were strictly instructed to not discuss details regarding the selection of measurement reaches or the results of the streamflow measurements with other groups. For the remainder of 18 September and all of 19 September, the 10 CS Flow groups rotated between the seven sites in the Dhobi watershed. To ensure that measurements could be compared with each other, four S4W-Nepal interns travelled between sites to verify that CS Flow groups performed measurements on the same streams in the same general
20 locations. All eight measurements on the Nakkhu watershed were performed in similar fashion on 20 September.

Using the same schedule of the CS Flow groups, the "expert" group (three authors) visited the same 15 sites. At each site, in addition to performing float, salt dilution, and Bernoulli measurements, the "expert" group performed (1) reference flow measurements per Sect. 2.3.2, (2) salt dilution calibration coefficient k dilution measurements per Sect. 2.3.3, and (3) an auto-
25 level survey to determine average stream slope. At each site, auto-level surveys included topographical surveys of stream water surface elevations with a AT-B4 24X Auto-Level [Topcon] at five locations including: 10 times and 5 times the stream width upstream of the reference flow measurement site (reference site), at the reference site, and 5 and 10 times the stream width downstream of the reference site. For each site, stream slope was taken as the average of the four slopes computed from the five water surface elevations measured.

All CS Flow and "expert" measurements were conducted under steady state conditions. Based on two S4W-Nepal citizen scientists' precipitation measurements (official government records aren't available until the subsequent year) nearby the Dhobi sites (i.e. roughly 3 km to the west and east), no measurable precipitation occurred during 18 and 19 September. Water level measurements from a staff gauge installed at site D3 taken at the beginning and end of 18 and 19 September confirmed

that water levels (and therefore flows) remained steady. On 20 September, 7 mm of precipitation was recorded by a S4W-Nepal citizen scientist in Tikabhairab which is roughly 1 km north of the eight measurement sites in the Nakkhu watershed. Based on field observations of the "expert" group, rain didn't start until 15:30 LT, and all CS Flow group measurements were completed before 15:30 LT. Three "expert" measurement sites were completed after 15:30 LT, but most rain was concentrated downstream (to the north) of these sites (i.e. N1, N2, and N3). Based on water level measurements performed at the beginning, middle, and end of measurements at these sites, no changes in water levels (and therefore flows) were observed. We also don't see any systematic impacts to the resulting comparison data for these sites (Table 5 and Fig. 4).

Once ODK forms from all 15 sites were finalized and submitted to the ODK Aggregate server, CS Flow and "expert" groups digitized breakthrough curves (i.e. time and EC) from EC videos in shared Google Sheet salt dilution flow calculators. Digitizations for all measurements were then reviewed for accuracy and completeness by the authors.

After the completion of Phase 2 field work, a Google Form survey was completed by 33 of the Phase 2 citizen scientists (Table 3). The purpose of the survey was to evaluate citizen scientists' perceptions of the three simple streamflow measurement methods. The survey questions forced participants to rank each method from 1 to 3. Questions were worded so that in all cases, a rank of 1 was most favourable and 3 was least favourable.

**Table 3. Summary of Phase 2 survey questions and the meanings of ranks, respectively.**

| # | Question | Rank 1 Meaning | Rank 3 Meaning |
|---|---|---|---|
| Q1 | Required training for each method | Least | Most |
| Q2 | Cost of equipment for each method | Least | Most |
| Q3 | Number of citizen scientists required for each method | Least | Most |
| Q4 | Data recording requirements for each method | Least | Most |
| Q5 | Complexity of procedure for each method | Least | Most |
| Q6 | Enjoyability of measurement method | Most | Least |
| Q7 | Safety of each method | Most | Least |
| Q8 | Accuracy of each method | Most | Least |

A tabular summary of the 15 Phase 2 measurement locations was developed (Table 5). To understand relative (normalized) errors, we calculated percent differences in relation to reference flow for each method. Averages of absolute value percent differences (absolute errors), average errors (bias), and standard deviations of errors were used as metrics to compare results among methods and between Phase 1 and 2. Box plots showing the distribution of CS Flow group measurement errors along with "expert" measurement errors for each method were developed (Fig. 4). To visualize the results of the citizen scientists' perception survey, a stacked horizontal bar plot grouped by streamflow measurement methods was developed (Fig. 5).

### 2.4.3    Citizen scientist application (Phase 3)

From 15 to 21 April (2018; pre-monsoon) and 21 to 25 September (2018; post-monsoon), 25 and 37 second and third-year engineering Bachelors' student citizen scientists, respectively, from Khwopa College of Engineering in Bhaktapur, Nepal joined S4W-Nepal's Citizen Science Flow (CS Flow) campaign. Citizen scientists formed 8 pre-monsoon and 10 post-

monsoon CS Flow groups of three or four people each, respectively. Ages of pre-monsoon citizen scientists ranged from 21 to 25; 7 were female and 18 were male (post-monsoon group composition is described in Sect. 2.4.2).

Post-monsoon Phase 3 measurements were performed by the same 10 CS Flow groups that performed Phase 2 citizen scientist evaluations. Therefore, additional training for these groups was not necessary. Training for pre-monsoon CS Flow groups

included a four-hour theoretical training on 15 April about the float and salt dilution streamflow measurement methods per Sect. 2.2. The theoretical training also introduced citizen scientists to ODK Android data collection application. For both pre- and post-monsoon Phase 3 measurements, the workflow was similar to that described in Sect. 2.4.2 (see supplementary material for details), with the exceptions of (1) skipping collection of Bernoulli data, and (2) only performing a "simplified" float measurement involving only two or three sub-sections in order to have a flow estimate for calculating the recommended

salt dose. Training was continued on the afternoon of 15 April with a two-hour field demonstration session in the Hanumante watershed located in the southwestern portion of the Kathmandu Valley (Fig. 6). During this field training, we worked with four groups at a time, and together performed "simplified" float and Bernoulli measurements at two sites.

After training was completed, citizen scientists were sent to the field to perform streamflow measurements as described above

in all 10 headwater catchments of the Kathmandu Valley (Fig. 6). All Phase 3 salt dilution EC breakthrough curve measurements were performed with inexpensive [HoneForest] meters. Once ODK forms from all Phase 3 measurements were finalized and submitted to the ODK Aggregate server, CS Flow groups digitized breakthrough curves (i.e. time and EC) from EC videos in shared Google Sheet salt dilution flow calculators. Digitizations for all measurements were then reviewed for accuracy and completeness by the authors. While not included in this paper, it is important to note that students analyzed the

collected flow data and finally presented oral and written summaries of their quality-controlled results to their faculty and peers at Khwopa College of Engineering.

While subsequent work will highlight the knowledge about spring and streamflows gained from these data, the purpose herein is more a proof of concept showing that the salt dilution method can be successfully applied at more sites with more people.

As such, a simple map figure is used to show the spatial distribution of measurements. The three streamflow gauging stations within the Kathmandu Valley (only one in a headwater catchment) operated by the official government agency responsible for streamflow measurements (i.e. the Department of Hydrology and Meteorology or DHM) are also included. Additionally,

histograms of flow and EC for pre- and post-monsoon are also shown. While measurements in pre- and post-monsoon were not all taken in the same locations, histograms can still be used to see seasonal changes in distributions.

## 3    Results

The following results section is organized into the same three phases included in the methodology (Sect. 2.4): initial evaluation
(Phase 1), citizen scientist evaluation (Phase 2), and citizen scientist flow application (Phase 3).

### 3.1    Initial evaluation results (Phase 1)

Reference flows evaluated in Phase 1 ranged from 6.4 to 240 L s$^{-1}$ (Table 4; sorted in ascending order by reference flow). Elevations of measurements ranged from 1313 to 1905 meters above mean sea level. Salt dilution calibration coefficients (k) averaged 2.79E-06 and ranged from 2.57E-06 to 3.02E-06. Absolute errors with respect to reference flows averaged 23, 15,
and 37 %, while biases for all methods were positive, averaging 8, 6, and 26 % for float, salt dilution, and Bernoulli methods, respectively. Standard deviations of errors were 29, 19, and 62 % for float, salt dilution, and Bernoulli methods, respectively. The largest salt dilution errors occurred for reference flows of 21 L s$^{-1}$ or less (i.e. sites 1 through 7), while float and Bernoulli errors were more evenly distributed throughout the range of observed flows. Field notes from Bernoulli flow measurements for two measurements (Site IDs 9 and 19) were destroyed by water damage, so Bernoulli flow and percent difference data
were not available for these sites. Detailed reports for reference flow measurements along with calculations for each simplified streamflow measurement method are included as supplementary material.

**Table 4: Summary of initial evaluation (Phase 1) measurement comparison data. Records sorted in ascending order by reference flow (Q Reference). Latitude and longitude in reference to the WGS84 datum. All flow values shown are shown in L s$^{-1}$ rounded to**
**the nearest integer for values greater than or equal to 10 and to the nearest tenth place for values less than 10. Percent differences (errors) calculated using Q Reference (FlowTracker) as the actual flow. Data summarized at the bottom with average, minimum (min), maximum (max), and standard deviation (std dev). Note that averages (avg *) shown in the summary area near the bottom for the last three columns (i.e. percent errors) include averages of absolute values of percent errors (i.e. absolute errors) shown underlined in parentheses. Null (empty) cells indicate that data for that site and parameter were either damaged (i.e. Q Bernoulli**
**for SiteIDs 9 and 19) or not collected in the field (i.e. missing k values). Average k (2.79E-06) was used to compute Q Salt for all Phase 1 sites.**

| Site ID | Date | Latitude | Longitude | Elev-ation (m) | k (cm μS$^{-1}$) | Q Ref-erence (L s$^{-1}$) | Q Float (L s$^{-1}$) | Q Salt (L s$^{-1}$) | Q Ber-noulli (L s$^{-1}$) | % Error Float | % Error Salt | % Error Ber-noulli |
|---|---|---|---|---|---|---|---|---|---|---|---|---|
| 1 | 02/03/17 | 27.78065 | 85.42426 | 1649 | | 6.4 | 7.4 | 4.3 | 8.8 | 16 | -34 | 37 |
| 2 | 18/04/17 | 27.78158 | 85.42385 | 1659 | | 6.9 | 8.0 | 7.5 | 10 | 15 | 9 | 45 |
| 3 | 10/03/17 | 27.79649 | 85.42177 | 1905 | 2.76E-06 | 11 | 7.8 | 12 | 8.8 | -28 | 10 | -19 |
| 4 | 24/04/17 | 27.70026 | 85.22077 | 1406 | | 17 | 19 | 19 | 18 | 11 | 13 | 5 |
| 5 | 22/03/17 | 27.57487 | 85.31314 | 1482 | 2.80E-06 | 18 | 20 | 24 | 19 | 12 | 38 | 5 |
| 6 | 19/04/17 | 27.77164 | 85.42657 | 1609 | | 19 | 28 | 28 | 22 | 48 | 49 | 16 |

| No | Date | Latitude | Longitude | Q Reference | k | Q Float | Q Salt | Q Bernoulli | | Float err | Salt err | Bernoulli err |
|---|---|---|---|---|---|---|---|---|---|---|---|---|
| 7 | 30/03/17 | 27.78691 | 85.32589 | 1364 | 2.57E-06 | 21 | 26 | 27 | 48 | 27 | 32 | 132 |
| 8 | 24/04/17 | 27.69620 | 85.23142 | 1382 | | 23 | 9.5 | 25 | 6.3 | -59 | 7 | -73 |
| 9 | 19/04/17 | 27.75406 | 85.42170 | 1355 | | 34 | 51 | 34 | | 52 | 0 | |
| 10 | 19/04/17 | 27.77154 | 85.42680 | 1609 | | 41 | 41 | 48 | 63 | 0 | 16 | 53 |
| 11 | 01/03/17 | 27.78483 | 85.44480 | 1877 | | 104 | 111 | 85 | 101 | 7 | -18 | -3 |
| 12 | 22/03/17 | 27.57542 | 85.31268 | 1477 | 2.67E-06 | 111 | 106 | 115 | 116 | -4 | 4 | 5 |
| 13 | 22/03/17 | 27.57410 | 85.31277 | 1481 | 2.83E-06 | 117 | 81 | 128 | 102 | -31 | 10 | -13 |
| 14 | 30/03/17 | 27.78627 | 85.32583 | 1356 | 2.74E-06 | 153 | 208 | 141 | 470 | 37 | -7 | 208 |
| 15 | 02/03/17 | 27.78156 | 85.42383 | 1659 | | 155 | 248 | 130 | 161 | 59 | -16 | 4 |
| 16 | 18/04/17 | 27.78168 | 85.42373 | 1663 | | 156 | 140 | 144 | 210 | -10 | -8 | 34 |
| 17 | 10/03/17 | 27.77932 | 85.42496 | 1653 | 2.80E-06 | 159 | 183 | 155 | 228 | 15 | -2 | 43 |
| 18 | 11/03/17 | 27.78505 | 85.44473 | 1877 | 2.91E-06 | 208 | 221 | 216 | 150 | 7 | 4 | -28 |
| 19 | 11/03/17 | 27.77514 | 85.43867 | 1806 | 3.02E-06 | 230 | 188 | 237 | | -18 | 3 | |
| 20 | 20/04/17 | 27.71106 | 85.35432 | 1313 | 2.78E-06 | 240 | 246 | 267 | 264 | 3 | 12 | 10 |
| | | | avg * -> | 1579 | 2.79E-06 | 92 | 97 | 92 | 111 | 8 (23) | 6 (15) | 26 (37) |
| | | | min -> | 1313 | 2.57E-06 | 6.4 | 7.4 | 4.3 | 6.3 | -59 | -34 | -73 |
| | | | max -> | 1905 | 3.02E-06 | 240 | 248 | 267 | 470 | 59 | 49 | 208 |
| | | | std dev -> | 190 | 1.22E-07 | 81 | 89 | 82 | 122 | 29 | 19 | 62 |

## 3.2 Citizen scientist evaluation results (Phase 2)

Reference flows evaluated in Phase 2 ranged from 4.2 to 896 L s$^{-1}$ (Table 5). Absolute errors for "expert" measurements averaged 41, 21, and 43 %, while biases for all methods were positive, averaging 41, 19, and 40 % for float, salt dilution, and Bernoulli methods, respectively (Table 5 and Fig. 4). Standard deviations of "expert" errors were 34, 26, and 51 % for float, salt dilution, and Bernoulli methods, respectively. Salt dilution calibration coefficients (k) averaged 2.95E-06 and ranged from 2.62E-06 to 3.42E-06. Measurement sites in the Dhobi watershed were pool and drop stream types, with slopes ranging from 0.076 to 0.148 m m$^{-1}$. Streambeds for these sites were predominantly cobles, gravels, and sands. Smaller tributaries measured in the Nakkhu watershed (N2, N4, and N6) were also pool and drop stream types with slopes of 0.105, 0.091, and 0.055 m m$^{-1}$, respectively. The remainder of the sites in the Nakkhu watershed were pool and riffle stream types with slopes ranging from 0.020 to 0.075 m m$^{-1}$.

**Table 5: Summary of (Phase 2) measurement comparison sites including salt dilution calibration coefficient (k), resulting reference flows (Q Reference), "expert" streamflow measurement method flows (Q Float, Q Salt, and Q Bernoulli), and corresponding "expert" measurement errors. Date and time associated with "expert" measurements, and represent the time that the expert ODK form was started in the field. Latitude and longitude in reference to the WGS84 datum. All flow values shown are shown in L s$^{-1}$ rounded to the nearest integer for values greater than or equal to 10 and to the nearest tenth place for values less than 10. Percent differences (errors) calculated using Q Reference (FlowTracker) as the actual flow. Data summarized at the bottom with average, minimum (min), maximum (max), and standard deviation (std dev). Note that averages (avg *) shown in the summary area near**

the bottom for the last three columns (i.e. percent errors) include averages of absolute values of percent errors (i.e. absolute errors) shown underlined in parentheses.  Average k (2.95E-06) was used to compute Q salt for all Phase 2 and 3 sites.

| Site ID | Date | Time | Latitude | Longitude | k (cm μS$^{-1}$) | Slope (m m$^{-1}$) | Q Reference (L s$^{-1}$) | Expert Q Float (L s$^{-1}$) | Expert Q Salt (L s$^{-1}$) | Expert Q Bernoulli (L s$^{-1}$) | Expert % Error Float | Expert % Error Salt | Expert % Error Bernoulli |
|---|---|---|---|---|---|---|---|---|---|---|---|---|---|
| D1 | 18/09/18 | 14:42 | 27.79246 | 85.37166 | 2.76E-06 | 0.099 | 137 | 150 | 134 | 122 | 10 | -2 | -11 |
| D2 | 18/09/18 | 15:46 | 27.79263 | 85.37158 | 2.70E-06 | 0.091 | 253 | 364 | 258 | 356 | 44 | 2 | 41 |
| D3 | 18/09/18 | 13:41 | 27.79213 | 85.37136 | 2.62E-06 | 0.076 | 417 | 551 | 500 | 396 | 32 | 20 | -5 |
| D4 | 18/09/18 | 12:44 | 27.79189 | 85.37162 | 2.69E-06 | 0.139 | 78 | 77 | 84 | 81 | -1 | 7 | 3 |
| D5 | 19/09/18 | 10:18 | 27.79071 | 85.36966 | 2.80E-06 | 0.148 | 184 | 243 | 207 | 287 | 32 | 12 | 56 |
| D6 | 19/09/18 | 11:52 | 27.79052 | 85.36695 | 3.42E-06 | 0.134 | 36 | 84 | 47 | 88 | 132 | 30 | 146 |
| D7 | 19/09/18 | 13:11 | 27.78791 | 85.36912 | 2.87E-06 | 0.126 | 55 | 60 | 86 | 52 | 10 | 56 | -6 |
| N1 | 20/09/18 | 17:35 | 27.56525 | 85.31356 | 2.90E-06 | 0.025 | 437 | 699 | 548 | 540 | 60 | 25 | 24 |
| N2 | 20/09/18 | 16:59 | 27.56615 | 85.31214 | 3.37E-06 | 0.105 | 4.2 | 7.3 | 4.0 | 11 | 73 | -5 | 158 |
| N3 | 20/09/18 | 16:02 | 27.56935 | 85.31277 | 2.93E-06 | 0.075 | 340 | 392 | 548 | 445 | 15 | 61 | 31 |
| N4 | 20/09/18 | 15:21 | 27.56916 | 85.31200 | 2.71E-06 | 0.091 | 25 | 40 | 27 | 33 | 61 | 8 | 33 |
| N5 | 20/09/18 | 12:56 | 27.57328 | 85.31263 | 3.08E-06 | 0.022 | 407 | 607 | 700 | 545 | 49 | 72 | 34 |
| N6 | 20/09/18 | 13:33 | 27.57408 | 85.31226 | 2.95E-06 | 0.055 | 105 | 151 | 103 | 136 | 44 | -2 | 30 |
| N7 | 20/09/18 | 11:50 | 27.57558 | 85.31269 | 3.35E-06 | 0.044 | 896 | 944 | 814 | 839 | 5 | -9 | -6 |
| N8 | 20/09/18 | 10:59 | 27.57516 | 85.31345 | 3.11E-06 | 0.020 | 270 | 382 | 284 | 453 | 41 | 5 | 68 |
| | | | | **avg * ->** | 2.95E-06 | 0.083 | 243 | 317 | 290 | 292 | 41 (41) | 19 (21) | 40 (43) |
| | | | | **min ->** | 2.62E-06 | 0.020 | 4.2 | 7.3 | 4.0 | 10.8 | -1 | -9 | -11 |
| | | | | **max ->** | 3.42E-06 | 0.148 | 896 | 944 | 814 | 839 | 132 | 72 | 158 |
| | | | | **std dev ->** | 2.62E-07 | 0.043 | 235 | 281 | 265 | 244 | 34 | 26 | 51 |

Box plots of CS Flow group errors combined with "expert" measurement errors for float (a), salt dilution (b), and Bernoulli (c) methods show that errors, for both "expert" and CS Flow groups, are least for the salt dilution method (Fig. 4).  The number of CS Flow group measurements used to develop individual box plots ranged from 6 to 12 for each site and totalled 117 for all 15 sites.  Two groups measured site D3 twice, so even though there were only 10 groups, there were 12 measurements available for comparison for this site.  For the remainder of sites (except N5), problems with either capturing, compressing, uploading, or interpreting the video of EC used for determining salt dilution flow limited the number of usable measurements to less than the number of groups (i.e. 10).  Absolute errors for CS Flow group measurements averaged 63, 28, and 131 %, while biases for all methods were positive, averaging 52, 7, and 127 % for float, salt dilution, and Bernoulli methods, respectively.  Standard deviations of CS Flow group errors were 82, 36, and 225 % for float, salt dilution, and Bernoulli methods, respectively.

For the float method (Fig. 4.a), 13 median CS Flow group errors were positive, while two sites (i.e. D3 and N7) were negative. Float "expert" errors (i.e. red circles) were within the interquartile range (IQR; blue boxes between the first and third quartile) of CS Flow group errors for 10 out of 15 sites. One float "expert" error and 21 CS Flow group errors were over 100 %. Float error medians and distributions were more variable in the Dhobi watershed than the Nakkhu watershed. For the salt dilution method (Fig. 4.b), seven median CS Flow group errors were positive, while eight were negative. Salt dilution "expert" errors (i.e. red circles) were within the IQR of CS Flow group errors for 7 out of 15 sites. Zero salt dilution "expert" errors and two CS Flow group errors were over 100 %. Salt dilution error distributions were more compact for the Dhobi watershed compared to the Nakkhu watershed. For the Bernoulli method (Fig. 4.c), all 15 median CS Flow group errors were positive. Bernoulli "expert" errors (i.e. red circles) were within the IQR of CS Flow group errors for 3 out of 15 sites. Two Bernoulli "expert" errors and 50 CS Flow group errors were over 100 %. Similar to float results, Bernoulli error medians and distributions were more variable in the Dhobi watershed than the Nakkhu watershed.

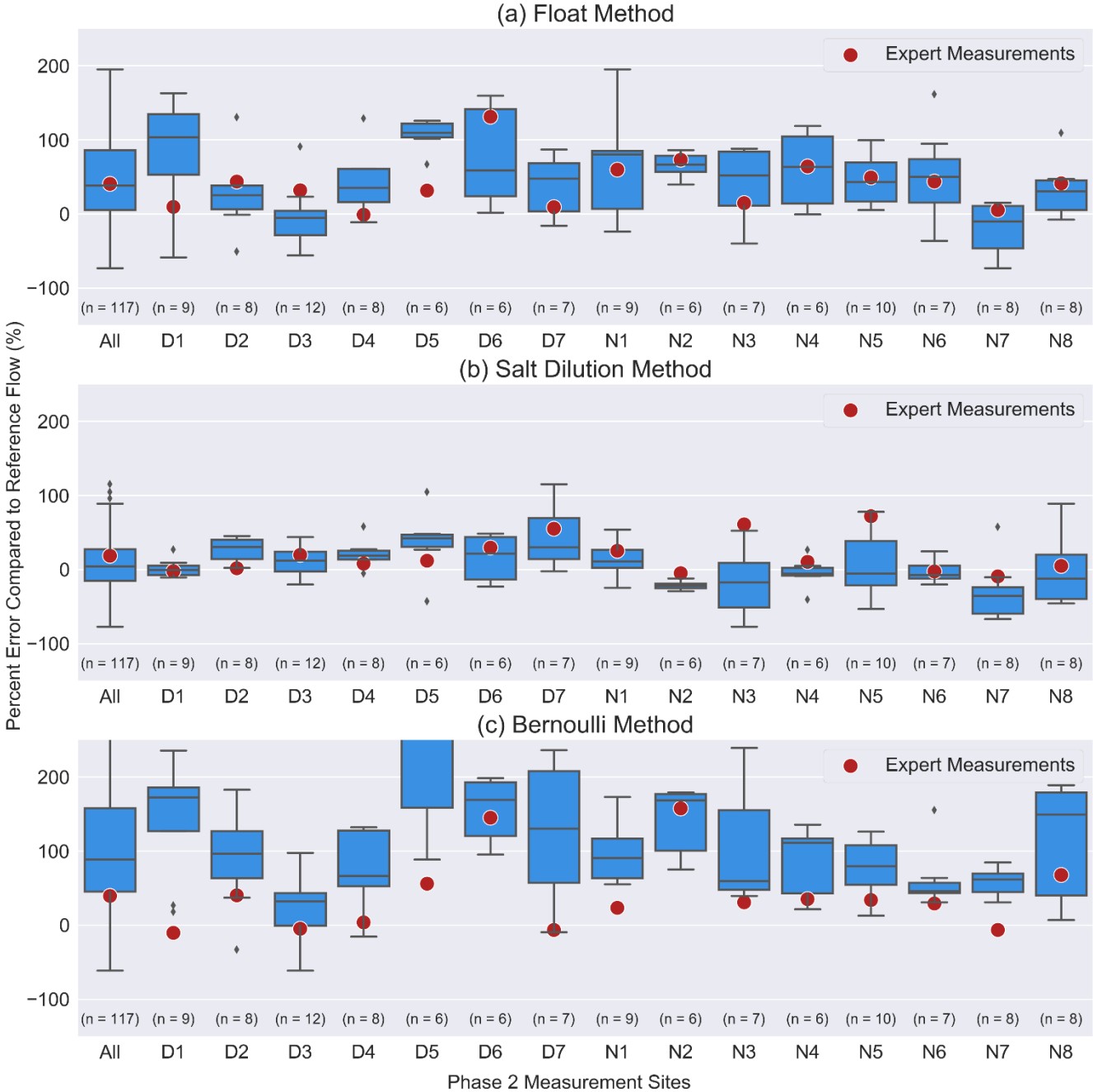

**Figure 4:** Box plots showing distribution of CS Flow group percent errors compared to reference flows for (a) float, (b) salt dilution, and (c) Bernoulli streamflow measurement methods. A summary of "All" measurements followed by the 15 Phase 2 measurement sites (i.e. D1 to D7 in the Dhobi watershed and N1 to N8 in the Nakkhu watershed) are shown on the horizontal axes. Percent errors for "expert" measurements for each site and method are shown as red circles. The "expert" measurements shown for "All" are the mean of all "expert" measurements for each method. Sample sizes for each method and each site are shown in parentheses above each site label. Boxes show the interquartile range between the first and third quartiles of the dataset, while whiskers extend to show minimum and maximum values of the distribution, except for points that are determined to be "outliers" (shown as diamonds),

which are more than 1.5 times the interquartile range away from the first or third quartiles. To facilitate comparison between sub-panels, vertical axes are fixed from -150 to 250 percent. In certain cases, portions of the error distribution are outside of the fixed range (e.g. Site D5 for (c) Bernoulli method).

Overall, citizen scientists ranked the float method most favourably (43.2 % of Rank 1 selections; average of blue bars) compared to Bernoulli and salt dilution methods, at 30.3 and 26.5 %, respectively (Fig. 5). In contrast, citizen scientists ranked the salt dilution method least favourably (64.0 % of Rank 3 selections; average of tan bars) compared to Bernoulli and float methods, at 18.6 and 17.4 %, respectively. Most citizen scientists (72.7 %) thought the float method required the least amount of training (Q1), followed by the Bernoulli and salt dilution methods. Citizen scientists thought the Bernoulli method required the smallest investment in equipment (45.5 %; Q2), the fewest number of citizen scientists (54.5 %; Q3), and least amount of data recording (42.4 %; Q4). Additionally, citizen scientists found the float method to be the least complex (48.5 %; Q5), most enjoyable (60.6 %; Q6), and safest (42.4 %; Q7) method. Finally, most citizen scientists (75.8 %) thought the salt dilution method was most accurate (Q8), followed by the float and Bernoulli methods. The complete results from the survey are included as supplementary material.

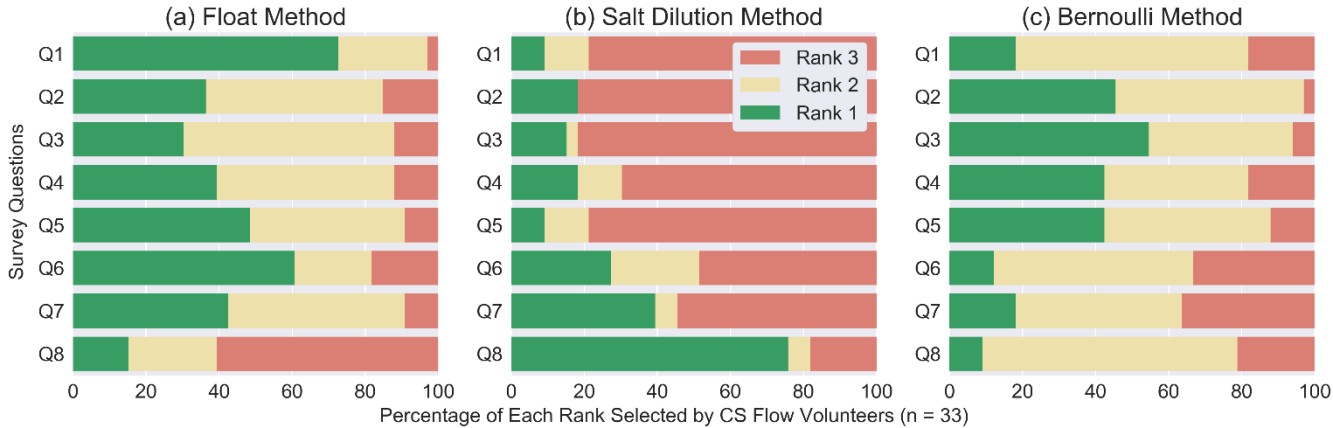

Figure 5: Results of the CS Flow group perception questions for (a) float, (b) salt dilution, and (c) Bernoulli methods. Questions Q1 through Q8 are shown on the vertical axis. Percentage of each rank selected by CS Flow citizen scientists (n = 33) are shown on the horizontal axis. Questions were worded so that in all cases, a rank of 1 was most favourable and 3 was least favourable. Questions are as follows (also included in Table 3): Q1 - Required training (1 least and 3 most); Q2 - Cost of equipment (1 least and 3 most); Q3 - Number of citizen scientists required (1 least and 3 most); Q4 - Data recording requirements (1 least and 3 most); Q5 - Complexity of procedure (1 least and 3 most); Q6 - Enjoyability of measurement (1 most enjoyable and 3 least enjoyable); Q7 - Safety (1 safest and 3 least safe); Q8 - Accuracy (1 most accurate and 3 least accurate).

### 3.3    Citizen scientist application results (Phase 3)

Observed flows from the CS Flow campaign (n = 131 pre-monsoon; n = 133 post-monsoon) were distributed among the 10 perennial headwater catchments of the Kathmandu Valley and ranged from 0.4 to 425 L s$^{-1}$ and 1.1 to 1804 L s$^{-1}$ in the pre- and post-monsoon, respectively (Fig. 6.a and 6.b). The three locations in the Kathmandu Valley where the Nepal Department of Hydrology and Meteorology (DHM) measures either water levels or flows (gauges) are included on Fig. 6.a and 6.b to

illustrate the difference in spatial resolutions between the two datasets. Note that only one of the three DHM gauging stations is in a headwater catchment (i.e. Bagmati). Histograms of flow (Fig 6.c and 6.d) and EC (Fig. 6.e and 6.f) show the increase in flows and the expected decrease in EC from pre- to post-monsoon.

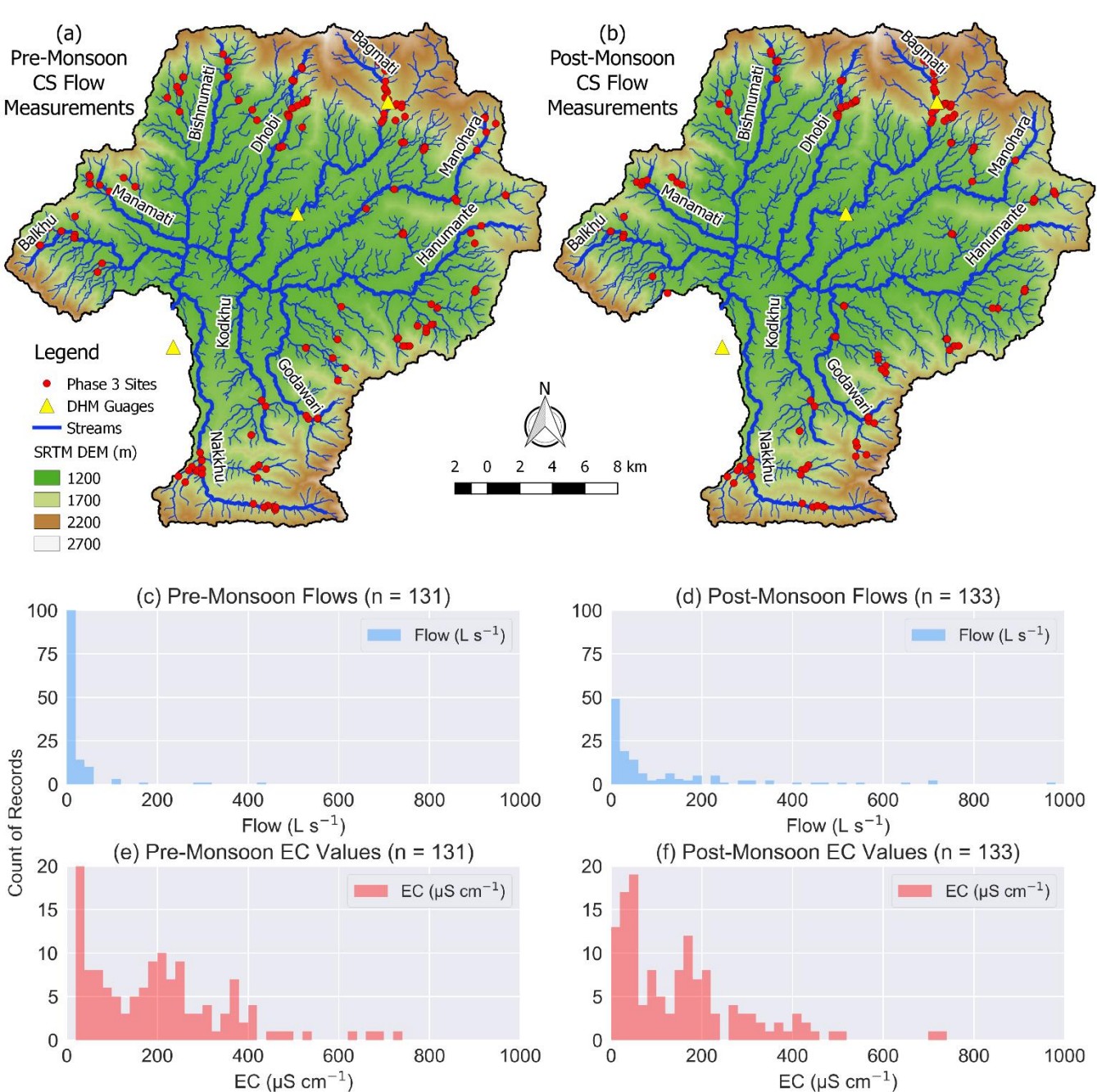

**Figure 6: CS Flow campaign measurement locations (n = 131 pre-monsoon; n = 133 post-monsoon) within the Kathmandu Valley for (a) pre- and (b) post-monsoon. Histograms show distributions of measured flows in L s$^{-1}$ ((c) and (d)) and EC in µS cm$^{-1}$ ((e) and (f)). Bins are set to 20 units wide for both flow and EC. Three flow measurements for the post-monsoon (d) that were above 1000 L s$^{-1}$ are not shown: 1059, 1287, and 1804. Three Department of Hydrology and Meteorology (DHM) gauging stations shown as yellow triangles.**

## 4    Discussion

Of the simple streamflow measurement methods evaluated in this paper, salt dilution provides the most accurate streamflow measurements for both "experts" and citizen scientists alike. In both Phases 1 and 2, salt dilution method resulted in the lowest absolute errors and biases (Table 6) compared to float and Bernoulli methods.

**Table 6: Summary of average absolute errors (Avg Abs Error), average biases (Avg Bias), and error standard deviations (Std Dev Error) for Phase 1 and 2 measurements. All values shown as percentages rounded to the nearest integer.**

| Phase | Performed by | Metric | Float Method | Salt Dilution Method | Bernoulli Method |
|---|---|---|---|---|---|
| 1 | Authors | Avg Abs Error (%) | 23 | 15 | 37 |
| | | Avg Bias (Avg Error (%)) | 8 | 6 | 26 |
| | | Std Dev Error (%) | 29 | 19 | 62 |
| 2 | "Expert" (Authors) | Avg Abs Error (%) | 41 | 21 | 43 |
| | | Avg Bias (Avg Error (%)) | 41 | 19 | 40 |
| | | Std Dev Error (%) | 34 | 26 | 51 |
| 2 | CS Flow Groups | Avg Abs Error (%) | 63 | 28 | 131 |
| | | Avg Bias (Avg Error (%)) | 52 | 7 | 127 |
| | | Std Dev Error (%) | 82 | 36 | 225 |

### 4.1    Initial evaluation discussion (Phase 1)

Our first research question was: *Which simple streamflow measurement method provides the most accurate results when performed by "experts?"* Based on Phase 1 "expert" measurements, we found that salt dilution had the lowest absolute error (i.e. 15 %), compared to float and Bernoulli methods (i.e. 23 and 37 %, respectively; Table 4).

The largest salt dilution errors occurred for reference flows of 21 L s$^{-1}$ or less, while float and Bernoulli errors appeared to be more evenly distributed through the range of observed flows. Because salt dilution measurements of low flows require less salt and water, it is possible that larger relative measurement errors caused while measuring these small quantities led to larger overall measurement errors. However, this is not substantiated in Phase 2 results, so additional research is required in this area.

Our experience in the field was that float velocity measurements in slow moving and shallow areas were difficult to perform. The combination of turbulence and boundary layer impacts from the streambed and the overlying air mass often made floating objects on the surface travel in non-linear paths, adding uncertainty to distance and time measurements. In the literature, challenges with applying the float method in shallow depths is supported by USBR (2001) and Escurra (2004), who showed that uncertainty in surface velocity coefficients (i.e. the ratio of surface velocity to actual mean velocity of the underlying water column; C from Eq. (1)) increased as depth decreased, especially below 0.3 m. The impacts of shallow depths on surface velocity coefficient C should be the focus on additional research.

A primary challenge we experienced with Bernoulli measurements was keeping the flat plate at the same vertical location while rotating the plate from parallel to perpendicular to the flow direction (Sect. 2.2.3). This was usually due to the bottom of the flat plate being set on a streambed consisting of sands and gravels that could be easily disturbed during rotation. Slow water velocities, and correspondingly small changes in Bernoulli depths (Eq. 4) further compounded this issue. Adding a circular metal plate to the bottom of the flat plate used for Bernoulli depth measurements could help minimize these uncertainties.

Based on the ten measured k values in Phase 1, using an average k for all salt dilution measurements caused the largest percent difference in salt dilution flow (Eq. 2) for site 7 (8.6 % increase in flow) followed by site 19 (7.6 % decrease in flow). For Phase 2, using average k values for all salt dilution measurements caused the largest percent difference in salt dilution flow (Eq. 2) for site D6 (13.7 % decrease in flow) followed by site D3 (12.6 % increase in flow). Because observed absolute error distributions from Phase 1, and especially Phase 2, are larger than errors introduced by using average k values (sometimes by more than an order of magnitude), we do not think our overall findings are negatively impacted by using average k values. However, because of the sensitivity of salt dilution measurements to k (Eq. 2), future work should focus on improving understanding of the variables affecting k. Specifically, spatial and temporal variability in k due to changes in stream water chemistry should be investigated prior to applying the salt dilution methodology described in this paper in other areas. For citizen science projects in other areas, we recommend that locally appropriate average k values be determined from measurements at multiple sites to understand spatial variability. Additional k measurements should also be repeated in different seasons to understand temporal variability.

### 4.2    Citizen scientist evaluation discussion (Phase 2)

Our second research question was: *Which simple streamflow measurement method provides the most accurate results when performed by citizen scientists?* Based on Phase 2 citizen scientist measurements, we found that salt dilution had the lowest absolute error (i.e. 28 %) compared to float and Bernoulli methods (i.e. 63 and 131 %; Fig. 4).

While absolute error distributions for citizen scientists followed the same trend to that of "expert" measurements, the relative increases in errors for float (41 to 63 %; increase of 54 %) and Bernoulli (43 to 131 %; increase of 205 %) were larger than that of salt dilution (21 to 28 %; increase of 33 %). This could be due in part to the fact that salt dilution measurement errors may be less sensitive to a lack of field data collection experience. For example, as long as turbulent mixing conditions are present (which can be controlled by proper site selection during the experimental design phase), citizen scientists can primarily introduce errors into salt dilution measurements by (1) making mistakes in measurement or recording of amounts of salt and/or water used to prepare tracer solutions, (2) not thoroughly mixing tracer solution until all salt is dissolved, (3) not providing enough distance between salt injection and EC measurement points (recommended as 25 stream widths by Day, 1977; Butterworth et al., 2000; Moore, 2005), or (4) recording videos of EC changes that are difficult to read. Each of these sources of error can be minimized by implementing relatively easy to follow protocols like "be sure to mix the salt and water until you can't see the salt any longer." In contrast, while performing float and Bernoulli measurements, citizen scientists need to accurately characterize (1) average stream depth, (2) stream width, and (3) average water velocity. Characterizing average depth and velocity requires several individual measurements, each coming with the chance of introducing measurement errors. Additionally, selecting the number of sub-sections required, and selected representative locations for each of these sub-sections can be difficult, even for people with extensive streamflow data collection experience. These factors may help explain the wider error distributions observed in float and Bernoulli methods compared to salt dilution (Fig. 4). Additional training might also help to close the observed differences between salt dilution error distributions and that of float and Bernoulli methods.

Our third research question was: *What are citizen scientists' perceptions of the required training, cost, accuracy, etc. of the evaluated simple streamflow measurement methods?* Based on a survey of 33 citizen scientists, we found that volunteers ranked the float method most favourably (43.2 % of Rank 1 selections) compared to Bernoulli and salt dilution methods, at 30.3 and 26.5 %, respectively (Fig. 5).

Regarding question number 4 from the perception survey (i.e. data recording requirements), it is interesting to note that salt dilution received the least favourable ranking, meaning that citizen scientists perceived salt dilution to require the greatest amount of data. Our perception was that salt dilution, in terms of individual pieces of information, requires the least amount of data recording. This ranking may be explained by either (1) the amount of meta data collected about salt dilution measurements (i.e. GPS and photos of salt injection and EC measurement locations; see Sect. 2.4.2 and supplementary material for details) or by (2) citizen scientists' perception of using a digital EC meter and smartphone video as recording lots of individual pieces of data, when in some ways a video can be thought of as a single observation. Whereas results from float and Bernoulli method measurements are available immediately in the ODK from, the post processing requirements of EC breakthrough curve data to solve for salt dilution flow may also lead to the perception that salt dilution measurements have higher data recording requirements.

Citizen scientists ranked float method safest, followed by salt dilution, and finally Bernoulli. We found this result to be somewhat counter intuitive, because salt dilution is the only method that can be performed without entering the stream, whereas for float and Bernoulli measurements the entire stream must be waded across to get depth and velocity data. Because the perception survey was performed after Phase 2 evaluations where all three methods were performed consecutively, it may not have been obvious to citizen scientists that salt doses could be obtained without entering the stream from visual estimates of channel width, depth, and water velocity.

In terms of perceived measurement accuracy (question 8), 75.8 % of citizen scientists ranked salt dilution as the most accurate method. This ranking was performed before any quantitative results were reviewed. Our experience is that reading a value from a digital meter often gives an unfounded sense of measurement accuracy. Salt dilutions' perceived accuracy may be due to it being the only method that directly involves a digital measurement device (i.e. EC meter).

"Expert" absolute errors for float, salt dilution, and Bernoulli increased from 23, 15, and 37 % in Phase 1 to 41, 21, and 43 % in Phase 2. For the float method, this increase in error may be partially explained by the overall increase in flows from pre-monsoon (Phase 1; average reference flow of 92 L s$^{-1}$) to post-monsoon (Phase 2; average reference flow of 243 L s$^{-1}$). Our experience was that increased flow and velocity in high gradient headwater streams made it more difficult to perform float measurements. This was mostly due to an increase in turbulence resulting in more non-linear flow lines and increased relative measurement uncertainty for shorter float times (assuming distances were held constant). For the Bernoulli method however, our hypothesis was that increased velocities would on average reduce measurement errors, because of decreased relative measurement uncertainty for larger Bernoulli depth changes. This hypothesis however was not supported by the data. The challenge of pulsing flows which require citizen scientists to visually average short period (i.e. seconds or less) water level fluctuations may also counteract the otherwise larger Bernoulli depth changes. We do not have any explanations for the overall increase in salt dilution method absolute error from 15 to 21 % from Phase 1 to Phase 2. Unlike the Phase 1 results, we also do not see a concentration of larger errors at the lower reference flows in Phase 2.

### 4.3    Citizen scientist application discussion (Phase 3)

To proceed with Phase 3, we had to select a preferred simple streamflow measurement method. Based on the results from Phases 1 and 2, the salt dilution method had the lowest absolute errors, biases, and error standard deviations for both "experts" and citizen scientists. Therefore, from an accuracy perspective, salt dilution was the preferred approach. However, the results of our perception survey showed that citizen scientists thought the float method was most enjoyable (Q6) and required the least amount of training (Q1). Another important consideration was that salt dilution is the only method that doesn't require citizen scientists to enter and cross the stream, and therefore can be safely performed over a broader range of flow conditions. While the enjoyment of measurements is an important motivational factor for citizen scientists, we concluded that accuracy

and safety were ultimately more important. Considering all these factors, we selected the salt dilution method as the preferred approach.

Finally, our fourth research question was: *Can citizen scientists apply the selected streamflow measurement method at a larger scale?* Based on measurements from pre- (n = 131) and post-monsoon (n = 133) in the Kathmandu Valley, citizen scientists can apply salt dilution streamflow measurements at a larger scale; however, challenges of recruiting, training, and motivating citizen scientists, along with data management issues require further investigation.

The CS Flow campaigns provided us with a unique opportunity to evaluate the preferred salt dilution streamflow measurement method with more people at more sites. In addition to the valuable streamflow data that will help us characterize the water supply situation in the Kathmandu Valley with greater precision for pre- and post-monsoon periods, we also learned several practical lessons about how to scale citizen science-based streamflow measurements. For example, our experience was that digitizing breakthrough curves from ODK captured EC videos took roughly 15 to 30 minutes per site, depending on video length and quality. Additionally, managing EC change videos can be a significant challenge if videos are recorded at a smartphones' native resolution. In some cases, each minute of high definition video can be nearly 100 MB. Uploading such large files, and subsequently storing and accessing them can be challenging and costly. These difficulties can be solved by improved training and protocols regarding video collection settings and, when necessary, video compression.

## 5    Conclusions and future work

Compared to float and Bernoulli, the salt dilution method consistently yielded the most accurate streamflow measurement results for authors and citizen scientists alike. Given ongoing global declines in the amount of streamflow data being collected by traditional entities, salt dilution measurements performed by young researchers and citizen scientists could play an important role in closing this data gap. While globally applicable, this is especially true for headwater catchments in developing regions.

With regards to young researchers (i.e. science and engineering minded students from primary through graduate school ages), performing salt dilution streamflow measurements has the benefits of (1) filling data gaps and (2) improving the quality and applicability of students' educational experience. We suggest that science and engineering educators should make smartphone-based data collection activities a core component of their curricula. Moreover, these data should be collected together with globally active partners to ensure standardization and open access to data.

As a step in this direction, SmartPhones4Water and S4W-Nepal in partnership with local educators are working towards broader applications of salt dilution streamflow measurements in Nepal and beyond. Importantly, variability in the calibration

coefficient (k) should be evaluated over larger ranges of time, geology, and water quality. Another practical challenge requiring specific attention is the transfer, management, and digitization of break through curve video files. The information content of additional headwater streamflow data should be explored, especially regarding the trade-offs between observation density and accuracy. Efforts should focus on how to effectively recruit and motivate young researchers and citizen scientists

to participate in citizen science streamflow measurements. Lastly, emphasis should be placed on exploring these and other citizen science related questions in the relatively unexplored Asian context.

## 6 Data availability

The data used in this paper are provided as supplementary material.

## 7 Author contribution

Jeffrey C. Davids had the initial idea for this investigation and designed the experiments in collaboration with Martine M. Rutten, Wessel David van Oyen, and Nick van de Giesen. Field work was performed by Jeffrey C. Davids, Anusha Pandey, Nischal Devkota, Wessel David van Oyen, and Rajaram Prajapati. Jeffrey C. Davids prepared the manuscript with valuable contributions from all co-authors.

## 8 Competing interests

The authors declare that they have no conflict of interest.

## 9 Acknowledgements

This work was supported by the Swedish International Development Agency under Grant number 2016-05801; and by SmartPhones4Water (S4W). We appreciate the dedicated efforts of Annette van Loosen, Bhumika Thapa, Sunil Duwal, citizen

scientists from Khwopa College of Engineering, Anurag Gyawali, Anu Grace Rai, Sanam Tamang, Eliyah Moktan, Surabhi Upadhyay, Amber Bahadur Thapa, Pratik Shrestha, Kristi Davids, and the rest of the S4W-Nepal team of young researchers. Thanks to Kate Happee, Niek Moesker, Nick N. Overkamp, and Rick van Bentem from the 2018 multi-disciplinary group of Master's students from Delft University of Technology for their fresh energy during post-monsoon field work. We would also like to thank Dr. Ram Devi Tachamo Shah, Dr. Deep Narayan Shah, Dr. Narendra Man Shakya, and Dr. Steve Lyon for their

supervision and support of this work. A special thanks to SonTek for their donation of a FlowTracker acoustic Doppler velocimeter that was used for the reference flow measurements discussed in this paper and many more to come. Finally, thanks to the reviews for their useful comments.

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
