# Peer review of "Citizen science flow - an assessment of simple streamflow measurement methods"

_Hydrology and Earth System Sciences, 2018_

## Referee Comment (RC1) · Anonymous Referee #1 · 15 Sep 2018

This manuscript aims at studying the potential for citizen science streamflow measurement methods. Citizen science is so far underused in hydrology and studies on this topic are, thus, much welcome. The manuscript starts with a well-written introduction, where several relevant studies are cited. After this promising start, however, I was rather disappointed by the study. I really like the aim of this study, and I appreciate the attempt to evaluate the suitability of different streamflow gauging methods, but in the end, I have three major concerns. These are related to 1) the study design and data collection, 2) the data analyses and 3) statements that are not supported by the data analyses presented in the manuscript.

A more detailed discussion of these issues and some minor comments are provided below.

[Figure]

I am afraid that the concerns related to the available data require additional data to be collected. Frankly, I would say the presented work is an interesting pre-study, but a better study design and data collection are needed to obtain useful results. Publishing the preliminary results as presented here could do more harm than good as people might use the conclusions without being aware that there actually was little data evidence. Given the importance of the topic, I hope the authors will be able to do this and will resubmit a study, which addresses the issues they raise in this manuscript.

1) There are several severe flaws in the study design and in the end I am afraid the authors did not collect the data that would be needed to address the questions they wanted to study.

a. It is highly unfortunate that there are no concurrent flow measurements for the 'true' flow available. Flow measurements taken a few weeks apart are just not the basis for a serious evaluation. It is also surprising the different 'citizen scientists' were asked to measure streamflow at different sites. It would have much more informative to let them measure the same stream and about the same conditions.

b. The authors mention that three salt dilution measurements were excluded as outliers. While they present some explanation (which I do not fully agree), they do not present anything that would help to detect such cases in an application where there is no comparison with any other gauging. In other words, in a real application, these values would pass undetected, and the potential error, thus, would be much larger than reported here. Note that almost half of the cases with comparison streamflow data were excluded! Again, it is unfortunate that the authors need to very speculative about what might have happened because of the study design.

c. Related to the above comment, one potential issue are mistakes that could be done by 'citizen scientists'. With a better study design (e.g., more groups at the same place, 'secret' observer, . . .), this could have been addressed.

d. Basically, there are two separate questions: 1) which of the 'simple' gauging methods provides best results (with 'perfect' persons) and 2) how re the methods used

by 'citizen scientists'. By deciding the best methods already after the first step, the authors, unfortunately, do not fully explore which method is most suitable for citizen science approaches.

e. The accuracy of the salt dilution measurement depends largely on the selected site (mixing, flow volume, and velocity...), and depending on the site, thus, different methods might be most suiatble. Again, this is an important aspect that could have been adddressed with a better study design.

f. A minor point related to the study design: when the aim is to obtain relations of the calibrated k-factors with elevation or other variables, k should have been determined at as may places as possible and not just half of them. I am also not sure whether it is reasonable to use the mean k value was for the 10 locations without individual measurements, but the individual values for the others. I would rather have expected to use the mean or some regionalized values for ALL locations to ensure comparability.

2) The data analyses contain some questionable use of statistics:

a. Averaging of errors (tables 12): averaging positive and negative errors just does not make any sense, this makes the results look much better than they are. Instead, one should base the analyses on the absolute values so that positive and negative errors do not cancel out each other

b. The correlations shown in figures 3 or 4 (and reported in the abstract) are misleading. These are spurious correlations! Comparing streams of different size, of course, one gets high r2 values. Imagine two persons would measure the height of a group of people, even if the individual measurements would be off by 5 cm, the correlation of the heights would still be large simply because some people are much taller than others. Please be more careful when using statistics.

3) The statements in section 4.4 are not really supported by the data in this study. The number of persons needed in each group, for instance, has not been tested. Also, the inexpensive EC meter has not been tested (or has it? Comparison?).

As another example, the statement of enjoyability seems not to be supported, and actually, the other methods have not been tested with 'citizen scientists.

The authors also need to explain much better which type of citizen scientist they refer to. Form the title (where the term citizen science is used twice!), the abstract, the introduction and section 4.4 one gets the impression that this is about citizen science in a broad meaning. However, looking more closely at what has been done, it seems that the work does not address the participation of the general public in science but is based on selected individuals, which received a significant education. This is fine, but is a rather special case of citizen science.

4) Minor comments:

a. P4: which factor for c was used in the end? Variable or constant. This needs to be included in the steps.

b. P8: too little information is given about the 'citizen scientists': how old? Gender? Students, but which topic (how much hydrology or environmental engineering?), how large groups, . . ..

c. What is the purpose of showing figure 2?

d. P5L15: where does the value of 1667 g per m3/2 come from. Moore (2005) recommend a different value

e. Tables 1 and 2: providing runoff with four digits seems a bit too accurate, especially given that the observations actually were weeks apart.

f. Please check the author guidelines, especially with regard to the date format and equations

---

## Referee Comment (RC2) · Anonymous Referee #2 · 15 Sep 2018

The paper presents results on three simple and easy to use discharge estimation methods appropriate for citizen science (SC) that the authors applied in the Kathmandu Valley, Nepal. They assessed the agreement between the methods and compared estimated discharge to selected measurements using a doppler radar device. The text is short but mainly well written, the graphical presentation is clear and appealing. I recommend to state explicit research questions at the end of the introduction (currently missing). I have also some major concerns about parts of the analysis and the interpretation of the results: 1) While the authors do well in terms of reporting statistical significance of their results, the use of the Pearson Correlation Coefficients seems not appropriate for the properties of the dataset. I therefore recommend the non-parametric Spearman Rank Correlation and the associate non-parametric statis-

tical test. 2) I question whether it is meaningful or informative to correlate the slope of the salt dilution calibration k to latitude or longitude and elevation and would suggest to skip (or better explain this analysis). 3) Instead I recommend to also show the comparison of discharge estimated by salt dilution and by the Bernoulli method. 4) I would ask the authors to quantitatively prove that they can compare discharge estimates taken during the CS-campaign with doppler radar observed discharge taken +/- one month (!!) before/after the campaign or skip that part. In the discussion they state themselves that the flow might have decreased during that time. As the remaining analysis is probably too short for a full publication, I suggest the authors to check whether their dataset would allow additional analysis e.g., on the difference of the quality of the measurements taken by experts and citizens (see my suggestions in the pdf). The current paper is interesting, the dataset promising but the current state of analysis is not enough for a full publication. I therefore encourage the authors for major revisions and additional data analysis.

In the following I summarize my suggestions for the individual sections and ask the authors to also check my detailed comments and suggestions that I have included in the pdf (uploaded as supplement):

1) Introduction: The introduction is on the short side and starts a bit philosophical. I would focus more on streamflow and introduce to the problem that large parts of the words still have limited number of gauging stations (especially remote and developing countries) and that measuring devices - while still decreasing in costs - have their limitations. Some other citizen science studies are briefly mentioned but the findings of other studies could be described a bit more in detail. The same applies to the existing methods for low-cost streamflow assessments. Their pros and cons could be compared using a table. I also do not agree that the are no tools on the market, that allow direct measurements of discharge with smartphones and added one link to an example. The research questions should be clearly formulated at the end of the introduction. Please also see more specific comments directly marked in the pdf (uploaded as supplement).

2) Methods: The method section needs a better description of the experimental setup; study area, test with students, repetitions y/n etc. (parts are mentioned at the end of the method section but should be stated at the beginning)! The catchments and streams used for testing need a better description (see my suggestions in the pdf). The same applies to the training of the students. The explanation of the different methods is long but can be useful for some non-hydrological readers. I suggest to consider to present all this information in the introduction section. I would however include a list of objective criteria why these three and not other methods have been selected. Please also see more specific comments directly marked in the pdf (uploaded as supplement). The method section should be clearer about the two datasets collected a) dataset with n=20 samples (I assume collected by the authors themselves = exports) and the CS-campaign with n=145 samples collected by citizens. One issue seems critical to me: Authors compare observed discharge using the doppler radar with CS-discharge measurements done +/- 1 moth earlier/later. The authors should prove statistically that the mean daily flow in the month before and after the CS-discharge measurements is not significantly different. In fact authors state in the discussion section that flows decreased during that period.

3) Results: Graphical presentation of the results is good and I appreciate that the authors report about statistical significance of their results. For some of the dataset I suggest to used the Spearman Rank Correlation Coefficient as the assumptions for using the Pearson Correlation seem to be not fulfilled. I also suggest to mention that, while statistically significant, some of the relations show relations difficult to interpret (definitely not linear or exponential but complex or clustered). I ask the authors to explain why they think Figure 2 is informative to the reader expect for presenting the measurements. I question whether it is meaningful or informative to correlate the slope of the salt dilution calibration k to latitude or longitude and elevation and would suggest to skip this analysis. Instead I recommend to also show the comparison of discharge estimated by salt dilution and by the Bernoulli method. As mentioned in the method section I would ask the authors to quantitatively prove that they can compare discharge

estimates taken during the CS-campaign with doppler radar observed discharge taken +/- one month before/after the campaign or skip that part. Please also see more specific comments directly marked in the pdf (uploaded as supplement).

4) Discussion: The discussion is short and not very into depth. Parts of it would better fit into the result section. While the background concentration is certainly affected by the geology I have strong doubts that correlating k with latitude and longitude or elevation is meaningful. At least better explain why the authors think these predictors are meaningful and the correlations not spurious. I suggest the authors to check whether their dataset would allow additional analysis e.g., on the difference of the quality of the measurements taken by experts and citizens (see my suggestions in the pdf). 5) Summary and future work Is well written in general. However, addressing the outcome of this work in the light of research questions (that I suggest to include in the introduction) would improve this section.

I hope these suggestions are useful to work on a more advanced version of the manuscript!

Please also note the supplement to this comment:
https://www.hydrol-earth-syst-sci-discuss.net/hess-2018-425/hess-2018-425-RC2-supplement.pdf

**Supplement:**

[Figure]

**Citizen science flow - an assessment of citizen science streamflow measurement methods**

Jeffrey C. Davids[1,2], Martine M. Rutten[1], Anusha Pandey[3], Nischal Devkota[3], Wessel David van Oyen[4], Rajaram Prajapati[3], and Nick van de Giesen[1]

[1]Water Management, Civil Engineering and Geosciences, Delft University of Technology, TU Delft Building 23, Stevinweg 1, 2628 CN, Delft, Netherlands
[2]SmartPhones4Water-CA, 3881 Benatar Way, Suite G, Chico, CA, 95928, USA
[3]SmartPhones4Water-Nepal, Damodar Marg, Thusikhel, 44600, Lalitpur, Nepal
[4]Engineering and Applied Sciences, Rotterdam University, Rotterdam, G.J. de Jonghweg 4-6, 3015 GG, Rotterdam, Netherlands

*Correspondence to*: Jeffrey C. Davids (j.c.davids@tudelft.nl)

**Abstract.** Wise management of water resources requires data. Nevertheless, the amount of streamflow data being collected globally continues to decline. Involving citizen scientists to generate hydrologic data can potentially help fill this growing hydrological data gap. Our aim herein was to (1) evaluate three potential citizen science streamflow measurement methods (i.e. float, salt dilution, and Bernoulli run-up), (2) select a preferred approach, and (3) pilot test the selected approach at a larger scale. We performed 20 side-by-side evaluation measurements in headwater catchments of the Kathmandu Valley. We used mid-section measurements from an acoustic Doppler velocimeter as reference flows. Evaluated flows ranged from 0.006 to 0.240 $m^3$ $s^{-1}$. Linear regressions forced through the origin for scatter plots with reference flows had slopes of 1.05, 1.01, and 1.26 with r-squared values of 0.90, 0.98, and 0.61, for float, salt dilution, and Bernoulli run-up methods, respectively. After selecting the salt dilution method as the preferred approach, we performed larger scale pilot testing in a one-week Citizen Science Flow campaign (CS Flow) involving 20 volunteers. Observed flows (n = 145) ranged from 0.0004 to 0.425 $m^3$ $s^{-1}$ and were distributed among the 10 headwater catchments of the Kathmandu Valley. At locations with reference flows available (n = 5), a linear regression forced through the origin between reference flows and CS Flow measurements had a slope of 0.90 with an r-squared value of 0.97. Future work should evaluate the feasibility of applying citizen science salt dilution streamflow measurements to larger regions.

**1 Introduction**

Lord Kelvin, a 19th century Scottish physicist and mathematician, wisely said, "… the first essential step in the direction of learning any subject is to find principles of numerical reckoning and practicable methods for measuring some quality connected with it (Kelvin 1883)." With regards to our natural resources, if we aim to wisely steward them, we must first
 learn to measure them. While it might sound trivial, collecting and, worse yet, interpreting point measurements of precipitation, evapotranspiration, infiltration, and soil moisture at the catchment scale is fraught with challenges. Indeed, the

**Number: 1  Author: reviewer  Subject: Note  Date: 2018-09-14 10:49:54**

I think the first paragraph is a bit philosophical. I would focus more on streamflow and introduce to the problem that large parts of the words still have limited number of gauging stations (especially remote and developing countries) and that measuring devices - while still decreasing in costs - have their limitations.

The following paragraph is well suited as introduction!

[Figure]

importance of measuring streamflow is underpinned by the reality that it is the only truly integrated representation of the entire catchment that we can plainly observe (McCulloch 1996).

Despite growing demand, the amount of streamflow data actually being collected continues to decline in several parts of the

5  world, especially in Africa, Latin America, Asia, and even North America (Hannah et al. 2011; Van de Giesen et al. 2014; Feki et al. 2016; Tauro et al. 2018). Specifically, there is an acute shortage of streamflow data in headwater catchments (Kirchner 2006) and developing regions (Mulligan 2013). The reasons for this trend are various,
 but the situation is perpetuated by a lack of understanding among policy makers and citizens alike regarding the importance of streamflow data, which leads to persistent funding challenges (Kundzewicz 1997; Pearson 1998). This is further compounded by the reality

10  that the hydrological sciences research community has focused much of its efforts in recent decades on advancing modeling techniques, while innovation in methods for generating the data these models depend on has been relegated to a lower priority (Mishra and Coulibaly 2009).

Considering these challenges, alternative methods for generating streamflow and other hydrological data are being explored

15  (Tauro et al. 2018). For example, developments in using remote sensing to estimate streamflow are being made (Tourian et al. 2013; Durand et al. 2014), but applications in small headwater streams are expected to remain problematic (Tauro et al. 2018). Utilizing cameras for measuring streamflow is also a growing field of research (Muste et al. 2008; Le Coz et al. 2010; Dramais et al. 2011; Le Boursicaud et al. 2016), but it is doubtful that these methods will be broadly applied in headwater catchments in developing regions in the near future because of high costs and lack of technical capacity. In these

20  cases, however, involving citizen scientists to generate hydrologic data can potentially help fill the growing global hydrological data gap (Lowry and Fienen 2012; Buytaert et al. 2014; Sanz et al. 2014; Davids et al. 2017; van Meerveld et al. 2017; Assumpção et al. 2018).

Kruger and Shannon (2000) define citizen science as the process of involving citizens in the scientific process as researchers.

25  Citizen science often uses mobile technology (e.g. smartphones) to obtain georeferenced digital data at many sites, in a manner that has the potential to be easily scaled (O'Grady et al. 2016). Turner and Richter (2011) partnered with citizen scientists to map the presence or absence of flow in ephemeral streams. Fienen and Lowry (2012) showed that citizen science text message based measurements of water level can have acceptable errors. Mazzoleni et al. (2015) showed that flood predictions can be improved by assimilating citizen science water level observations into hydrological models. Le

30  Coz et al. (2016) used citizen scientist photographs to improve the understanding and modeling of flood hazards. Davids et al. (2017) showed that lower frequency observations like those produced by citizen scientists can provide meaningful hydrologic information. Van Meerveld et al. (2017) showed that citizen science observations of stream level class can be informative for deriving model based streamflow time series of ungauged basins.

**Number: 1   Author: reviewer   Subject: Note   Date: 2018-09-14 10:50:51**

various is rather vague: List some of the reasons!

**Number: 2   Author: reviewer   Subject: Note   Date: 2018-09-14 10:53:02**

add more recent papers e.g.
Burt & McDonnell (2015): Whither field hydrology ...WRR, (https://doi.org/10.1002/2014WR016839
)

[Figure]

While the previously referenced studies focus mainly on involving citizen scientists for observing stream levels, we were primarily concerned with the possibility of enabling citizen scientists to take direct measurements of streamflow. Using keyword searches using combinations of "citizen science", "citizen hydrology", "community monitoring", "streamflow monitoring", "streamflow measurements", and "discharge measurements," we could not find any specific work about how citizen scientists, equipped with modern tools like smartphones, could take streamflow measurements directly themselves. Instead, to develop potential citizen science streamflow measurement methods to evaluate further, we turned to the vast body of general knowledge about the collection of streamflow data.

Streamflow measurement techniques suggested in the United States Bureau of Reclamation Water Measurement Manual (USBR 2001) that seemed potentially applicable for citizen scientists included: deflection velocity meters consisting of shaped vanes projecting into the flow along with a method to measure deflection; the slope area method whereby the slope of the water surface in a uniform reach is measured and combined with the Manning formula; and pitot tubes for measuring velocity heads. The float and salt dilution methods described by several authors also seemed applicable (British Standards Institute 1964; Rantz 1982; Fleming and Henkel 2001; Escurra 2004; Moore 2004a, 2004b, and 2005; Herschy 2014). Finally, Wilm and Storey (1944) and Church and Kellerhals (1970) introduced the velocity head rod, or what we later refer to as the Bernoulli run-up method, involving measurement of stream velocity heads with a thin flat plate.

Based on these recommendations, the strengths and limitations discussed in the corresponding literature, and practical considerations about how citizen scientists could implement the different approaches, we selected three approaches for further evaluation: float, salt dilution, and Bernoulli run-up. Our primary aims in this paper were to (1) evaluate these three potential citizen science streamflow measurement methods, (2) select a preferred approach, and (3) pilot test the preferred approach at a larger scale.

**2 Materials and methods**

**2.1 Citizen science streamflow measurement methods evaluated**

The procedures for each of the three citizen science streamflow measurement methods evaluated are described in the following sections.

**2.1.1 Float**

The float method is based on the velocity-area principle. Total streamflow (Q) in cubic meters per second ($m^3 \ s^{-1}$) was calculated with Eq. (1):

Eq. (1) $\quad Q = \sum_{i=1}^{n} C * VF_i * d_i * w_i$

**Number: 1  Author: reviewer  Subject: Note  Date: 2018-09-14 11:01:02**

Maybe the key words were not appropriate to find these solution. There are some smartphone apps available that can do that e.g., http://www.photrack.ch/dischargeapp.html
Please delet the statement and widen your search. There are apps available that can be used widely

**Number: 2  Author: reviewer  Subject: Note  Date: 2018-09-14 11:08:32**

I think it would be good to better describe how citizen science can be one possible way to close the data gap but also mention difficulties!

**Number: 3  Author: reviewer  Subject: Note  Date: 2018-09-14 11:07:32**

the description of the methods could be more detailed

**Number: 4  Author: reviewer  Subject: Note  Date: 2018-09-14 11:04:30**

Add cits that hat used this method would add to the introduction)

**Number: 5  Author: reviewer  Subject: Note  Date: 2018-09-14 11:05:15**

.. crossestion area and the ...

**Number: 6  Author: reviewer  Subject: Note  Date: 2018-09-14 11:06:11**

explain the reader how the salt dilution method works in 1/2 or 1 senses like you did for the other methods

**Number: 7  Author: reviewer  Subject: Note  Date: 2018-09-14 11:10:24**

You should summarize the pros and cons and limitations of these methods for the reader. This is importent also for you to argue, why you chose the three ones for this study! Otherwise this choce is sunjective.

You could add a table with the pros and cons of each method and citations.

**Number: 8  Author: reviewer  Subject: Note  Date: 2018-09-14 17:44:41**

I could imagine to present the following information in the introduction section (and include the Manning-Strickler Method as another method  (that you later decided not to use) because the way you present it is a introduction to these methods. In the method section you could then more describe the experimental setup hwo you tested these methods for thier suitbility for CS.

**Number: 9  Author: reviewer  Subject: Note  Date: 2018-09-14 11:24:10**

I think this sentence is not necessary and can be skipped

**Number: 10  Author: reviewer  Subject: Replace  Date: 2018-09-14 11:33:59**

Float Method

**Number: 11  Author: reviewer  Subject: Note  Date: 2018-09-14 11:33:44**

Start with a description how the method works in general. You measure cross section (subdeviding it into n-sub-sections) and flow velocity and the apply a coefficients for accounting for friction losses

**Number: 12  Author: reviewer  Subject: Replace  Date: 2018-09-14 11:30:47**

is

where C is a unitless coefficient to account for the fact that surface velocity is typically higher than average velocity (typically in the range of 0.66 to 0.80 depending on depth; USBR 2001), $VF_i$ is surface velocity from float in meters per second (m s$^{-1}$), $d_i$ is depth (m), and $w_i$ is width (m) of each sub-section (i = 1 to n, where n is the number of stations).

5 Surface velocity for each sub-section was determined by measuring the amount of time it takes for a floating object to move a certain distance. For floats we used sticks found on site. Sticks are widely available (i.e. easiest for citizen scientists), generally float (except for the densest varieties of wood), and depending on their density are between 40 and 80% submerged, which minimizes wind effects.

10 Float measurements involved the following steps:

1. Selected stream reach with straight and uniform flow
2. Divided cross section into several sub-sections (n, typically between 5 and 20)
3. For each section, measured and recorded
    15  a. The depth in the middle of the sub-section
    b. The width of the sub-section
    c. The time it takes a floating object to move a known distance downstream (typically 1 or 2 m) in the middle of the sub-section
4. Solved for streamflow (Q) with Eq. (1)

20 **2.1.2 Salt dilution**

There are two basic types of salt dilution flow measurements: slug (previously known as instantaneous) and continuous rate (Moore 2004a). Salt dilution measurements are based on the principle of the conservation of mass. In the case of the slug method, a single known volume of high concentration salt solution is introduced to a stream and the electrical conductivity (EC) is measured over time at a location sufficiently downstream to allow good mixing (Moore 2005). In contrast,
25 continuous rate salt dilution method involves introducing a known flow rate of salt solution into a stream (Moore 2004b). Slug method salt dilution measurements are broadly applicable in streams with flows up to 10 m$^3$ s$^{-1}$ with steep gradients and low background EC levels (Moore 2005). For the sake of citizen scientist repeatability, we chose to only investigate the slug method, because of the added complexity of measuring the flow rate of the salt solution for the continuous rate method.

30 Streamflow (Q; m$^3$ s$^{-1}$) was solved for using Eq. (2) (Rantz 1982; Moore 2005):

Eq. (2)   $Q = \dfrac{V}{k \sum_{i=1}^{n}(EC(t) - EC_{BG})\Delta t}$

**Number: 1  Author: reviewer  Subject: Replace  Date: 2018-09-14 11:30:40**

is

**Number: 2  Author: reviewer  Subject: Note  Date: 2018-09-14 11:32:41**

Mention the difference between surface flow velocity and velocity at dept and flow velocity neat the banks and the middle of the stream. Mention also the difficulty that the float can get stuck.

**Number: 3  Author: reviewer  Subject: Note  Date: 2018-09-14 11:35:10**

mention also how EC values are than translated into dischage!

**Number: 4  Author: reviewer  Subject: Note  Date: 2018-09-14 11:35:52**

Mention limitations of the salt dilution method

**Number: 5  Author: reviewer  Subject: Note  Date: 2018-09-14 11:43:00**

It is more common to express the equation in terms of concentrations mot EC!

[Figure]

where V is the total volume of tracer introduced into the stream (m$^3$), k is the calibration constant in centimeters per microsiemens (cm µS$^{-1}$), n is the number of measurements taken during the breakthrough curve (unitless), EC(t) is the EC at time t (µS cm$^{-1}$), EC$_{BG}$ is the background EC (µS cm$^{-1}$), and $\Delta$t is the change in time between EC measurements (s).

We performed the following steps when making a salt dilution measurement:

1. Select
 stream reach with turbulence to facilitate vertical and horizontal mixing
2. Determined upstream point for introducing the salt solution and a downstream point for measuring EC

    a. A rule of thumb in the literature is to separate these locations roughly 25 stream widths apart (Day 1977; Butterworth et al. 2000; Moore 2005)

3. Estimated flow rate visually by estimated width, average depth, and average velocity
4. Prepared salt solution based on the following guidelines (adapted from Moore 2005)
    a. 10000 ml of stream water for every 1 m$^3$ s$^{-1}$ of estimated streamflow

    b. 1667 g of salt for every 1 m$^3$ s$^{-1}$ of estimated streamflow
    c. Thoroughly mix salt and water until all salt is dissolved
    d. Following these guidelines ensured a homogenous salt solution with 1 to 6 salt to water ratio by mass

5.  (Moore 2004b) to determine calibration constant (k) relating changes in EC values in micro Siemens per centimeter (µS cm$^{-1}$) in the stream to relative concentration of introduced salt solution (RC)

    a. Made diluted secondary solution by mixing 500 ml of stream water and 5 ml of salt solution
    b. Measured background stream water EC (EC$_{BG}$)
    c. Added known volume (typically 1 or 2 milliliters (ml)) of secondary solution to 500 ml of stream water in dilution cylinder
    d. Measured new dilution cylinder EC

    e. Repeated steps 5.c and 5.d until the full range of expected EC values were observed
    f. Calculated RC for each measurement point
    g. Plotted EC on the horizontal axis and RC on the vertical axis
    h. Performed linear regression
    i. Obtained k from the slope of the linear regression

6. Dumped salt solution at upstream location
7. Measured EC at downstream location during salinity breakthrough until EC returns to EC$_{BG}$
    a. Recorded a video of the EC meter screen at the downstream location and later digitized the values using the time from the video and the EC values from the meter
8. Solved for streamflow (Q) with Eq. (2)

Number: 1   Author: reviewer   Subject: Note   Date: 2018-09-14 11:43:54

I would use present tense for the method description

Number: 2   Author: reviewer   Subject: Replace   Date: 2018-09-14 11:49:16

Establish the calibration curve relating EC values to actual salt concentrations (Moore 2004b) ...

[Figure]

**2.1.3 Bernoulli run-up**
[Figure]

Similar to the float method, Bernoulli run-up (or Bernoulli) is based on the velocity-area principle. Total streamflow (Q; m$^3$ s$^{-1}$) was calculated with Eq. (3):

5    Eq. (3)   $Q = \sum_{i=1}^{n} VB_i * d1_i$

where $VB_i$ is velocity from Bernoulli run-up (m s$^{-1}$), $d1_i$ is depth (m), and $w_i$ is width (m) of each sub-section (i = 1 to n). Area for each sub-section is the product of the width and the depth in the middle of each sub-section. Velocity for each sub-section ($VB_i$) was determined by measuring the "run-up" or change in water level on a thin meter stick from when the stick

10  was inserted parallel and then perpendicular to the direction of flow. The basic principle is that "run-up" on a flat plate inserted perpendicular to flow is proportional to velocity based on the solution to Bernoulli's equation. Velocity ($VB_i$; m s$^{-1}$) was calculated from Bernoulli's principle with Eq. (4):

Eq. (4)   $VB_i = \sqrt{2g * (d2_i - d1_i)}$

where g is the gravitational constant (m s$^{-2}$) and $d2_i$ and $d1_i$ are the water depths (m) when the flat plate was perpendicular and parallel to the direction of flow, respectively.

Bernoulli run-up measurements involved the following steps:

1.  Selected constricted stream with elevated velocity to increase the difference between d1i and d2i
2.  Divided cross section into several sub-sections (n, typically between 5 and 20)
3.  For each section, measured and recorded
    a.  The depth with a flat plate held perpendicular to flow (d2i or the "Run-up" depth)
25      b.  The depth with a flat plate held parallel to flow (d1i or the actual water depth)
    c.  The width of the sub-section
4.  Solved for streamflow (Q) with Eq. (3) and Eq. (4)

**2.2**
[Figure]
 Reference flow

30  To evaluate the different citizen science flow measurement methods, a reference (or actual) flow for each site was needed. We used a SonTek FlowTracker acoustic Doppler velocimeter (ADV) to determine reference flows. The United States Geological Survey (USGS) mid-section method was used, following guidelines from USGS Water Supply Paper 2175

**Number: 1  Author: reviewer  Subject: Note  Date: 2018-09-14 12:23:15**

I am not familiar with this method and also the USBR (2001) includes two short paragraphs about this method. Do you use a standardized device for that? What size is the plat you are submerging into the water.

In any case the method is an alternative for estimating/measuring the flow velocity, the rest is the area-flow-velocity method

**Number: 2  Author: reviewer  Subject: Note  Date: 2018-09-14 12:13:06**

please give citation

**Number: 3  Author: reviewer  Subject: Note  Date: 2018-09-14 11:51:20**

why d1i and not just di?

**Number: 4  Author: reviewer  Subject: Note  Date: 2018-09-14 12:32:15**

First you introduce to the various methods used in the sudy! That's fine. Now I would suggest to explain the experimental setup, how you tested the applicability of the methods. (paragrahs below would fit here well)

IMPORTANT: Early in your method section (or as a separate section) describe what type of streams we are looking at in your study. Width, depths, closssection, steepnes, roughness, lamnar, turbulent, sediment transport, etc. This is important to assess whether these methods are appropriate to measure discharge or not!

**Number: 5  Author: reviewer  Subject: Note  Date: 2018-09-14 12:29:13**

I assume this device results ina surface flow velocity. What about your cross section? How detailed did you determine it?

[Figure]

(Rantz 1982), along with instrument specific recommendations from SonTek's FlowTracker manual (SonTek 2009). The FlowTracker ADV has a stated velocity measurement accuracy of within one percent (SonTek 2009). Flow measurement errors, calculated with an International Standards Organization (ISO) approach built into the FlowTracker software, are typically in the range of 3 to 10 %. Reference flow errors in this study are discuss in Section 4.5. A compilation of the
5    measurement reports generated by the FlowTracker ADV are included as supplementary material.

**2.3    Flow measurement method evaluation and analysis**

We first summarized flow measurement method evaluation results in map and tabular form (Fig. 1; Table 1). Measurement ID can be used to link data between the map and table. We used scatter plots to compare reference flow (x-axis) to the three
10    flow measurement methods evaluated (y-axis) to visualize and interpret results from each method. We fitted these points with a linear regression forced through the origin. To understand relative (normalized) errors, we calculated percent differences in relation to reference flow for each method. To better understand possible explanations for observed variability in our results, we performed a correlation analysis. For each method, we performed a Pearson's r correlation analysis (Lee Rodgers and Nicewander 1988) between the absolute value of percent difference in flow and (1) reference flow, (2) average
15    velocity, (3) $EC_{BG}$, and (4) slope. Slope values were developed using elevations from the Google Earth Digital Elevation Model (DEM) obtained
 along the centreline of the stream alignment both 100 meters upstream and downstream of each measurement point (retrieved July 2nd, 2018). While using DEM data for slope calculations is clearly inferior to performing topographic surveys in the field, this was not possible due to lack of equipment and time; therefore, these slope data are the best available numbers.

**2.4    Salt dilution calibration coefficient (k) analysis**

Arguably, one of the most complicated portions of a salt dilution measurement is performing the dilution test to determine the calibration coefficient k (Moore 2004b). To determine if the dilution test needs to be repeated for each citizen science measurement, we analyzed all k values determined during this study. In addition to the mean, range, and standard deviation,
25    we performed a Pearson's r correlation analysis (Lee Rodgers and Nicewander 1988) to see if k showed statistically significant trends with latitude, longitude, elevation, and $EC_{BG}$.

**2.5    2.5. CS Flow campaign - pilot testing of salt dilution method**

Based on the initial results from this study, S4W-Nepal developed an Open Data Kit (ODK; Anokwa et al. 2009) form for
30    citizen scientists to perform salt dilution measurements. The general workflow was (1) selected an appropriate measurement reach with good mixing and minimal bank storage, (2) performed a simplified float measurement (i.e. only a 3 or 4 depths

**Page:7**

[Figure]

Please mention the spatial resolution of the DEM. How many cells where considered in the 100 up and 100 downstream section to calculate slope?

Do you mean that this calibration is most difficult for citizens to do acuratly?

Why are yo then trying to correlate it with latitude and longitude, elevation and ECBG? Please explain better!

This is about the experimental setup and would fit better at an earlier spot in the method section.

What is S4W-Nepal! Please introduce earlier so the reader knows it! I assume iit is a certain project name!

You have describe the method beforehand. If you procedure is the same as applied by Anokwa et al., 2009 do not repeat it but refer to your description.

and velocities), (3) used the float flow estimate and $EC_{BG}$ to provide citizen scientists recommended salt/water dose, (4) used pre-weight packets of salt (e.g. 10 g, 20 g, 50 g, 100 g, etc.) to prepare tracer solution, (5) added tracer solution to stream and recorded video of EC breakthrough curve, (6) submitted form to ODK Aggregate server, (6) digitized breakthrough curve (i.e. time and EC) in shared Google Sheet salt dilution flow calculator.

During S4W-Nepal's Citizen Science Flow (CS Flow) campaign (15th to 21st of April 2018; Fig. 6), student volunteers from Khwopa College of Engineering were recruited, trained, divided into groups by sub-watershed, and sent to the field to perform salt dilution flow measurements. In the second week, student volunteers used a salt dilution Google Sheet flow calculator to digitize collected measurement data and compute flow (see supplementary material for Excel version).

10   Students analyzed data (third week) and finally presented oral and written summaries of their quality-controlled results. S4W-Nepal currently leverages the enthusiasm and schedule breaks in the academic calendar of young researchers to perform campaigns to improve our pre and post monsoon understanding of stone spouts (Nepali: dhunge dhara), land use, and now streamflow.

15   To analyze the generated streamflow data, we developed a scatter plot between flow estimates from the simplified float method (used to calculated salt dosage) and the salt dilution flow results. At locations were S4W-Nepal takes regular FlowTracker measurements, we compared the most recent S4W-Nepal observation(s) to CS Flow salt dilution measurements. Because salt dilution measurements were performed during the pre-monsoon period when precipitation is minimal, hydrographs are relatively steady with gradual recession over time as the South Asian Monsoon approaches.

20   Therefore, we did not expect differences in time (e.g. plus or minus one month roughly) between the two measurements to greatly impact the resulting comparisons.

**3    Results**

**3.1    Summary of evaluation measurements**

We performed sets of evaluation measurements at 20 sites within the Kathmandu Valley, Nepal (Fig. 1) at elevations ranging

25   from 1313 to 1905 meters above mean sea level. Flows evaluated ranged from 0.006 to 0.240 $m^3$ $s^{-1}$ (Table 1; sorted in ascending order by reference flow). Percent differences averaged 7.9, 8.2, and 25.7 % and standard deviations (std dev) of 29.1, 17.2, and 61.9 % were observed for float, salt dilution, and Bernoulli methods, respectively (Table 1). Field notes from Bernoulli flow measurements for two measurements (Msmt IDs 17041903 and 17031102) were destroyed by water damage, so flow and percent difference data were not available. Plots of EC and change in EC as a function of time for all 20 salt

30   dilution measurements are shown in Fig. 2. Additional data for evaluation measurements is included as supplementary material.

**Number: 1  Author: reviewer  Subject: Note  Date: 2018-09-14 13:22:14**

You need to indroduce to these watershed. How many, were, how big what average flow during the campaign etc. Refer to Fig. 1!!

**Number: 2  Author: reviewer  Subject: Note  Date: 2018-09-14 12:46:05**

how long were they trained, was it a theoretical explanation or a hands-on training actually applying the methods themselves under supervision?

**Number: 3  Author: reviewer  Subject: Note  Date: 2018-09-14 12:47:06**

Start with this sentence!

**Number: 4  Author: reviewer  Subject: Note  Date: 2018-09-14 12:48:32**

Explain more: Where are these stations. Are these the same as the students performed the salt dilution tests. Mark these stations in a map!

**Number: 5  Author: reviewer  Subject: Note  Date: 2018-09-14 15:54:23**

One month (!!) is really long! You need to prove your assumption with statistical data. E.g. mean daily runoff at the gauging stations for each month in the pre-monsoon period and a statistical test, showing that the mean daily discharge is not statistically significantly different!

If you do not have the data to prove this a) consider to skip this part or b) take a dataset of a different gauging station of the region (e.g. operated by a local authority or so)  and prove it based on this data.

**Number: 6  Author: reviewer  Subject: Insert Text  Date: 2018-09-14 13:25:11**

... a xx km2 Catchment xx km NW of yyy-city!

**Number: 7  Author: reviewer  Subject: Insert Text  Date: 2018-09-14 13:26:02**

Reference flow ...

**Number: 8  Author: reviewer  Subject: Note  Date: 2018-09-14 13:29:54**

percent difference to what? I think you men mean percent difference between the float method, the salt dilution method and the Bernoulli method relative the the doppler measurements?

**Number: 9  Author: reviewer  Subject: Note  Date: 2018-09-14 13:30:38**

exclude them form you data set and from the map

[Figure]

[Figure]

Figure 1: Map figure showing the topography (green to tan to white color gradation) of the Kathmandu Valley from a Shuttle Research Telemetry Mission (SRTM 2000) Digital Elevation Model (DEM), resulting stream network (Davids et al. 2018), and locations of flow measurements (msmts). Measurement points are labelled with measurement ID (msmt_id). Names of the ten historically perennial tributaries are shown.

**Number: 1  Author: reviewer  Subject: Note  Date: 2018-09-14 13:31:46**

first order streams ?

**Number: 2  Author: reviewer  Subject: Note  Date: 2018-09-14 13:37:11**

the labels partly cover the points.

Even if you (internally used these long digit labels, consider to re-lable them by 1 to 20.

**Number: 3  Author: reviewer  Subject: Note  Date: 2018-09-14 13:33:16**

do not split the legend for DEM
do not use uneven classbreaks for you r colours!
Msmts is explained in the text but not intuitive.

[Figure]

**Table 1: Tabular summary of measurement comparison data. Records sorted in ascending order by reference flow (Q Reference). Latitude and longitude in reference to the WGS84 datum. All flow values shown are shown in m$^3$ s$^{-1}$ rounded to the thousandth place. Percent differences calculated using Q Reference (FlowTracker) as the actual flow. Data summarized at the bottom with average, minimum (min), maximum (max), and standard deviation (std dev). Note that measurement ID (Msmt ID) is comprised of two digits for year, month, date, and measurement number starting at 01 each day.**

| Msmt ID | Latitude | Longitude | Elev- ation (m) | Q Reference (m3 s-1) | Q Float (m3 s-1) | Q Salt (m3 s-1) | Q Bernoulli (m3 s-1) | % Diff- erence Float | % Diff- erence Salt | % Diff- erence Bernoulli |
|---|---|---|---|---|---|---|---|---|---|---|
| 17030202 | 27.78065 | 85.42426 | 1649 | 0.006 | 0.007 | 0.006 | 0.009 | 15.6 | -12.5 | 37.5 |
| 17041802 | 27.78158 | 85.42385 | 1659 | 0.007 | 0.008 | 0.007 | 0.010 | 15.9 | 7.2 | 44.9 |
| 17031001 | 27.79649 | 85.42177 | 1905 | 0.011 | 0.008 | 0.012 | 0.009 | -28.4 | 11 | -19.3 |
| 17042401 | 27.70026 | 85.22077 | 1406 | 0.017 | 0.019 | 0.019 | 0.018 | 11.2 | 11.2 | 4.7 |
| 17032201 | 27.57487 | 85.31314 | 1482 | 0.018 | 0.020 | 0.024 | 0.019 | 12.4 | 37.3 | 5.1 |
| 17041901 | 27.77164 | 85.42657 | 1609 | 0.019 | 0.028 | 0.027 | 0.022 | 48.4 | 47.3 | 16.7 |
| 17033001 | 27.78691 | 85.32589 | 1364 | 0.021 | 0.026 | 0.030 | 0.048 | 27.2 | 43.2 | 132 |
| 17042402 | 27.69620 | 85.23142 | 1382 | 0.023 | 0.010 | 0.025 | 0.006 | -59.2 | 5.2 | -73 |
| 17041903 | 27.75406 | 85.42170 | 1355 | 0.034 | 0.051 | 0.033 |  | 51.6 | -0.9 |  |
| 17041902 | 27.77154 | 85.42680 | 1609 | 0.041 | 0.041 | 0.047 | 0.063 | 0.5 | 14.6 | 53.2 |
| 17030101 | 27.78483 | 85.44480 | 1877 | 0.104 | 0.111 | 0.088 | 0.102 | 6.9 | -15.9 | -2.6 |
| 17032203 | 27.57542 | 85.31268 | 1477 | 0.111 | 0.106 | 0.120 | 0.116 | -4.3 | 8 | 5.1 |
| 17032202 | 27.57410 | 85.31277 | 1481 | 0.117 | 0.081 | 0.126 | 0.102 | -30.7 | 8 | -13.2 |
| 17033002 | 27.78627 | 85.32583 | 1356 | 0.153 | 0.208 | 0.144 | 0.470 | 36.5 | -5.6 | 207.9 |
| 17030201 | 27.78156 | 85.42383 | 1659 | 0.155 | 0.248 | 0.176 | 0.161 | 59.3 | 13.1 | 3.5 |
| 17041803 | 27.78168 | 85.42373 | 1663 | 0.156 | 0.140 | 0.142 | 0.210 | -10.4 | -8.9 | 34.4 |
| 17031002 | 27.77932 | 85.42496 | 1653 | 0.159 | 0.183 | 0.155 | 0.228 | 15.2 | -2.8 | 43.4 |
| 17031101 | 27.78505 | 85.44473 | 1877 | 0.208 | 0.221 | 0.207 | 0.150 | 6.5 | -0.6 | -27.8 |
| 17031102 | 27.77514 | 85.43867 | 1806 | 0.230 | 0.188 | 0.219 |  | -18 | -4.8 |  |
| 17042002 | 27.71106 | 85.35432 | 1313 | 0.240 | 0.246 | 0.264 | 0.264 | 2.7 | 10.2 | 10.1 |
|  |  | average -> | 1579 | 0.092 | 0.098 | 0.094 | 0.112 | 7.9 | 8.2 | 25.7 |
|  |  | min -> | 1313 | 0.006 | 0.007 | 0.006 | 0.006 | -59.2 | -15.9 | -73.0 |
|  |  | max -> | 1905 | 0.240 | 0.248 | 0.264 | 0.470 | 59.3 | 47.3 | 207.9 |
|  |  | std dev -> | 190 | 0.081 | 0.089 | 0.081 | 0.122 | 29.1 | 17.2 | 61.9 |

Number: 1  Author: reviewer  Subject: Note  Date: 2018-09-14 13:42:16

Empty cells indicate lost data records

Number: 2  Author: reviewer  Subject: Note  Date: 2018-09-14 13:40:12

... but 01, 02 03 are not replicates at the same location! Make that clear!

Consider to re-label the msmts from 1 to 20 and give the date in a separate colum.

[Figure]

[Figure]

**Figure 2: Plots of EC (µS cm⁻¹; blue trace) and change in EC (µS cm⁻¹; green trace) as a function of time (s) for the 20 salt dilution evaluation measurements. Measurement ID (Msmt ID; Table 1) shown at the top right of each subplot (i.e. a through t).**
[Figure]

**3.2 Flow and calibration coefficient (k) results**

**3.2.1 Flow scatter plots**

Scatter plots between reference and observed flows with linear regressions forced through the origin had slopes of 1.05, 1.01, and 1.26 for float, salt dilution, and Bernoulli methods, respectively (Fig. 3). A slope of one represents zero systematic bias, whereas values over one represent positive bias, and values less than one represent negative bias. Therefore, for all the methods evaluated we observed different degrees of positive bias. $R$-squared values were 0.90, 0.98, and 0.61 for float, salt dilution, and Bernoulli methods, respectively (Fig. 3). $R$-squared values represent the goodness of fit between the regression and the observed data; values closer to one represent a better fit. This can also be seen by the observations for salt dilution plotting closest to the regression line, whereas float and Bernoulli points in general plot farther away from the regression line.

Number: 1  Author: reviewer  Subject: Note  Date: 2018-09-14 13:46:16

What was your intention to show this rather big Figure?

Number: 2  Author: reviewer  Subject: Note  Date: 2018-09-14 13:49:30

better say: "an over- or under estimation of the estimated discharge relative to the reference discharge.

Number: 3  Author: reviewer  Subject: Replace  Date: 2018-09-14 13:48:26

>1

Number: 4  Author: reviewer  Subject: Note  Date: 2018-09-14 13:52:51

Bias would be systematic behaviour of all datapoints shifted up or down. Your regression line indicates a very slight overestimation of the estimated discharge to the reference.

[Figure]

[Figure]

**Figure 3: Scatter plots between reference flow and observed flow for (a) float, (b) salt dilution, and (c) Bernoulli. Note there is one Bernoulli measurement point (17033002) that is outside of the plot space shown (fixed from 0.0 to 0.3 for consistency). Linear regressions and r-squared values shown on the bottom right of each sub-plot.**

**3.2.2 Flow error correlations**

We found statistically significant correlations (n = 20, p = 0.1 to 0.378) between the absolute value of percent error for float and average velocity (Avg Vel; sub-plot 2; r = -0.48) and salt dilution percent error and reference flow (Q Ref; sub-plot 5; r = -0. (Fig. 4). In both cases, the correlation coefficient was negative, indicating an inverse relationship between the variables. No statistically significant correlations were observed between the remaining pairs of variables.

**Number: 1  Author: reviewer  Subject: Note  Date: 2018-09-14 14:44:28**

Good you tested the significance! But I suggest to also mention that the functional relation between the valiabels is not clear (not linear, not exponetial).

**Number: 2  Author: reviewer  Subject: Note  Date: 2018-09-14 13:55:17**

Typically alpah would be chosen to 0.05. Please argue in the text why you use  0.1

**Number: 3  Author: reviewer  Subject: Note  Date: 2018-09-14 14:46:21**

Attention: Use the Spearman Rank correlation and approbriate non-parametric statistical test for your type of data. The assumptions for Pearson Correlation are not fulfilled

**Number: 4  Author: reviewer  Subject: Note  Date: 2018-09-14 13:55:49**

also give p value

**Number: 5  Author: reviewer  Subject: Note  Date: 2018-09-14 13:55:55**

also give p value

[Figure]

[Figure]

[Figure]

**Figure 4: Scatter plots between reference flow (Q Ref; m³ s⁻¹), average water velocity (Avg Vel; m s⁻¹), slope, and background EC (EC BG; μS cm⁻¹) and absolute value (Abs) of percent errors for float, salt dilution, and Bernoulli. Pearson r values shown on the upper right of each subplot (i.e. a through l).**

**3.2.3    Salt dilution calibration coefficient (k) results**

The mean calibration coefficient (k) from measurements performed in the field was $2.81 \times 10^{-6} \pm 2.66 \times 10^{-7}$ (95 % confidence interval; n = 10, min = $2.57 \times 10^{-6}$, max = $3.05 \times 10^{-6}$, std dev = $1.33 \times 10^{-7}$). We used mean k to compute salt dilution flows for the remaining 10 measurements. We found statistically significant correlations (n = 10, p = 0.1, r ≥ 0.549)

10    between the calibration coefficient (k) and Longitude (r = 0.60) and Elevation (r = 0.61; Fig. 5). In both cases, the correlation coefficient was positive, indicating a direct relationship between the variables. No statistically significant correlations were observed between the remaining pairs of variables.

**Number: 1   Author: reviewer   Subject: Note   Date: 2018-09-14 13:56:39**

please also show p values for the relations

**Number: 2   Author: reviewer   Subject: Note   Date: 2018-09-14 14:42:41**

For this type of dataset you need to used the Spearman Rank correlation coefficient and the associated non-parametric statistical test!

**Number: 3   Author: reviewer   Subject: Insert Text   Date: 2018-09-14 16:41:11**

th the reference measurements taken by the doppler radar.

**Number: 4   Author: reviewer   Subject: Note   Date: 2018-09-14 14:51:15**

what are "remaining measuremetns"

Stricktly k is site and time specific and needs to be determined for each site and at least once a day! k is highly sensitive for the outcome of the discharge. Using an average value is not advisable!

If you were forced to do so, clearly state the reasons in the text and indicate the affected values. Also discuss the error you introduced by this procedure.

**Number: 5   Author: reviewer   Subject: Note   Date: 2018-09-14 14:51:37**

see my comment on alpha above

**Number: 6   Author: reviewer   Subject: Note   Date: 2018-09-14 14:55:05**

please better describe and argeu why k should be a function of latitude and longitude or elevation. I do not understand the physical based rational. Make clear why you think this is no spurious correlation.

[Figure]

[Figure]

**Figure 5: Scatter plots between (a) Latitude in degrees, (b) Longitude in degrees, (c) Elevation in meters above mean sea level (m), and (d) background EC in μS cm⁻¹ and the salt dilution calibration coefficient (k).   Pearson's r values shown on the upper right of each sub-plot.**

**3.3    3.3. CS Flow campaign results**

From the 15th to the 21st of April 2018, 20 students from Khwopa College of Engineering in Bhaktapur, Nepal joined S4W-Nepal's CS Flow campaign.   After four hours of training (i.e. two hours classroom and two hours in the field), the student volunteers performed 14 salt dilution streamflow measurements in the 10 sub-watersheds of the Kathmandu Valley (Fig. 6).

10    Observed flows ranged from 0.0004 to 0.425 m³ s⁻¹ (a summary of the measurement data is included as supplementary material).

**Number: 1  Author: reviewer  Subject: Note  Date: 2018-09-14 14:57:56**

This is good information! Copy it to the method section where you describe the experimental setup!

**Number: 2  Author: reviewer  Subject: Note  Date: 2018-09-14 15:00:00**

It is nit fully crear: The data you presented in Tabl. 1 are done by the authors?

[Figure]

**Figure 6: CS Flow Campaign measurement locations (n = 145) within the Kathmandu Valley. Circular symbol colors are graduated by observed flow rate, categorized by quartile. Larger flows (i.e. darker symbols) were observed on the mainstems (i.e. wider blue lines) of the 10 tributaries of mati River in the Kathmandu Valley.**

Scatter plots between flow estimates from the simplified float method (used to calculated salt dosage) and the salt dilution flow results show that systematic differences increase as the observed flow rate decreases (Fig. 7).

**Number: 1 Author: reviewer Subject: Note Date: 2018-09-14 15:01:53**

consider using even class breaks to GIS pre-determined ones.

**Number: 2 Author: reviewer Subject: Note Date: 2018-09-14 15:02:18**

Good that you state the categorization method!

**Number: 3 Author: reviewer Subject: Note Date: 2018-09-14 15:39:51**

to better distinguish between "observed flow" measured with the doppler radar (dataset described before) I would call this here differently. You compare two different estimation methods with each other.

[Figure]

Figure 7: Scatter plots between salt dilution measurements on the horizontal axis and simplified float estimates on the vertical axis. All data are shown on subplot (a), flows below 0.05 m³ s⁻¹ are included on subplot (b), and flows below 0.01 m³ s⁻¹ are shown on subplot (c). A linear fit with corresponding r-squared values are shown. Note that the vertical axis scales for subplots (b) and (c) are fixed at three times the horizontal axis scale to ensure that all data are visible.

We identified five locations where S4W-Nepal had performed FlowTracker measurements that could be used as reference flows within roughly one month (plus or minus) of the CS Flow salt dilution measurements (Table 2). Comparable flows ranged from 0.012 and 0.111 m³ s⁻¹. The average error between CS Flow salt dilution and S4W-Nepal FlowTracker measurements was -6.3 %, with a standard deviation of 11.5 %. Linear regression forced through the origin between reference flows and CS Flow measurements had a slope of 0.90 with an r-squared value of 0.97.

**Number: 1 Author: reviewer Subject: Note Date: 2018-09-14 16:18:18**

here you call it simplified float flow but in the method you call it float flow. Please be consistent.

**Number: 2 Author: reviewer Subject: Note Date: 2018-09-14 15:34:47**

please include p-value for all three plots

**Number: 3 Author: reviewer Subject: Note Date: 2018-09-14 15:35:25**

what is meant with "fixed at three times"?

**Number: 4 Author: reviewer Subject: Note Date: 2018-09-14 15:45:28**

see my comment in the method section on the validity of this comparison with data +/- one months before/after the CS-campain.

**Table 2:** *Comparison between CS Flow salt dilution and S4W-Nepal FlowTracker measurements. Five measurements were identified for evaluation. In one case (i.e. CS Flow Msmt Date 4/16/2018 10:03), a linear interpolation between two S4W-Nepal measurements (i.e. 3/15/2018 7:15 and 5/23/2018 14:23) was made because measurements for both March and May were available. Data summarized at the bottom with average, minimum (min), maximum (max), and standard deviation (std dev).*

[Figure]

[Figure]

| S4W-Nepal SiteID | CS Flow Msmt Date | S4W-Nepal Msmt Date | CS Flow Salt Dilution Q (m3 s-1) | S4W-Nepal FlowTracker Reference Q (m3 s-1) | % Difference |
|---|---|---|---|---|---|
| DB02 | 4/16/2018 10:03 | 3/15/2018 7:15 and 5/23/2018 14:23 | 0.0 | 0.0492 | 7.5% |
| BM02 | 4/16/2018 13:22 | 3/15/2018 9:49 | 0.0169 | 0.0172 | -1.7% |
| NA02 | 4/18/2018 13:28 | 3/31/2018 4:39 | 0.0940 | 0.1110 | -15.3% |
| BA01 | 4/20/2018 11:30 | 3/30/2018 9:24 | 0.0090 | 0.0118 | -23.7% |
| NK03 | 4/18/2018 13:57 | 5/16/2018 12:22 | 0.0461 | 0.0454 | 1.5% |
| | | average -> | 0.0438 | 0.0469 | -6.3% |
| | | min -> | 0.0090 | 0.0118 | -23.7% |
| | | max -> | 0.0940 | 0.1110 | 7.5% |
| | | std dev -> | 0.0301 | 0.0353 | 11.5% |

[Figure]

[Figure]

**4    Discussion**

**4.1    Preferred measurement method**

Based on 20 flow measurements performed in this study, we concluded that the salt dilution method will (1) provide the
10   most accurate streamflow data (at least for the range of flows observed), and (2) will be the easiest method for citizen scientists to repeat in the field with limited amounts of training and equipment (see Section 4.4 on citizen scientist repeatability).

While all flow measurement methods evaluated had positive bias, salt dilution showed the closest agreement to the
15   reference flow with an average over estimation of only one percent (based on the linear regression), followed by float and Bernoulli at 5 and 25 %, respectively (Fig. 3).  The standard deviation for errors was 17 % for salt dilution, and 29 and 62 % for float and Bernoulli, respectively.  Additionally, r-squared values indicated that salt dilution had the least amount of variance from the trend line (i.e. closest to one).

20   Only three salt dilution measurements (i.e. 17032201, 17041901, and 17033001) had percent differences larger than 20 %, and these were all positively biased in relatively small streams (flows between 0.018 and 0.021 m$^3$ s$^{-1}$).  While we can't be certain, we suggest that these errors may be due hyporheic exchanges that removed some salt solution from the measurement

**Number: 1  Author: reviewer  Subject: Note  Date: 2018-09-14 15:47:58**

You need to give prove that this is a valid procedure!

**Number: 2  Author: reviewer  Subject: Note  Date: 2018-09-14 16:19:43**

Is this valeu the average of both measurements? Better sate both values individually to be consistent with all other values.

**Number: 3  Author: reviewer  Subject: Note  Date: 2018-09-14 15:49:19**

Please indicate these sites in the maps!

**Number: 4  Author: reviewer  Subject: Note  Date: 2018-09-14 15:51:38**

Why not show the agreement between the salt dilution and the Bernoulli method. If the Bernoulli method has not been part of the CS-campain you need to better describe this in the method section. Right now  one would assume all three methods were used in the SC-campain.

**Number: 5  Author: reviewer  Subject: Note  Date: 2018-09-14 16:29:51**

Why not also discuss the other methods and give a suggestion at the end?

**Number: 6  Author: reviewer  Subject: Note  Date: 2018-09-14 16:20:38**

see my comment on "bias" above

**Number: 7  Author: reviewer  Subject: Note  Date: 2018-09-14 16:21:47**

This reads more like results. Consider to shift this information to the result section.

**Number: 8  Author: reviewer  Subject: Note  Date: 2018-09-14 16:24:16**

Give the standard deviation, coefficient of variation or variance to say something about variance.

[Figure]

reach before lateral and vertical mixing could fully occur. In other words, it is possible that some of the salt solution became "underflow" shortly after the injection point and did not return to the surface stream prior to the EC measurement location. As observed, this "removal" of salt solution would lead to a systematic overestimation of flow. If these three measurements are removed, the mean and standard deviation for salt dilution method percent differences become 2 and 9 %, respectively.

5   These percent differences fall within the expected range of uncertainty presented in the literature for salt dilution gauging (Day 1976; USBR 2001; Moore 2004a; Herschy 2014). Excluding these three errors, and assuming errors are normally distributed, we expect that salt dilution measurements will be within roughly ±18 % (95 % confidence interval).

**4.2   Measurement error correlation**
[Figure]

10  In Fig. 4, three "outlier" percent errors for salt dilution measurements were seen in the middle row of sub-plots (5 through 8) as clusters of three points towards the top of each sub-plot. After removing these points, the Pearson's r value decreased to 0.26, and the correlation became statistically insignificant (n = 20, p = 0.1, r > 0.378). Therefore, we are cautious to conclude that error in salt dilution measurements decreases as the amount of streamflow increases. Errors in salt dilution measurements appeared to be uncorrelated with the other variables evaluated (e.g. average velocity, slope, and $EC_{BG}$).

The other observed statistically significant correlation was an inverse relationship between average velocity and error in float measurements. Our experience in the field validated that slow moving (and shallow) float velocity measurements were difficult to perform. The combination of turbulence and boundary layer impacts from the streambed and the overlying air mass often made floating objects on the surface travel in non-linear paths, adding uncertainty to distance and time

20  measurements. Challenges with applying the float method in shallow depths was supported USBR (2001) and Escurra (2004), who showed that uncertainty in surface velocity coefficients (i.e. the ratio of surface velocity to actual mean velocity of the underlying water column) increased as depth decreased, especially below 0.3 m.

**4.3   Salt dilution calibration coefficient (k)**

25  Moore (2005) suggests that k depends on (1) the ratio of salt and water in the tracer solution and (2) the chemical composition of the stream water. To minimize variability in k due to changes in salt concentration, a fixed ratio of salt to water (e.g. 1 to 6 by mass) should be consistently used to prepare tracer solutions (as it was during this investigation). Significant correlations observed between k and longitude and elevation may be due to changes in water chemistry that co-vary with these independent variables. Measurements performed in the northeastern portion (i.e. higher longitude) of the

30  Kathmandu Valley were higher in altitude. Geology in the north of the Kathmandu Valley is a mixture of weathered igneous and metamorphic parent material (e.g. gneiss, phyllite, schist, etc.). Geology surrounding measurements in the southwest of the Kathmandu Valley is dominated by sedimentary and slightly metamorphosed deposits of sand, silt, and clay (Shrestha et

**Number: 1  Author: reviewer  Subject: Note  Date: 2018-09-14 16:38:34**

This section should also mention potential differences in errors when the three different methods are applied by you as experts and by citizens or at least refer to the section you have on CS.

**Number: 2  Author: reviewer  Subject: Note  Date: 2018-09-14 16:44:58**

Please also mention the error associated with the Doppler radar as a function of discharge (information provided by the selling company) Also the Dopper radar has it's error!

**Number: 3  Author: reviewer  Subject: Note  Date: 2018-09-14 16:32:43**

please use the a), b), c), ...to refer to the panels you take about to be clear.

**Number: 4  Author: reviewer  Subject: Insert Text  Date: 2018-09-14 16:42:43**

as a function of discharge

**Number: 5  Author: reviewer  Subject: Note  Date: 2018-09-14 16:33:55**

I think this is interesting to mention in the result section! As it changes the outcome of your analysis!

**Number: 6  Author: reviewer  Subject: Note  Date: 2018-09-14 16:47:36**

Not clear what you are referring to. also Fig 4? But the it is the Absolute percent error of the salt dilution measurement and the doppler radar-based dischage. Please also sy Fig 4 xy.

**Number: 7  Author: reviewer  Subject: Note  Date: 2018-09-14 16:36:45**

Did you do a float velocity test  only once? Typically one can do several estimates and average the values to compensate for the effects you mention. Maybe mention waht you did in detail in the method section.

**Number: 8  Author: reviewer  Subject: Note  Date: 2018-09-14 16:50:08**

In the first sentence you state the two factors influencing k. Lat and Long are most likely not relevant and meaningful influencing factors, just spurious correlations.

[Figure]

al. 2012). These differences in geology could impact water chemistry through water-rock interactions (Lasaga 1984) and ultimately impact k. Additional work should focus on improving our understanding of the variables affecting k.
[Figure]
 Specifically, spatial variability in k due to changes in stream water chemistry should be investigated prior to applying the salt dilution methodology described in this paper into other areas.

**4.4 Citizen scientist repeatability**

In this context, repeatability refers to the overall likelihood that the measurement method can be successfully repeated by citizen scientists. Along these lines, there were several practical observations in the field worth briefly discussing. While difficult to quantitatively evaluate, we offer the following qualitative observations regarding the selection of a preferred

10 citizen science streamflow measurement method.

- **Required training** - Float and salt dilution require similar amount of training, which from our experience we estimate to be roughly four hours, involving both classroom and field time. The amount of training is strongly dependent on the background of the volunteers. Bernoulli requires additional training for how to minimize vertical

15 movement of the metal measurement plate.

- **Cost of equipment** - All methods require a SmartPhone, measuring scale, and measuring tape. Additionally, salt dilution requires an inexpensive EC meter (e.g. $15 HoneForest Water Quality Tester), a graduated cylinder, and a bucket.

- **Number of citizen scientists required** - Teams of at least two citizen scientists are recommended for all methods;

20 teams of three were preferred in our experience.
[Figure]

- **Data recording requirements** - For float and Bernoulli, depth, width, and velocity (including distance and time) data needs to be recorded at multiple locations. Salt dilution only requires some basic data entry and a video of the breakthrough curve.

- **Complexity of procedure** - Float and Bernoulli require detailed transects of the stream. Bernoulli is extremely

25 sensitive to vertical movements in the metal measurement plate. Bernoulli is always very difficult for low velocities.

- **Enjoyability of measurement** - We found that citizen scientists generally enjoyed watching the salt dilution breakthrough curves and found them less repetitive than the tasks associated with float and Bernoulli methods. Bernoulli measurements can be frustrating when trying to keep the metal measurement plate from moving

30 vertically, especially when there is a soft streambed.

- **Safety** - Both float and Bernoulli measurements require citizen scientists to wade through the stream. At certain flow rates this clearly poses a safety risk, especially for people who cannot swim. A clear benefit of the salt dilution

**Number: 1  Author: reviewer  Subject: Note  Date: 2018-09-14 16:52:30**

Different geology effects the background concentration, for sure, but should not effect k.
Please reconsider the content of this paragraph.

**Number: 2  Author: reviewer  Subject: Note  Date: 2018-09-14 17:09:13**

I think this section is useful for the reader.
For you chosen title I would encourage you to check your data and see if you can use it for additional research questions concerning the Citizen Science aspect.
E.g. the quality of the measurements could be a function of the person/or group that performed the calibration. Comparing your dataset (n=20) wiht the CS dataset (n=145) you could maby see if experts do "better" than "citizes" or not.

**Number: 3  Author: reviewer  Subject: Note  Date: 2018-09-14 17:01:26**

This is an important information you need to mention in the method section. You actually did not have 20 students but 7 teams of 3 persons?

method is that everything but the simplified float estimate can be performed from the stream bank. Note that the results from the simplified float method are only used to determine the salt dosing. If the salt dosing doesn't provide a large enough change in EC to be clearly observed, the measurement can be repeated with a higher flow estimate and corresponding increase in tracer solution.

**4.5    Uncertainty in reference flows**

[Figure]

Uncertainty in measurements of reference flow (i.e. actual flow) affected uncertainty in our evaluation of the three flow measurement methods. Based on an ISO discharge uncertainty calculation within the SonTek FlowTracker software, the uncertainties in flow ranged from 2.5 to 6.0 %, with a mean of 3.7 %. Based on the literature (Rantz 1982; Harmel 2006;
10   Herschy 2014), these uncertainties in reference flows are towards the lower end of the expected range for field measurements of streamflow. Therefore, we do not think that any systematic biases or uncertainties in our data change the results of this paper.

**4.6    Reynolds number**

[Figure]

15   Turbulent mixing of flow is an important aspect of salt dilution flow measurements. Reynolds number is typically used as a quantitative measure of turbulence in fluid flow. In addition to density and viscosity, fluid velocity and a characteristic length are required for calculating Reynolds number. Many of our measurements were performed in mountainous headwater streams with high slopes. To collect the most accurate reference flow measurements, however, we selected the lowest
 gradient stretch of the stream. The cross sections used for reference flow measurements were typically in the widest and
20   deepest reaches of the stream to ensure the most laminar flow lines for accurate velocity and area measurements. Using velocity and characteristic length data from these reference flow measurement locations to calculate Reynolds number would not have been representative of the actual average Reynolds number of the stream reaches used for the salt dilution measurements. This is because selected stream reaches for salt dilution included steep gradients with lots of mixing, which were often either entirely upstream or downstream of reference flow locations. Therefore, we did not include Reynolds
25   number in the correlation analysis for salt dilution.

**4.7    Flow measurement methods not evaluated**
[Figure]

We initially considered including the slope area method (USBR 2001) based on the Manning's equation for evaluation. The concept was to use a long clear flexible tube of a known length (e.g. 20 m) to measure the slope of the water surface using
30   the principle of a water level. The tube was completely submerged and filled with water. The upstream end of the tube remained submerged and the entrance was held perpendicular to flow to ensure that only the pressure head of the stream was

**Number: 1  Author: reviewer  Subject: Note  Date: 2018-09-14 17:34:13**

Ah, here you mention the acuracy of the doppler radar device! Good!

I assume the errror is depending on the flow velocity and has a lower and upper rage of proper application given by the producer. Pease mention this here.

**Number: 2  Author: reviewer  Subject: Note  Date: 2018-09-14 17:45:13**

maybe union with the section before

**Number: 3  Author: reviewer  Subject: Note  Date: 2018-09-14 17:35:45**

give range and mean slope you used in brackets

Where this step and pool river bads?

**Number: 4  Author: reviewer  Subject: Note  Date: 2018-09-14 17:45:50**

move to the other methods that I think would all better fit in the introduction section.

[Figure]

sensed at the tube inlet. The tube was stretched out longitudinally along the stream reach, and the downstream end of the tube was exposed to the atmosphere. The difference in water levels inside the downstream end of the tube and the stream water level immediately outside of the tube
 measured. This change in head was divided by the total length of the tube to determine the slope of the water surface.

Within the first few days of field work, we concluded that this method was not suitable for piloting at the types of sites we were investigating. Because we were particularly interested in high gradient headwater streams, the primary problem was finding a stretch of stream long enough that was flowing at normal depth, without backwater and drops in the water surface caused by sudden changes in channel geometry (both longitudinally and latitudinally). An additional challenge of this

10 method is that uncertainty in flow measurements are linearly proportional to uncertainty in estimations of the roughness coefficient (n), and n is difficult to estimate visually, especially for citizen scientists. Therefore, Manning's method was not included in this investigation. Despite our experience, we suggest that in certain settings with long straight reaches of uniform flow, Manning's may still be an appropriate citizen science flow measurement method. To simplify observations of the change in head, the Manning's method would also benefit from the installation of upstream and downstream staff gauges

15 survey to a common datum. However, this would make the method difficult and costly to implement, and therefore less scalable.

**4.8   CS Flow campaign**

The CS Flow Campaign provided us with a unique opportunity to evaluate the preferred salt dilution citizen science

20 streamflow measurement method at a larger scale. In addition to the valuable streamflow data that will help us characterize the hydrological situation in the Kathmandu Valley with greater precision we also learned many practical lessons about how to apply citizen science based streamflow generation methods at a broader scale. Unfortunately, there was no systematic way to evaluate the accuracy of all the measurements performed. However, at the five locations were S4W-Nepal FlowTracker measurements were available, the resulting errors ($\mu$ = -6.7 %, std dev = 11.5 %; Table 2) where comparable to

25 our initial evaluation data ($\mu$ = 8.2 %, std dev = 17.2 %; Table 1).

Linear regression forced through the origin between reference flows and salt dilution measurements had slopes of 0.90 and 1.01 with an r-squared values of 0.97 and 0.98 for CS Flow measurements and our initial evaluation measurements, respectively. Goodness of fit was similar, but while evaluation measurements had a slight positive bias (1 %), CS Flow

30 measurements had a larger negative bias (-10 %). One possible explanation for this is that the three CS Flow comparisons that had negative percent differences (i.e. BM02, NA02, and BA01; Table 2) all used reference flows performed prior to CS Flow measurements. Since hydrographs during this season are gradually receding prior to the onset of the South Asian Monsoon, it is possible that the actual flow of the streams decreased between reference flow and CS Flow observations.

**Number: 1  Author: reviewer  Subject: Note  Date: 2018-09-14 17:40:27**

I like your method! Still the section is too long for a method you finally decided to be not practical and did not use in this paper! You could shorten the text and should mention the Manning-Strickler Method as alternative (that you finally not chose) in the Introduction toughest with all other methods.

**Number: 2  Author: reviewer  Subject: Note  Date: 2018-09-14 17:47:08**

Ah, you need to stat in the method section that the CS campaign (n + 145) only was on applying salt dilution not all other methods.

**Number: 3  Author: reviewer  Subject: Note  Date: 2018-09-14 18:01:28**

You dataset is good and includes a lot of potential. However the campaign needed to be repeated to learn more about the hydrology of the area. You could discuss if this is feasible or not.

**Number: 4  Author: reviewer  Subject: Note  Date: 2018-09-14 18:09:54**

see my concern about the reference measurements +/- 1 month befoer/after the CS campain.

**Number: 5  Author: reviewer  Subject: Note  Date: 2018-09-14 18:11:11**

If this is the case than you definitely cannot take the dropper radar based discharge as reference for you CS campain!

[Figure]

As flows decreased, we observed a progressively increasing positive bias between simplified float estimates and salt dilution measurements (Fig. 7). This finding is congruent with previous efforts to characterize the dynamic relationship between channel depth and surface velocity coefficients (USBR 2001; Escurra 2004). Average stream depths were often on the order

5 of a few c
eters for the headwater catchments observed. Surface velocity coefficients provided by USBR (2001) range from 0.66 to 0.80 with increasing depths, but are held constant at 0.66 for depths less than 0.3 meters. Our results indicate that for flows less than 0.01 $m^3$ $s^{-1}$ a surface velocity coefficient of 0.5 would be more appropriate. The strength of the relationship between salt dilution and float streamflows also deteriorates as flows decrease (i.e. r-squared equals 0.83, 0.69, and 0.50 for plots 1, 2, and 3, respectively). This suggests that surface velocity coefficients are highly variable at low flow

10 rates and correspondingly shallow depths.

**5    Summary and future work**

Our aim in this paper was to (1) evaluate possible citizen science streamflow measurement methods, (2) select a preferred approach, and (3) pilot test the selected method in a real-world setting. We evaluated three different approaches (i.e. float,

15 salt dilution, and Bernoulli run-up) by performing 20 side by side comparison measurements in headwater catchments of the Kathmandu Valley. We used USGS mid-section discharge measurements from a SonTek FlowTracker acoustic Doppler velocimeter as reference flows. Evaluated flows ranged from 0.006 to 0.240 $m^3$ $s^{-1}$. Linear regressions forced through the origin for scatter plots with reference flows had slopes of 1.05, 1.01, and 1.26 with r-squared values of 0.90, 0.98, and 0.61, for float, salt dilution, and Bernoulli run-up methods, respectively. The salt dilution method was selected as the preferred

20 approach based on its favourable quantitative results compared to the other methods, and other qualitative factors concerning citizen science repeatability. The approach was then pilot tested in a CS Flow Campaign, which involved 20 volunteers performing 145 measurements, ranging from 0.0004 to 0.425 $m^3$ $s^{-1}$, distributed among the 10 headwater catchments of the Kathmandu Valley. While there was no way to evaluate the accuracy of all 145 measurements, five of the measurements were performed in locations where USGS mid-section method discharge measurements had been performed. For these five

25 locations, a linear regression forced through the origin between reference flows and CS Flow measurements had a slope of 0.90 with an r-squared value of 0.97.

Motivated by these promising results, future work should further evaluate the feasibility of applying citizen science based salt dilution streamflow measurements to larger areas of Nepal and beyond. Issues of how to motivate citizen scientists and

30 young researchers (i.e. all science and engineering minded students from primary through graduate school ages) to participate in citizen science streamflow measurement efforts should receive additional attention, especially in the relatively

**Number: 1  Author: reviewer  Subject: Note  Date: 2018-09-14 18:13:23**

If your flow was in the order of a few cm than standard friction coefficients/correction factors need to be adapted.

**Number: 2  Author: reviewer  Subject: Note  Date: 2018-09-14 18:13:53**

in terms of what?

**Number: 3  Author: reviewer  Subject: Note  Date: 2018-09-14 18:16:07**

please mention this relevant information in the method section!! Age, education level etc.)

[Figure]

[Figure]

unexplored context of citizen science in Asia. Finally, the assumption of a constant calibration coefficient (k) should be evaluated over a larger sample size covering a broader range of geological and water quality conditions.

**6    Data availability**

5   The data used in this paper are provided as supplementary material.

**7    Author contribution**

Jeffrey C. Davids had the initial idea for this investigation and designed the experiments in collaboration with Martine M. Rutten, Wessel David van Oyen, and Nick van de Giesen. Field work was performed by Jeffrey C. Davids, Anusha Pandey, Nischal Devkota, Wessel David van Oyen, and Rajaram Prajapati. Jeffrey C. Davids prepared the manuscript with valuable
10   contributions from all co-authors.

**8    Competing interests**

The authors declare that they have no conflict of interest.

**9    Acknowledgements**

This work was supported by the Swedish International Development Agency under Grant number 2016-05801; and by
15   SmartPhones4Water (S4W). We appreciate the dedicated efforts of Annette van Loosen, Bhumika Thapa, Sunil Duwal, volunteers from Khwopa College of Engineering, Anurag Gyawali, Anu Grace Rai, Sanam Tamang, Eliyah Moktan, Kristi Davids, and the rest of the S4W-Nepal team. We would also like to thank Dr. Ram Devi Tachamo Shah, Dr. Deep Narayan Shah, Dr. Narendra Man Shakya, and Dr. Steve Lyon for their supervision and support of this work. Finally, a special thanks to SonTek for their donation of a FlowTracker acoustic Doppler velocimeter that was used for the reference flow
20   measurements discussed in this paper and many more to come.

This research was performed in the context of a larger citizen science project called SmartPhones4Water or S4W (Davids et al. 2017; Davids et al. 2018; www.SmartPhones4Water.org). S4W focuses on leveraging citizen science, mobile technology, and young researchers to improve lives by strengthening our understanding and management of water. S4W's first pilot
25   project, S4W-Nepal, initially concentrated on the Kathmandu Valley, and is now expanding into other regions of the country and beyond. All of S4W's efforts, including the research herein, have a focus on simple field data collection methods that

No Comments.

[Figure]

can be standardized and scaled so that young researchers and citizen scientists can help fill data gaps in other data scarce regions.

**10    References**

Anokwa Y, Hartung C, Brunette W, 2009. Open source data collection in the developing world. Computer 42(10):97–99.
5    http://doi.ieeecomputersociety.org/10.1109/MC.2009.328

Assumpção, T.H., Popescu, I., Jonoski, A. and Solomatine, D.P., 2018. Citizen observations contributing to flood modelling: opportunities and challenges. Hydrology and Earth System Sciences, 22(2), pp.1473-1489.

10    British Standards Institute, 1964. Method of Measurement of Liquid Flow in Open Channels British Standards 3680: Part 3.

Butterworth, J.A., Hewitt, E.J. and McCartney, M.P., 2000. Discharge measurement using portable dilution gauging flowmeters. Water and Environment Journal, 14(6), pp.436-441.

15    Buytaert, W., Zulkafli, Z., Grainger, S., Acosta, L., Alemie, T.C., Bastiaensen, J., De Bièvre, B., Bhusal, J., Clark, J., Dewulf, A. and Foggin, M., 2014. Citizen science in hydrology and water resources: opportunities for knowledge generation, ecosystem service management, and sustainable development. Frontiers in Earth Science, 2, p.26.

Church, M. and Kellerhals, R., 1970. Stream gauging techniques for remote areas using portable equipment. Technical
20    Bulletin No. 25, Inland Waters Branch, Department of Energy, Mines, and Resources.

Davids, J.C., van de Giesen, N. and Rutten, M., 2017. Continuity vs. the crowd—tradeoffs between continuous and intermittent citizen hydrology streamflow observations. Environmental management, 60(1), pp.12-29.

25    Davids, J.C., Rutten, M.M., Shah, R.D.T., Shah, D.N., Devkota, N., Izeboud, P., Pandey, A. and van de Giesen, N., 2018. Quantifying the connections—linkages between land-use and water in the Kathmandu Valley, Nepal. Environmental monitoring and assessment, 190, pp.1-17.

Day, T.J., 1976. On the precision of salt dilution gauging. Journal of Hydrology, 31(3-4), pp.293-306.
30
Day, T.J., 1977. Observed mixing lengths in mountain streams. Journal of Hydrology, 35(1977), pp.125-136.

No Comments.

[Figure]

Dramais, G., Le Coz, J., Camenen, B. and Hauet, A., 2011. Advantages of a mobile LSPIV method for measuring flood discharges and improving stage–discharge curves. Journal of Hydro-Environment Research, 5(4), pp.301-312.

Durand, M., Neal, J., Rodríguez, E., Andreadis, K.M., Smith, L.C. and Yoon, Y., 2014. Estimating reach-averaged discharge
5 for the River Severn from measurements of river water surface elevation and slope. Journal of Hydrology, 511, pp.92-104.

Escurra, J., 2004. Field Calibration of the Float Method in Open Channels.

Feki, H., Slimani, M. and Cudennec, C., 2016. Geostatistically based optimization of a rainfall monitoring network
10 extension: case of the climatically heterogeneous Tunisia. Hydrology Research, p.nh2016256.

Fienen, M.N. and Lowry, C.S., 2012. Social. Water—A crowdsourcing tool for environmental data acquisition. Computers & Geosciences, 49, pp.164-169.

15 Fleming, B. and Henkel, D., 2001. Community-based ecological monitoring: a rapid appraisal approach. Journal of the American Planning Association, 67(4), pp.456-465.

Hannah, D.M., Demuth, S., van Lanen, H.A., Looser, U., Prudhomme, C., Rees, G., Stahl, K. and Tallaksen, L.M., 2011. Large-scale river flow archives: importance, current status and future needs. Hydrological Processes, 25(7), pp.1191-1200.
20
Harmel, R.D., Cooper, R.J., Slade, R.M., Haney, R.L. and Arnold, J.G., 2006. Cumulative uncertainty in measured streamflow and water quality data for small watersheds. Transactions of the ASABE, 49(3), pp.689-701.

Herschy, R.W., 2014. Streamflow measurement. CRC Press.
25
Kelvin, W., 1883. From lecture to the Institution of Civil Engineers, London (3 May 1883),'Electrical Units of Measurement', Popular Lectures and Addresses (1889), 1, 80–81.

Kirchner, J.W., 2006. Getting the right answers for the right reasons: Linking measurements, analyses, and models to
30 advance the science of hydrology. Water Resources Research, 42(3).

Kruger, L.E. and Shannon, M.A., 2000. Getting to know ourselves and our places through participation in civic social assessment. Society & Natural Resources, 13(5), pp.461-478.

No Comments.

[Figure]

[Figure]

Kundzewicz, Z.W., 1997. Water resources for sustainable development. Hydrological Sciences Journal, 42(4), pp.467-480.

Lasaga, A.C., 1984. Chemical kinetics of water-rock interactions. Journal of geophysical research: solid earth, 89(B6), pp.4009-4025.

Le Coz, J., Hauet, A., Pierrefeu, G., Dramais, G. and Camenen, B., 2010. Performance of image-based velocimetry (LSPIV) applied to flash-flood discharge measurements in Mediterranean rivers. Journal of hydrology, 394(1-2), pp.42-52.

Le Coz J, Patalano A, Collins D, Guillén NF, García CM, Smart GM, Bind J, Chiaverini A, Le Boursicaud R, Dramais G, Braud I. Crowdsourced data for flood hydrology: Feedback from recent citizen science projects in Argentina, France and New Zealand. Journal of Hydrology. 2016 Oct 1;541:766-77.

Le Boursicaud, R., Pénard, L., Hauet, A., Thollet, F. and Le Coz, J., 2016. Gauging extreme floods on YouTube: application of LSPIV to home movies for the post-event determination of stream discharges. Hydrological Processes, 30(1), pp.90-105.

Lee Rodgers, J. and Nicewander, W.A., 1988. Thirteen ways to look at the correlation coefficient. The American Statistician, 42(1), pp.59-66.

Mazzoleni, M., Verlaan, M., Alfonso, L., Monego, M., Norbiato, D., Ferri, M. and Solomatine, D.P., 2015. Can assimilation of crowdsourced streamflow observations in hydrological modelling improve flood prediction?. Hydrology & Earth System Sciences Discussions, 12(11).

McCulloch, J.S.G., 1996. Book Review: Streamflow measurement. Reginald W. Herschy, 1995, E. and FN Spon, an imprint of Chapman and Hall, London, 524 pp., ISBN 0-419-19490-8. Journal of Hydrology, 176, pp.285-286.

Mishra, A.K. and Coulibaly, P., 2009. Developments in hydrometric network design: A review. Reviews of Geophysics, 47(2).

Moore, R.D. 2004a. Introduction to salt dilution gauging for streamflow measurement: Part I. Streamline Watershed Management Bulletin 7(4):20–23.

Moore, R.D. 2004b. Introduction to salt dilution gauging for streamflow measurement Part II: Constant-rate injection. Streamline Watershed Management Bulletin 8(1):11–15.

No Comments.

[Figure]

Moore, R.D. 2005. Introduction to salt dilution gauging for streamflow measurement Part III: Slug Injection Using Salt in Solution. Streamline Watershed Management Bulletin 8(2):1–6.

Mulligan, M., 2013. WaterWorld: a self-parameterising, physically based model for application in data-poor but problem-rich environments globally. Hydrology Research, 44(5), pp.748-769.

Muste, M., Fujita, I. and Hauet, A., 2008. Large-scale particle image velocimetry for measurements in riverine environments. Water resources research, 44(4).

O'Grady, M.J., Muldoon, C., Carr, D., Wan, J., Kroon, B. and O'Hare, G.M., 2016. Intelligent sensing for citizen science. Mobile Networks and Applications, 21(2), pp.375-385.

Pearson, C.P., 1998. Changes to New Zealand's national hydrometric network in the 1990s. Journal of Hydrology (New Zealand), pp.1-17.globally. Hydrology Research, 44(5), pp.748-769.

Rantz, S.E. 1982. Measurement and computation of streamflow: volume 2, computation of discharge (No. 2175). USGS.

Sanz, S.F., Holocher-Ertl, T., Kieslinger, B., Sanz, F., Candida, G. and Silva, G., 2014. White Paper on Citizen Science in Europe, Socientize Consortium.

Shrestha, S., Pradhananga, D., and Pandey, V.P., 2012. Kathmandu Valley Groundwater Outlook, Asian Institute of Technology (AIT), The Small Earth Nepal (SEN), Center of Research for Environment Energy and Water (CREEW), International Research Center for River Basin Environment - University of Yamanashi (ICRE-UY), Kathmandu, Nepal.

SonTek, 2009. FlowTracker Handheld ADV Technical Manual Firmware Version 3.7 Software Version 2.30. SonTek/YSI.

SRTM (Shuttle Radar Topography Mission), 2000. https://lta.cr.usgs.gov/SRTM1Arc, Accessed 9/14/16.

Tauro, F., Selker, J., van de Giesen, N., Abrate, T., Uijlenhoet, R., Porfiri, M., Manfreda, S., Caylor, K., Moramarco, T., Benveniste, J. and Ciraolo, G., 2018. Measurements and Observations in the XXI century (MOXXI): innovation and multi-disciplinarity to sense the hydrological cycle. Hydrological Sciences Journal, 63(2), pp.169-196.

Tourian, M.J., Sneeuw, N. and Bárdossy, A., 2013. A quantile function approach to discharge estimation from satellite altimetry (ENVISAT). Water Resources Research, 49(7), pp.4174-4186.

No Comments.

[Figure]

[Figure]

Turner, D.S. and Richter, H.E., 2011. Wet/dry mapping: using citizen scientists to monitor the extent of perennial surface flow in dryland regions. Environmental Management, 47(3), pp.497-505.

5   US Department of the Interior Bureau of Reclamation (USBR), 2001. Water measurement manual.

Van de Giesen, N., Hut, R. and Selker, J., 2014. The Trans-African Hydro-Meteorological Observatory (TAHMO). Wiley Interdisciplinary Reviews: Water, 1(4), pp.341-348.

10  Van Meerveld, H.J., Vis, M.J. and Seibert, J., 2017. Information content of stream level class data for hydrological model calibration. Hydrology and Earth System Sciences, 21(9), pp.4895-4905.

Wilm, H.G. and Storey, H.C., 1944. Velocity-head rod calibrated for measuring stream flow. Civil Engineering, 14, pp.475-476.

No Comments.

---

## Author Comment (AC1) · 2 Oct 2018

Responses to Reviewer #1's Comments:

Please see general responses to your helpful comments below in blue (original comments in **black**). Once all the edits (per the details below) are finalized, a marked-up version of the manuscript showing all changes, along with specific responses to reviewers' comments will be provided.

This manuscript aims at studying the potential for citizen science streamflow measurement methods. Citizen science is so far underused in hydrology and studies on this topic are, thus, much welcome. The manuscript starts with a well-written introduction, where several relevant studies are cited. After this promising start, however, I was rather disappointed by the study. I really like the aim of this study, and I appreciate the attempt to evaluate the suitability of different streamflow gauging methods, but in the end, I have three major concerns. These are related to 1) the study design and data collection, 2) the data analyses and 3) statements that are not supported by the data analyses presented in the manuscript. A more detailed discussion of these issues and some minor comments are provided below.

RESPONSE: Your three main concerns are well received. It was timely to receive your comments when we did because we were just starting our post-monsoon Citizen Science (CS) Flow campaign in Kathmandu. Based on your comments, and Reviewer #2's feedback, we were able to design and implement additional data collection which was performed from 18 to 20 of September 2018. We feel that once these data have been incorporated along with the other suggested edits the revised manuscript will be strengthened.

From 18 to 20 of September 2018, we facilitated measurements at 15 sites in two different watersheds in the Kathmandu Valley. Ten CS Flow groups, each comprised of three students, performed all three methods (i.e. float, salt dilution, and Bernoulli) at each site. At the same time, an "expert" group (authors) performed the same three methods at the same sites, along with a FlowTracker ADV reference flow measurement. After the field measurements, all the CS Flow participants completed a survey about their experiences with (and perceptions of) each simple measurement method.

In the original version of the manuscript, there was some confusion about the two different data sets being evaluated, including who actually had generated the data. An additional table has been added to clarify the three phases (and datasets) of the study: (1) initial evaluation (authors), (2) citizen scientist evaluation (authors and CS Flow groups), and (3) citizen scientist application (CS Flow groups).

| # | Phase | Description | Performed By | Period | Season |
|---|-------|-------------|--------------|--------|--------|
| 1 | Initial Evaluation | Initial evaluation of three simple flow measurement methods (i.e. float, salt dilution, and Bernoulli) along with FlowTracker ADV reference flow measurements at 20 sites within the Kathmandu Valley. Reference flows ranging from 6.4 to 240 liters per second (L $s^{-1}$). | Authors | March/ April 2017 | Pre-monsoon |

| 2 | Citizen Scientist Evaluation | Citizen Scientist evaluation of three simple flow measurement methods (i.e. float, salt dilution, and Bernoulli) along "expert" and FlowTracker ADV reference flow measurements at 15 sites within the Kathmandu Valley.  Reference flows ranging from 4.2 to 896 L s$^{-1}$. | Authors for "expert" and reference flows PLUS 10 Citizen Science Flow groups for simple methods | September 2018 | Post-monsoon |
|---|---|---|---|---|---|
| 3 | Citizen Scientist Application | Salt dilution measurements at roughly 150 sites in the 10 perennial watersheds of the Kathmandu Valley.  Float measurements with a small number of sub-sections (e.g. 3 to 5) performed at each site to determine salt quantities. | 17 Citizen Science Flow groups (7 from April and 10 from September) | April and September 2018 | Pre and Post Monsoon |

I am afraid that the concerns related to the available data require additional data to be collected. Frankly, I would say the presented work is an interesting pre-study, but a better study design and data collection are needed to obtain useful results. Publishing the preliminary results as presented here could do more harm than good as people might use the conclusions without being aware that there actually was little data evidence. Given the importance of the topic, I hope the authors will be able to do this and will resubmit a study, which addresses the issues they raise in this manuscript.

1. There are several severe flaws in the study design and in the end I am afraid the authors did not collect the data that would be needed to address the questions they wanted to study.
    a. It is highly unfortunate that there are no concurrent flow measurements for the 'true' flow available. Flow measurements taken a few weeks apart are just not the basis for a serious evaluation. It is also surprising the different 'citizen scientists' were asked to measure streamflow at different sites. It would have much more informative to let them measure the same stream and about the same conditions.

RESPONSE:  At all 15 sites, the new data we collected has CS measurements and "expert" measurements of each simple method along with a reference flow with the FlowTracker ADV.  We believe this now provides the data we need to make a comparison.

    b. The authors mention that three salt dilution measurements were excluded as outliers. While they present some explanation (which I do not fully agree), they do not present anything that would help to detect such cases in an application where there is no comparison with any other gauging. In other words, in a real application, these values would pass undetected, and the potential error, thus, would be much larger than reported here. Note that almost half of the cases with comparison streamflow data were excluded! Again, it is unfortunate that the authors need to very speculative about what might have happened because of the study design.

RESPONSE: We will re-evaluate this concern in light of the new data.  Your point about not being able to detect such errors without a reference flow is well received and will be incorporated into the revised version of the manuscript.

c. Related to the above comment, one potential issue are mistakes that could be done by 'citizen scientists'. With a better study design (e.g., more groups at the same place, 'secret' observer, . . .), this could have been addressed.

RESPONSE: Because the new measurements are from the same 15 sites, we are now able to identify the variability in CS Flow group measurements, compared to the actual value, and "expert" values with the same methods (i.e. float, salt dilution, or Bernoulli).

d. Basically, there are two separate questions: 1) which of the 'simple' gauging methods provides best results (with 'perfect' persons) and 2) how re the methods used by 'citizen scientists'. By deciding the best methods already after the first step, the authors, unfortunately, do not fully explore which method is most suitable for citizen science approaches.

RESPONSE: To help answer both of these questions, the CS Flow groups now performed all three methods. Participants also completed a quantitative evaluation of their experience with (and perception of) each measurement methods to improve our understanding of citizen science suitability.

e. The accuracy of the salt dilution measurement depends largely on the selected site (mixing, flow volume, and velocity. . .), and depending on the site, thus, different methods might be most suiatble. Again, this is an important aspect that could have been adddressed with a better study design.

RESPONSE: Each CS Flow group was allowed to select their specific salt dilution measurement reach independently. They also performed Float and Bernoulli measurements, so now we will be able to make a full comparison of which method performed best, and how this varied with the type of site.

f. A minor point related to the study design: when the aim is to obtain relations of the calibrated k-factors with elevation or other variables, k should have been determined at as may places as possible and not just half of them. I am also not sure whether it is reasonable to use the mean k value was for the 10 locations without individual measurements, but the individual values for the others. I would rather have expected to use the mean or some regionalized values for ALL locations to ensure comparability.

RESPONSE: For the first 20 measurement sites performed by the authors, we will use the average of the 10 K values obtained for all 20 sites.

2. The data analyses contain some questionable use of statistics:
   a. Averaging of errors (tables 12): averaging positive and negative errors just does not make any sense, this makes the results look much better than they are. Instead, one should base the analyses on the absolute values so that positive and negative errors do not cancel out each other

RESPONSE: The averages for errors now average the absolute values so that positive and negative errors do not cancel each other out.

      b. The correlations shown in figures 3 or 4 (and reported in the abstract) are misleading. These are spurious correlations! Comparing streams of different size, of course, one gets high r2 values. Imagine two persons would measure the height of a group of people, even if the individual measurements would be off by 5 cm, the correlation of the heights would still be large simply because some people are much taller than others. Please be more careful when using statistics.

RESPONSE: Instead of scatter plots, we will instead present box plots showing the distribution of error for each site.

3. The statements in section 4.4 are not really supported by the data in this study. The number of persons needed in each group, for instance, has not been tested. Also, the inexpensive EC meter has not been tested (or has it?  comparison?).

RESPONSE: Participants in the post-monsoon (i.e. phase 2) measurements completed quantitative evaluations of the methods, so now we have additional evidence to answer these questions. Additionally, testing of the inexpensive EC meters has been performed and incorporated into the manuscript.

      a. As another example, the statement of enjoyability seems not to be supported, and actually, the other methods have not been tested with 'citizen scientists. The authors also need to explain much better which type of citizen scientist they refer to. Form the title (where the term citizen science is used twice!), the abstract, the introduction and section 4.4 one gets the impression that this is about citizen science in a broad meaning. However, looking more closely at what has been done, it seems that the work does not address the participation of the general public in science but is based on selected individuals, which received a significant education. This is fine, but is a rather special case of citizen science.

RESPONSE: Participants in the post-monsoon (i.e. new) measurements completed quantitative evaluations of the methods, so now we have additional evidence to answer these questions.  Additional clarifying language has been added to make it clear that we are initially targeting students as citizen scientists, but that this represents a narrow swatch of possible citizen scientists.  Our experience has shown that in countries like Nepal, students provide an important "first wave" of citizen scientists, who can later promote citizen science to local community members.  Over time, it is our goal to continue to expand the type of citizen scientists that we target and engage.

4. Minor comments:
      a. P4: which factor for c was used in the end? Variable or constant. This needs to be included in the steps.

RESPONSE: A constant C factor of 0.8 was used.  This has been added to the steps.

      b. P8: too little information is given about the 'citizen scientists': how old? Gender? Students, but which topic (how much hydrology or environmental engineering?), how large groups, . . ..

RESPONSE: A summary of the age, gender, major, and size of groups has now been included.

      c. What is the purpose of showing figure 2?

RESPONSE: The original purpose was to show the breakthrough curves in order to illustrate that they are similar to standard curves.  However, due to the large number of figures from the newly collected data, these graphs have now been removed.

      d. P5L15: where does the value of 1667 g per m3/2 come from. Moore (2005) recommend a different value

RESPONSE: We have clarified the text to now state that an approximate average of values shown in Table 1 of Moore (2005) are the basis for our salt dose recommendations.  The average mass of the studies in Moore (2005) is 1600 g per $m^3 s^{-1}$.

      e. Tables 1 and 2: providing runoff with four digits seems a bit too accurate, especially given that the observations actually were weeks apart.

RESPONSE: Table 2 has now been removed.  Because all flows observed are less than 1 $m^3 s^{-1}$, we have decided to present flows in liters per second ($L s^{-1}$).  Flow greater than or equal to 100 $L s^{-1}$ are now shown to the nearest integer.  Flows less than 100 $L s^{-1}$ are shown with two significant digits.

      f. Please check the author guidelines, especially with regard to the date format and equations

RESPONSE: The author guidelines for date format and equations has been checked and the necessary revisions have been made.

---

## Author Comment (AC3) · 24 Oct 2018

Please see the supplement for our revised manuscript (mark up and clean version) per your comments and our additional field work.

Please also note the supplement to this comment:
https://www.hydrol-earth-syst-sci-discuss.net/hess-2018-425/hess-2018-425-AC3-supplement.zip
* * *

---

## Author Comment (AC4) · 24 Oct 2018

Please see the supplement for our revised manuscript (mark up and clean version) per your comments and our additional field work. The zipped folder also contains responses to your helpful and detailed comments on the original manuscript.

Please also note the supplement to this comment: https://www.hydrol-earth-syst-sci-discuss.net/hess-2018-425/hess-2018-425-AC4-supplement.zip

---

## Author Response (AR1)

Dear Dr. Pfister and Reviewers,

We are pleased to submit this revised manuscript for further review and comment. Most importantly, we believe that the additional data and revised methodologies, results, and discussion described in more detail below (and in the revised manuscript) address the major concerns regarding data limitations and methodology originally identified by the reviewers.

It was timely to receive comments when we did because we were just starting our post-monsoon Citizen Science (CS) Flow campaign in Kathmandu. Based on the reviewers' comments we were able to design and implement additional data collection which was performed in mid-September.

From 18 to 20 of September 2018, we facilitated measurements at 15 sites in two different watersheds in the Kathmandu Valley. In the revised manuscript, these measurements are referred to as Phase 2. Ten CS Flow groups, each comprised of three students, performed all three methods (i.e. float, salt dilution, and Bernoulli) at each site. At the same time, an "expert" group (authors) performed the same three methods at the same sites, along with a FlowTracker ADV reference flow measurement. After the field measurements, all the CS Flow participants completed a survey about their experiences with (and perceptions of) each simple measurement method.

Importantly, in the original version of the manuscript, there was some confusion about the two different data sets being evaluated, including who actually had generated the data. An additional table has been added to clarify the three phases (and datasets) of the study: (Phase 1) initial evaluation (authors), (Phase 2) citizen scientist evaluation (authors and CS Flow groups), and (Phase 3) citizen scientist application (CS Flow groups).

Table 2 from the Manuscript (Sect. 2.2)

| # | Phase | Description | Performed By | Period | Season |
|---|-------|-------------|--------------|--------|--------|
| 1 | Initial Evaluation | Initial evaluation of three simple flow measurement methods (i.e. float, salt dilution, and Bernoulli) along with FlowTracker ADV reference flow measurements at 20 sites within the Kathmandu Valley. Reference flows ranging from 6.4 to 240 liters per second (L $s^{-1}$). | Authors | March/ April 2017 | Pre-monsoon |
| 2 | Citizen Scientist Evaluation | Citizen Scientist evaluation of three simple flow measurement methods (i.e. float, salt dilution, and Bernoulli) along "expert" and FlowTracker ADV reference flow measurements at 15 sites within the Kathmandu Valley. Reference flows ranging from 4.2 to 896 L $s^{-1}$. | Authors for "expert" and reference flows PLUS 10 Citizen Science Flow groups for simple methods | September 2018 | Post-monsoon |

| | | | | | |
|---|---|---|---|---|---|
| 3 | Citizen Scientist Application | Salt dilution measurements at roughly 150 sites in the 10 perennial watersheds of the Kathmandu Valley. Float measurements with a small number of sub-sections (e.g. 3 to 5) performed at each site to determine salt quantities. | 18 Citizen Science Flow groups (8 from April and 10 from September) | April and September 2018 | Pre and Post Monsoon |

The following is a list of major changes to the manuscript per the helpful comments from the reviewers and the additional data collected. The changes are organized by section. As you can see, the changes are somewhat extensive, so it is difficult to make a comprehensive list of changes. We trust that this summary and the tracked changes version later in this PDF will be sufficient for the purposes of seeing what has been modified.

- Miscellaneous
    - Revised author affiliations
- Abstract
    - Revised abstract per edits described below
- Introduction
    - Removed original first paragraph of the introduction
    - Combined portions of the first paragraph with the second in t
    - Added references for other smartphone apps that can be used to measure flow (Lüthi et al. 2014; Peña-Haro et al. 2018)
    - Added paragraph about the potential challenges with citizen science streamflow measurements including data quality and intermittent timing
    - Added Sect 1.2 - Description of eight different simple streamflow measurement methods originally considered along with the criteria and scoring for how the final three were selected (Table 1)
    - Added Sect. 1.3 - Expanded description of the three different simple streamflow measurement methods
    - Added Sect. 1.4 - Explicit statement of four research questions
    - Added Sect. 1.5 - Explicit description of SmartPhone4Water to give context early on in the manuscript of who is performing the research and why
- Materials and Methods
    - Added Sect. 2.1.1 - Describes the types of streams investigated in the paper
    - Added Sect. 2.1.2 - Describes how references flows were taken
    - Added Sect. 2.1.3 - Explicit discussion of salt dilution calibration coefficients (k) and which values were used for which phases of the project
    - Added Sect. 2.1.4 - Evaluation of cheap EC meter accuracy
        - Added Figure 1 - boxplot of cheap EC meter error for different measurement ranges
    - Added Sect. 2.2 - Reorganization of the three phases of the project (i.e. Phase 1 - Initial evaluation; Phase 2 - Citizen scientist evaluation; Phase 3 - Citizen scientist application)
        - Added Table 2 - Description of these three phases including who the field work was performed by, what period, and what season
    - Added Sect. 2.2.1 - Reorganized existing methodology to form Phase 1 methods

- o Added Sect. 2.2.2 - Included new methodology for Phase 2 citizen scientist evaluation at 15 sites with all three simple streamflow measurement methods and expert group methods with reference flow measurements
  - Included more details about the citizen scientists involved including age, education, number of participants, genders
  - Added methodology for citizen scientist surveys with eight questions to understand their perspectives on the three different streamflow measurement methods evaluated
- o Added Sect. 2.2.3 - Revised existing methodology to include new data from the fall (post-monsoon) in addition to the spring (pre-monsoon) 2018
  - Included more details about the citizen scientists involved including age, education, number of participants, genders
  - Added in brief discussion of the government gauging locations (number of stations and locations) to illustrate the importance of this work
- Results
  - o Reorganized results completely based on the three new phases of the investigation per the methods in Sect. 2
  - o Sect. 3.1 - Phase 1 Initial Evaluation
    - Modified Fig. 2 - added new SiteIDs and included a picture of a typical site
    - Modified Table 3 - Renamed 20 measurement sites from Phase 1 (i.e. 1 to 20) instead of complicated measurement ids
      - Included average of absolute errors in addition to average of errors (bias)
    - Removed breakthrough curves from Phase 1 data (old Fig. 2)
    - Removed scatter plots of streamflow measurement methods against reference flow (old Fig. 3)
    - Removed correlation analysis for measurement errors (old Fig. 4) and k (old Fig. 5)
  - o Sect. 3.2 - Phase 2 Citizen Scientist Evaluation
    - Added Fig. 3 - overview of the two watersheds for Phase 2 measurements including expanding view of the Dhobi and Nakkhu sites (n = 15)
    - Added Table 4 - summary of Phase 2 data collection for citizen scientist evaluation of the simple streamflow measurement methods
    - Added Fig. 4 - boxplots of CS Flow group errors for the 15 sites for each measurement method in addition to "expert" errors for all 15 sites for all three methods
    - Added Fig. 5 - summary of results from CS Flow group perception questions (n = 8) for the three different methods evaluated
  - o Sect. 3.3 - Phase 3 Citizen Scientist Application
    - Updated Phase 3 to include data from the pre-monsoon (spring) and post-monsoon (fall) of 2018
    - Modified Fig. 6 - updated figure with data from pre and post-monsoon along with flow and EC histogram to show seasonal changes in flow and water quality distributions

- ▪ Removed old scatter plot of float coefficients and associated discussion (not the focus of this paper)
- ▪ Removed old comparison of measured flows because of the time different between measurements and reference flows
- Discussion
  - o Reorganized discussion to match three phases discussed in methods and results
  - o Updated to explicitly answer each of the four research questions
  - o Sect. 4.1 - Phase 1
    - ▪ Added more discussion on challenges with Bernoulli measurements
    - ▪ Improved discussion regarding challenge of surface velocity measurements in shallow slow moving areas
    - ▪ Added discussion of the impact of using an average k on the resulting flow
  - o Sect. 4.2 - Phase 2
    - ▪ Completely new discussion regarding the CS Flow group evaluations
    - ▪ Added discussion of the comparison between CS Flow and "expert" group measurements
    - ▪ Added discussion of how Phase 1 results compare with "expert" results from Phase 2
    - ▪ Added discussion of the results of the perception survey+
  - o Sect. 4.3 - Phase 3
    - ▪ Added new discussion to review application of salt dilution measurements in the Kathmandu Valley in the pre and post-monsoon periods
- Summary
  - o Added Table 5 - Summary of average absolute errors, average biases, and error standard deviations for the three Phases of work
  - o Edited brief summary per the changes outlined above
  - o Updated final paragraph with recommendation to explore the information content of additional streamflow data and how to effectively recruit citizen scientists for future streamflow measurement campaigns
- References
  - o Added seven new references per the changes above

We very much look forward to receiving your feedback on the revised manuscript.

Best Regards,

jeff (for the rest of the authors)

Responses to Reviewer #1's Comments:

Please see general responses to your helpful comments below in blue (original comments in black). Once all the edits (per the details below) are finalized, a marked-up version of the manuscript showing all changes, along with specific responses to reviewers' comments will be provided. Most importantly, we believe that the additional data and revised methodologies described in more detail below (and in the revised manuscript) address the major concerns regarding data limitations and methodology.

This manuscript aims at studying the potential for citizen science streamflow measurement methods. Citizen science is so far underused in hydrology and studies on this topic are, thus, much welcome. The manuscript starts with a well-written introduction, where several relevant studies are cited. After this promising start, however, I was rather disappointed by the study. I really like the aim of this study, and I appreciate the attempt to evaluate the suitability of different streamflow gauging methods, but in the end, I have three major concerns. These are related to 1) the study design and data collection, 2) the data analyses and 3) statements that are not supported by the data analyses presented in the manuscript. A more detailed discussion of these issues and some minor comments are provided below.

RESPONSE: Your three main concerns are well received. It was timely to receive your comments when we did because we were just starting our post-monsoon Citizen Science (CS) Flow campaign in Kathmandu. Based on your comments, and Reviewer #2's feedback, we were able to design and implement additional data collection which was performed from 18 to 20 of September 2018. We feel that the revised manuscript which incorporates these new data along with the other suggested edits is strengthened.

From 18 to 20 of September 2018, we facilitated measurements at 15 sites in two different watersheds in the Kathmandu Valley. In the revised manuscript, these measurements are referred to as Phase 2. Ten CS Flow groups, each comprised of three students, performed all three methods (i.e. float, salt dilution, and Bernoulli) at each site. At the same time, an "expert" group (authors) performed the same three methods at the same sites, along with a FlowTracker ADV reference flow measurement. After the field measurements, all the CS Flow participants completed a survey about their experiences with (and perceptions of) each simple measurement method.

In the original version of the manuscript, there was some confusion about the two different data sets being evaluated, including who actually had generated the data. An additional table has been added to clarify the three phases (and datasets) of the study: (Phase 1) initial evaluation (authors), (Phase 2) citizen scientist evaluation (authors and CS Flow groups), and (Phase 3) citizen scientist application (CS Flow groups).

| # | Phase | Description | Performed By | Period | Season |
|---|-------|-------------|--------------|--------|--------|
| 1 | Initial Evaluation | Initial evaluation of three simple flow measurement methods (i.e. float, salt dilution, and Bernoulli) along with FlowTracker ADV reference flow measurements at 20 sites within the Kathmandu Valley. Reference flows | Authors | March/ April 2017 | Pre-monsoon |

| | | | | | |
|---|---|---|---|---|---|
| | | | | | ranging from 6.4 to 240 liters per second (L $s^{-1}$). |

| | | | | | |
|---|---|---|---|---|---|
| 2 | Citizen Scientist Evaluation | Citizen Scientist evaluation of three simple flow measurement methods (i.e. float, salt dilution, and Bernoulli) along "expert" and FlowTracker ADV reference flow measurements at 15 sites within the Kathmandu Valley. Reference flows ranging from 4.2 to 896 L $s^{-1}$. | Authors for "expert" and reference flows PLUS 10 Citizen Science Flow groups for simple methods | September 2018 | Post-monsoon |
| 3 | Citizen Scientist Application | Salt dilution measurements at roughly 150 sites in the 10 perennial watersheds of the Kathmandu Valley. Float measurements with a small number of sub-sections (e.g. 3 to 5) performed at each site to determine salt quantities. | 18 Citizen Science Flow groups (8 from April and 10 from September) | April and September 2018 | Pre and Post Monsoon |

I am afraid that the concerns related to the available data require additional data to be collected. Frankly, I would say the presented work is an interesting pre-study, but a better study design and data collection are needed to obtain useful results. Publishing the preliminary results as presented here could do more harm than good as people might use the conclusions without being aware that there actually was little data evidence. Given the importance of the topic, I hope the authors will be able to do this and will resubmit a study, which addresses the issues they raise in this manuscript.

As previously mentioned, based on your feedback, we were able to collect additional data that enables us to have more robust and useful results. These data, and the associated methodologies, results, and discussion have been included in the revised manuscript.

1. There are several severe flaws in the study design and in the end I am afraid the authors did not collect the data that would be needed to address the questions they wanted to study.
   a. It is highly unfortunate that there are no concurrent flow measurements for the 'true' flow available. Flow measurements taken a few weeks apart are just not the basis for a serious evaluation. It is also surprising the different 'citizen scientists' were asked to measure streamflow at different sites. It would have much more informative to let them measure the same stream and about the same conditions.

RESPONSE: At all 15 Phase 2 sites, the new data we collected has CS measurements and "expert" measurements of each simple method along with a reference flow with the FlowTracker ADV within at the most +/- one day without precipitation or observable changes in water levels. We believe this now provides the data we need to make a robust and meaningful comparison.

   b. The authors mention that three salt dilution measurements were excluded as outliers. While they present some explanation (which I do not fully agree), they do not present anything that would help to detect such cases in an application where there is no

comparison with any other gauging. In other words, in a real application, these values would pass undetected, and the potential error, thus, would be much larger than reported here. Note that almost half of the cases with comparison streamflow data were excluded! Again, it is unfortunate that the authors need to very speculative about what might have happened because of the study design.

RESPONSE: The newly collected data for Phase 2 has completely changed the way that we are comparing measurements from citizen science groups to "expert" measurements and reference flows in the revised manuscript.  No measurements have been excluded as outliers in the errors shown in Tables 3 and 4 and the summary table in Sect. 5 (Table 5).

c.  Related to the above comment, one potential issue are mistakes that could be done by 'citizen scientists'. With a better study design (e.g., more groups at the same place, 'secret' observer, . . .), this could have been addressed.

RESPONSE: Because the new Phase 2 measurements are from the same 15 sites, we are now able to identify the variability in CS Flow group measurements, compared to reference flow, and "expert" values with the same methods (i.e. float, salt dilution, or Bernoulli).

d.  Basically, there are two separate questions: 1) which of the 'simple' gauging methods provides best results (with 'perfect' persons) and 2) how re the methods used by 'citizen scientists'. By deciding the best methods already after the first step, the authors, unfortunately, do not fully explore which method is most suitable for citizen science approaches.

RESPONSE: To help answer both of these questions, the CS Flow groups now performed all three methods.  Participants also completed a quantitative evaluation (results summarized in Fig. 5) of their experience with (and perception of) each measurement methods to improve our understanding of citizen science suitability.

e.  The accuracy of the salt dilution measurement depends largely on the selected site (mixing, flow volume, and velocity. . .), and depending on the site, thus, different methods might be most suiatble. Again, this is an important aspect that could have been adddressed with a better study design.

RESPONSE: Within the same stream reach (i.e. no visible inflows or outflows), each CS Flow group was allowed to select their specific salt dilution measurement reach independently.  They also performed Float and Bernoulli measurements, so now we have a full comparison of which method performed best, and how this varied between sites (Fig. 4).

f.  A minor point related to the study design: when the aim is to obtain relations of the calibrated k-factors with elevation or other variables, k should have been determined at as may places as possible and not just half of them. I am also not sure whether it is reasonable to use the mean k value was for the 10 locations without individual

measurements, but the individual values for the others. I would rather have expected to use the mean or some regionalized values for ALL locations to ensure comparability.

RESPONSE: For the first 20 measurement sites performed by the authors (Phase 1), we now use an average K as now stated in in Sect. 2.1.3 (dedicated discussion of K) of the 10 K values obtained for all 20 sites. For Phases 2 and 3, we use an average K value from all 15 Phase 2 sites.

2. The data analyses contain some questionable use of statistics:
    a. Averaging of errors (tables 12): averaging positive and negative errors just does not make any sense, this makes the results look much better than they are. Instead, one should base the analyses on the absolute values so that positive and negative errors do not cancel out each other

RESPONSE: The averages for errors now average the absolute values so that positive and negative errors do not cancel each other out. We still include the average of errors as a measure of bias, but we refer to average absolute errors for errors (Tables 3 and 4).

    b. The correlations shown in figures 3 or 4 (and reported in the abstract) are misleading. These are spurious correlations! Comparing streams of different size, of course, one gets high r2 values. Imagine two persons would measure the height of a group of people, even if the individual measurements would be off by 5 cm, the correlation of the heights would still be large simply because some people are much taller than others. Please be more careful when using statistics.

RESPONSE: Instead of scatter plots, we instead present box plots showing the distribution of citizen science group errors for each site, along with "expert" measurement errors (Fig. 4). We now report errors of the methods as the average of the absolute errors (Tables 3 and 4).

3. The statements in section 4.4 are not really supported by the data in this study. The number of persons needed in each group, for instance, has not been tested. Also, the inexpensive EC meter has not been tested (or has it? comparison?).

RESPONSE: Participants in the post-monsoon (i.e. phase 2) measurements completed quantitative evaluations (perception surveys) of the methods, so we now have additional external evidence to answer these questions (Fig. 5). Additionally, testing of the inexpensive EC meters has been performed and incorporated into the manuscript (Sect. 2.1.4).

    a. As another example, the statement of enjoyability seems not to be supported, and actually, the other methods have not been tested with 'citizen scientists. The authors also need to explain much better which type of citizen scientist they refer to. Form the title (where the term citizen science is used twice!), the abstract, the introduction and section 4.4 one gets the impression that this is about citizen science in a broad meaning. However, looking more closely at what has been done, it seems that the work does not address the participation of the general public in science but is based on selected

individuals, which received a significant education. This is fine, but is a rather special case of citizen science.

RESPONSE: Participants in the post-monsoon (i.e. new) measurements completed quantitative evaluations of the methods, so now we have additional evidence to answer these questions (Fig. 5). Additional clarifying language has been added to the last paragraph of the introduction to make it clear that we are initially targeting students as citizen scientists, but that this represents a narrow swatch of possible citizen scientists. Our experience has shown that in countries like Nepal, students provide an important "first wave" of citizen scientists, who can later promote citizen science to local community members. Over time, it is our goal to continue to expand the type of citizen scientists that we target and engage.

4. Minor comments:
    a. P4: which factor for c was used in the end? Variable or constant. This needs to be included in the steps.

RESPONSE: A constant C factor of 0.8 was used. This has been added to the text in Sect. 1.3.1.

    b. P8: too little information is given about the 'citizen scientists': how old? Gender? Students, but which topic (how much hydrology or environmental engineering?), how large groups, . . ..

RESPONSE: A summary of the age, gender, major, and size of groups has now been included in the beginnings of the methodology for Phase 2 (Sect. 2.2.2) and Phase 3 (2.2.3).

    c. What is the purpose of showing figure 2?

RESPONSE: The original purpose was to show the breakthrough curves in order to illustrate that they are similar to typically observed salt dilution curves. However, due to the high number of tables and figures from the newly collected data, these graphs have now been removed.

    d. P5L15: where does the value of 1667 g per m3/2 come from. Moore (2005) recommend a different value

RESPONSE: We have clarified the text to now state that an approximate average of values shown in Table 1 of Moore (2005) are the basis for our salt dose recommendations. The average mass of the studies in Moore (2005) is 1600 g per $m^3$ $s^{-1}$. At a ratio of 6 liters of water for every 1 kg of salt, this ends up being 10 L + 1.6666 kg of tracer solution per $m^3$ $s^{-1}$ of flow.

    e. Tables 1 and 2: providing runoff with four digits seems a bit too accurate, especially given that the observations actually were weeks apart.

RESPONSE: Table 2 has now been removed. Because all but 3 flows observed (all in Phase 3) are less than 1 $m^3$ $s^{-1}$, we have decided to present flows in liters per second (L $s^{-1}$). Flows greater than or equal

to 100 L s$^{-1}$ are now shown to the nearest integer.  Flows less than 100 L s$^{-1}$ are shown with two significant digits (Tables 3 and 4).

    f.    Please check the author guidelines, especially with regard to the date format and equations

RESPONSE: The author guidelines for date format and equations have been checked and the necessary revisions have been made.

Responses to Reviewer #2's Comments:

Please see general responses to your helpful comments below in blue (original comments in black). A marked-up version of the manuscript is included following responses to reviewers' comments. Most importantly, we believe that the additional data and revised methodologies described in more detail below (and in the revised manuscript) address the major concerns regarding data limitations and methodology.

The paper presents results on three simple and easy to use discharge estimation methods appropriate for citizen science (SC) that the authors applied in the Kathmandu Valley, Nepal. They assessed the agreement between the methods and compared estimated discharge to selected measurements using a doppler radar device. The text is short but mainly well written, the graphical presentation is clear and appealing. I recommend to state explicit research questions at the end of the introduction (currently missing). I have also some major concerns about parts of the analysis and the interpretation of the results:

1. While the authors do well in terms of reporting statistical significance of their results, the use of the Pearson Correlation Coefficients seems not appropriate for the properties of the dataset. I therefore recommend the non-parametric Spearman Rank Correlation and the associate non-parametric statistical test.

RESPONSE: Following the suggestion of Reviewer #1, we will remove the correlation analyses and will present the information in a tabular summary with average absolute error and average error (bias). For the newly collected data (see response below point 4 below), we presented the information as box plots showing the distribution of error for each site for both the "experts" and citizen science groups.

2. I question whether it is meaningful or informative to correlate the slope of the salt dilution calibration k to latitude or longitude and elevation and would suggest to skip (or better explain this analysis).

RESPONSE: This analysis has been removed.

3. Instead I recommend to also show the comparison of discharge estimated by salt dilution and by the Bernoulli method.

plots show these comparisons between float, salt dilution, and Bernoulli in a
more comprehensive way including results from "experts" and citizen science groups

RESPONSE: The new box

.

4. I would ask the authors to quantitatively prove that they can compare discharge estimates taken during the CS-campaign with doppler radar observed discharge taken +/- one month (!!) before/after the campaign or skip that part. In the discussion they state themselves that the flow might have decreased during that time.

RESPONSE: Yes, the lack of timely reference flow measurements was a significant challenge with the initial data set. It was timely to receive your comments when we did because we were just starting our post-monsoon Citizen Science (CS) Flow campaign in Kathmandu. Based on your comments, and Reviewer #1's feedback, we were able to design and implement additional data collection which was performed from 18 to 20 (Phase 2) September and 21 to 25 (Phase 3). After incorporating these data, all with the other suggested edits, we feel that the revised manuscript is strengthened.

Summary of additional data collection (referred to as Phase 2 in the report): From 18 to 20 of September 2018, we facilitated measurements at 15 sites in two different watersheds in the Kathmandu Valley. Ten CS Flow groups, each comprised of three students, performed all three methods (i.e. float, salt dilution, and Bernoulli) at each site. At the same time, an "expert" group (authors) performed the same three methods at the same sites, along with a FlowTracker ADV reference flow measurement. After the field measurements, all the CS Flow participants completed a survey about their experiences with (and perceptions of) each simple measurement method.

As the remaining analysis is probably too short for a full publication, I suggest the authors to check whether their dataset would allow additional analysis e.g., on the difference of the quality of the measurements taken by experts and citizens (see my suggestions in the pdf). The current paper is interesting, the dataset promising but the current state of analysis is not enough for a full publication. I therefore encourage the authors for major revisions and additional data analysis.

RESPONSE: As stated above, we have been able to collect additional data that we believe overcomes the challenges associated with the initial dataset. We now have data that evaluates citizen science measurements of all three simple methods against "expert" measurements and timely reference (actual) flows. We also performed surveys of citizen scientists to understand their perceptions of the different methods.

In the following I summarize my suggestions for the individual sections and ask the authors to also check my detailed comments and suggestions that I have included in the pdf (uploaded as supplement):

1. Introduction: The introduction is on the short side and starts a bit philosophical. I would focus more on streamflow and introduce to the problem that large parts of the words still have limited number of gauging stations (especially remote and developing countries) and that measuring devices - while still decreasing in costs - have their limitations. Some other citizen science studies are briefly mentioned but the findings of other studies could be described a bit more in detail. The same applies to the existing methods for low-cost streamflow assessments. Their pros and cons could be compared using a table. I also do not agree that the are no tools on the market, that allow direct measurements of discharge with smartphones and added one link to an example. The research questions should be clearly formulated at the end of the introduction. Please also see more specific comments directly marked in the pdf (uploaded as supplement).

RESPONSE: We have removed the first paragraph. We also describe in more detail the problem of limited gauges, high costs, and other limitations. We have strengthened our description of the findings

of other studies.  The pros and cons of low-cost streamflow methods has been summarized in a table in the Introduction (Table 1).  Reference to the other projects using smartphones for discharge measurements has been included in the second paragraph on p3.  The research questions are more clearly stated in Sect. 1.3 towards the end of the introduction.  Your helpful and detailed comments in the supplemental material have also been addressed in the revised manuscript, and our detailed responses to each comment is included in this PDF.

2.  Methods: The method section needs a better description of the experimental setup; study area, test with students, repetitions y/n etc. (parts are mentioned at the end of the method section but should be stated at the beginning)! The catchments and streams used for testing need a better description (see my suggestions in the pdf). The same applies to the training of the students. The explanation of the different methods is long but can be useful for some non-hydrological readers. I suggest to consider to present all this information in the introduction section. I would however include a list of objective criteria why these three and not other methods have been selected. Please also see more specific comments directly marked in the pdf (uploaded as supplement). The method section should be clearer about the two datasets collected a) dataset with n=20 samples (I assume collected by the authors themselves = exports) and the CS-campaign with n=145 samples collected by citizens. One issue seems critical to me: Authors compare observed discharge using the doppler radar with CS-discharge measurements done +/- 1 moth earlier/later. The authors should prove statistically that the mean daily flow in the month before and after the CS-discharge measurements is not significantly different. In fact authors state in the discussion section that flows decreased during that period.

RESPONSE: Based on your comments, we have updated the methods section.  We have improved the description of the catchments and the training of the citizen scientists (CS).  The explanation of the methods has been moved to the introduction.  The objective criteria has been included in a flow measurement method summary table in the introduction (Table 1).  An additional table (Table 2) has been added to clarify the three phases (and datasets) of the study: (Phase 1) initial evaluation (authors), (Phase 2) citizen scientist evaluation (authors and CS Flow groups), and (Phase 3) citizen scientist application (CS Flow groups).

| # | Phase | Description | Performed By | Period | Season |
|---|-------|-------------|--------------|--------|--------|
| 1 | Initial Evaluation | Initial evaluation of three simple flow measurement methods (i.e. float, salt dilution, and Bernoulli) along with FlowTracker ADV reference flow measurements at 20 sites within the Kathmandu Valley.  Reference flows ranged from 6.4 to 240 L s$^{-1}$. | Authors | March/ April 2017 | Pre-monsoon |
| 2 | Citizen Scientist Evaluation | Citizen Scientist evaluation of three simple flow measurement methods (i.e. float, salt dilution, and Bernoulli) along "expert" and FlowTracker ADV reference flow measurements at 15 sites within the Kathmandu Valley.  Reference flows ranged from 4.2 to 896 L s$^{-1}$. | Authors for "expert" and reference flows PLUS 10 Citizen Science Flow groups for simple methods | September 2018 | Post-monsoon |

| | | Salt dilution measurements at roughly 150 sites in the 10 perennial watersheds of the Kathmandu Valley.  Float measurements with a small number of sub-sections (e.g. 3 to 5) performed at each site to determine salt quantities. | | | |
|---|---|---|---|---|---|
| 3 | Citizen Scientist Application | | 17 Citizen Science Flow groups (7 from April and 10 from September) | April and September 2018 | Pre and Post Monsoon |

The comparison of measurement methods has been edited, so that only measurements taken within +/- one day of each other are used, and then only if precipitation didn't occur and water levels were the same.

3. Results: Graphical presentation of the results is good and I appreciate that the authors report about statistical significance of their results. For some of the dataset I suggest to used the Spearman Rank Correlation Coefficient as the assumptions for using the Pearson Correlation seem to be not fulfilled. I also suggest to mention that, while statistically significant, some of the relations show relations difficult to interpret (definitely not linear or exponential but complex or clustered). I ask the authors to explain why they think Figure 2 is informative to the reader expect for presenting the measurements. I question whether it is meaningful or informative to correlate the slope of the salt dilution calibration k to latitude or longitude and elevation and would suggest to skip this analysis. Instead I recommend to also show the comparison of discharge estimated by salt dilution and by the Bernoulli method. As mentioned in the method section I would ask the authors to quantitatively prove that they can compare discharge estimates taken during the CS-campaign with doppler radar observed discharge taken +/- one month before/after the campaign or skip that part. Please also see more specific comments directly marked in the pdf (uploaded as supplement).

RESPONSE: Based on comments from Reviewer #1, the scatter plots have been removed.  We have also removed previous Figure 2.  The K correlation analysis (previous Figure 5) has been removed.  Yes, the time difference between CS Flow group and verification measurements was a significant challenge in the initial dataset.  The results from the newly collected datasets allow for a more robust comparison between the results from the simple flow measurement methods (for "experts" and citizen science groups) and the reference (or true) flow.  The remainder of the detailed and helpful comments in the supplemental material have been addressed.  We standardized the evaluation metrics to average absolute error, average error (bias), and standard deviation of error.

4. Discussion: The discussion is short and not very into depth. Parts of it would better fit into the result section. While the background concentration is certainly affected by the geology I have strong doubts that correlating k with latitude and longitude or elevation is meaningful. At least better explain why the authors think these predictors are meaningful and the correlations not spurious. I suggest the authors to check whether their dataset would allow additional analysis e.g., on the difference of the quality of the measurements taken by experts and citizens (see my suggestions in the pdf).

RESPONSE: The discussion section has been revised, and parts have been moved to the results section. We now clearly answer each research question in the respective subsections of the discussion.  The K

correlation analysis has been removed.  The new datasets have been used to evaluate variability in CS Flow group measurements with box plots, as compared to both "expert" measurements using the same method, and reference flows from the FlowTracker ADV.

5.  Summary and future work Is well written in general. However, addressing the outcome of this work in the light of research questions (that I suggest to include in the introduction) would improve this section.

RESPONSE: While the research questions (stated at the end of the introduction in Sect. 1.4) are explicitly addressed in the discussion, the summary now returns to the original research questions as well.

I hope these suggestions are useful to work on a more advanced version of the manuscript!

RESPONSE: Indeed, these comments, along with your detailed supplementary comments have been extremely helpful towards improving the quality of this manuscript!  We deeply appreciate your sincere efforts and investment of time.

Please also note the supplement to this comment:
https://www.hydrol-earth-syst-sci-discuss.net/hess-2018-425/hess-2018-425-RC2-supplement.pdf

RESPONSE: As mentioned earlier, a marked-up version of the manuscript showing all changes, along with specific responses/edits based on your comments provided in the supplement are included in the PDF following these responses to your general comments.

[Figure]

[Figure]

**Citizen science flow - an assessment of citizen science streamflow measurement methods**

Jeffrey C. Davids[1,2], Martine M. Rutten[1], Anusha Pandey[3], Nischal Devkota[3], Wessel David van Oyen[4], Rajaram Prajapati[3], and Nick van de Giesen[1]

[1]Water Management, Civil Engineering and Geosciences, Delft University of Technology, TU Delft Building 23, Stevinweg 1, 2628 CN, Delft, Netherlands
[2]SmartPhones4Water-CA, 3881 Benatar Way, Suite G, Chico, CA, 95928, USA
[3]SmartPhones4Water-Nepal, Damodar Marg, Thusikhel, 44600, Lalitpur, Nepal
[4]Engineering and Applied Sciences, Rotterdam University, Rotterdam, G.J. de Jonghweg 4-6, 3015 GG, Rotterdam, Netherlands

*Correspondence to*: Jeffrey C. Davids (j.c.davids@tudelft.nl)

**Abstract.** Wise management of water resources requires data. Nevertheless, the amount of streamflow data being collected globally continues to decline. Involving citizen scientists to generate hydrologic data can potentially help fill this growing hydrological data gap. Our aim herein was to (1) evaluate three potential citizen science streamflow measurement methods (i.e. float, salt dilution, and Bernoulli run-up), (2) select a preferred approach, and (3) pilot test the selected approach at a larger scale. We performed 20 side-by-side evaluation measurements in headwater catchments of the Kathmandu Valley. We used mid-section measurements from an acoustic Doppler velocimeter as reference flows. Evaluated flows ranged from 0.006 to 0.240 $m^3$ $s^{-1}$. Linear regressions forced through the origin for scatter plots with reference flows had slopes of 1.05, 1.01, and 1.26 with r-squared values of 0.90, 0.98, and 0.61, for float, salt dilution, and Bernoulli run-up methods, respectively. After selecting the salt dilution method as the preferred approach, we performed larger scale pilot testing in a one-week Citizen Science Flow campaign (CS Flow) involving 20 volunteers. Observed flows (n = 145) ranged from 0.0004 to 0.425 $m^3$ $s^{-1}$ and were distributed among the 10 headwater catchments of the Kathmandu Valley. At locations with reference flows available (n = 5), a linear regression forced through the origin between reference flows and CS Flow measurements had a slope of 0.90 with an r-squared value of 0.97. Future work should evaluate the feasibility of applying citizen science salt dilution streamflow measurements to larger regions.

**1 Introduction**

Lord Kelvin, a 19th century Scottish physicist and mathematician, wisely said, "… the first essential step in the direction of learning any subject is to find principles of numerical reckoning and practicable methods for measuring some quality connected with it (Kelvin 1883)." With regards to our natural resources, if we aim to wisely steward them, we must first learn to measure them. While it might sound trivial, collecting and, worse yet, interpreting point measurements of precipitation, evapotranspiration, infiltration, and soil moisture at the catchment scale is fraught with challenges. Indeed, the

[Figure]

Number: 1  Author: reviewer  Subject: Note  Date: 2018-09-14 10:49:54

I think the first paragraph is a bit philosophical. I would focus more on streamflow and introduce to the problem that large parts of the words still have limited number of gauging stations (especially remote and developing countries) and that measuring devices - while still decreasing in costs - have their limitations.

The following paragraph is well suited as introduction!

[revised manuscript text omitted]

"Various" has been removed and the sentence now reads:
"This data gap is perpetuated by a lack of understanding among policy makers and citizens alike regarding the importance of streamflow data, which leads to persistent funding challenges (Kundzewicz 1997; Pearson 1998)."

**Page:2**

Number: 1  Author: reviewer  Subject: Note  Date: 2018-09-14 10:50:51
various is rather vague: List some of the reasons!

Number: 2  Author: reviewer  Subject: Note  Date: 2018-09-14 10:53:02
add more recent papers e.g.
Burt & McDonnell (2015): Whither field hydrology ...WRR, (https://doi.org/10.1002/2014WR016839
)

The Burt and McDonnell reference has been added and now reads:
"This is further compounded by the reality that the hydrological sciences research community has focused much of its efforts in recent decades on advancing modeling techniques, while innovation in methods for generating the data these models depend on has been relegated to a lower priority (Mishra and Coulibaly 2009; Burt and McDonnell 2015), even though these data form the foundation of hydrology (Tetzlaff et al. 2017)."

[Figure]

While the previously referenced studies focus mainly on involving citizen scientists for observing stream levels, we were primarily concerned with the possibility of enabling citizen scientists to take direct measurements of streamflow. Using keyword searches using combinations of "citizen science", "citizen hydrology", "community monitoring", "streamflow monitoring", "streamflow measurements", and "discharge measurements," we could not find any specific work about how

5   citizen scientists, equipped with modern tools like smartphones, could take streamflow measurements directly themselves. Instead, to develop potential citizen science streamflow measurement methods to evaluate further, we turned to the vast body of general knowledge about the collection of streamflow data.

Streamflow measurement techniques suggested in the United States Bureau of Reclamation Water Measurement Manual

10   (USBR 2001) that seemed potentially applicable for citizen scientists included: deflection velocity meters consisting of shaped vanes projecting into the flow along with a method to measure deflection; the slope area method whereby the slope of the water surface in a uniform reach is measured and combined with the Manning formula; and pitot tubes for measuring velocity heads. The float and salt dilution methods described by several authors also seemed applicable (British Standards Institute 1964; Rantz 1982; Fleming and Henkel 2001; Escurra 2004; Moore 2004a, 2004b, and 2005; Herschy 2014).

15   Finally, Wilm and Storey (1944) and Church and Kellerhals (1970) introduced the velocity head rod, or what we later refer to as the Bernoulli run-up method, involving measurement of stream velocity heads with a thin flat plate.

Based on these recommendations, the strengths and limitations discussed in the corresponding literature, and practical considerations about how citizen scientists could implement the different approaches, we selected three approaches for

20   further evaluation: float, salt dilution, and Bernoulli run-up. Our primary aims in this paper were to (1) evaluate these three potential citizen science streamflow measurement methods, (2) select a preferred approach, and (3) pilot test the preferred approach at a larger scale.

**2   Materials and methods**

**2.1   Citizen science streamflow measurement methods evaluated**

25   The procedures for each of the three citizen science streamflow measurement methods evaluated are described in the following sections.

**2.1.1**

The float method is based on the velocity-area principle. Total streamflow (Q) in cubic meters per second (m$^3$ s$^{-1}$)  calculated with Eq. (1):

Eq. (1)   $Q = \sum_{i=1}^{n} C * VF_i * d_i * w_i$

Good tip! This has been edited to read:

"Using keyword searches using combinations of "citizen science", "citizen hydrology", "community monitoring", "streamflow monitoring", "streamflow measurements", "smartphone streamflow measurement", and "discharge measurements," we found that research on using smartphone video processing methods for streamflow measurement has been ongoing for nearly five years (Lüthi et al. 2014; Peña-Haro et al. 2018)."
* * *
**Number: 1  Author: reviewer  Subject: Note  Date: 2018-09-14 11:01:02**

Maybe the key words were not appropriate to find these solution. There are some smartphone apps available that can do that e.g., http://www.photrack.ch/dischargeapp.html
Please delet the statement and widen your search. There are apps available that can be used widely

**Number: 2  Author: reviewer  Subject: Note  Date: 2018-09-14 11:08:32**

I think it would be good to better describe how citizen science can be one possible way to close the data gap but also mention difficulties!

A paragraph has been added to the end of section 1.1 to make these good points.

**Number: 3  Author: reviewer  Subject: Note  Date: 2018-09-14 11:07:32**

the description of the methods could be more detailed

A summary table (Table 1) has now been included for each of the considered simple methods.

**Number: 4  Author: reviewer  Subject: Note  Date: 2018-09-14 11:04:30**

Add cits that hat used this method would add to the introduction)

Citations about these methods have been included.

**Number: 5  Author: reviewer  Subject: Note  Date: 2018-09-14 11:05:15**

.. crossestion area and the ...

The brief description of each method has been moved to Table 1.

**Number: 6  Author: reviewer  Subject: Note  Date: 2018-09-14 11:06:11**

explain the reader how the salt dilution method works in 1/2 or 1 senses like you did for the other methods

This is now included in Table 1.

**Number: 7  Author: reviewer  Subject: Note  Date: 2018-09-14 11:10:24**

You should summarize the pros and cons and limitations of these methods for the reader. This is importent also for you to argue, why you chose the three ones for this study! Otherwise this choce is sunjective.

You could add a table with the pros and cons of each method and citations.

This is now included in Table 1.

**Number: 8  Author: reviewer  Subject: Note  Date: 2018-09-14 17:44:41**

I could imagine to present the following information in the introduction section (and include the Manning-Strickler Method as another method (that you later decided not to use) because the way you present it is a introduction to these methods. In the method section you could then more describe the experimental setup hwo you tested these methods for thier suitbility for CS.

Table 1 is included in the introduction along with all the simple methods considered.

**Number: 9  Author: reviewer  Subject: Note  Date: 2018-09-14 11:24:10**

I think this sentence is not necessary and can be skipped

This sentence has been deleted.

**Number: 10  Author: reviewer  Subject: Replace  Date: 2018-09-14 11:33:59**

Float Method

Revision made.

**Number: 11  Author: reviewer  Subject: Note  Date: 2018-09-14 11:33:44**

Start with a description how the method works in general. You measure cross section (subdeviding it into n-sub-sections) and flow velocity and the apply a coefficients for accounting for friction losses

This is now included.

**Number: 12  Author: reviewer  Subject: Replace  Date: 2018-09-14 11:30:47**

is

Revision made.

[Figure]

ⓒ ①

where C is a unitless coefficient to account for the fact that surface velocity is typically higher than average velocity (typically in the range of 0.66 to 0.80 depending on depth; USBR 2001), $VF_i$ is surface velocity from float in meters per second (m s⁻¹), $d_i$ is depth (m), and $w_i$ is width (m) of each sub-section (i = 1 to n, where n is the number of stations).

5 Surface velocity for each sub-section
 was determined by measuring the amount of time it takes for a floating object to move a certain distance. For floats we used sticks found on site. Sticks are widely available (i.e. easiest for citizen scientists), generally float (except for the densest varieties of wood), and depending on their density are between 40 and 80% submerged, which minimizes wind effects. [2]

10 Float measurements involved the following steps:

1. Selected stream reach with straight and uniform flow
2. Divided cross section into several sub-sections (n, typically between 5 and 20)
3. For each section, measured and recorded
    a. The depth in the middle of the sub-section
    b. The width of the sub-section
    c. The time it takes a floating object to move a known distance downstream (typically 1 or 2 m) in the middle of the sub-section
4. Solved for streamflow (Q) with Eq. (1)

**2.1.2 Salt dilution**

There are two basic types of salt dilution flow measurements: slug (previously known as instantaneous) and continuous rate (Moore 2004a). Salt dilution measurements are based on the principle of the conservation of mass. In the case of the slug method, a single known volume of high concentration salt solution is introduced to a stream and the electrical conductivity (EC) is measured over time at a location sufficiently downstream to allow good mixing (Moore 2005). In contrast, [3]

25 continuous rate salt dilution method involves introducing a known flow rate of salt solution into a stream (Moore 2004b). Slug method salt dilution measurements are broadly applicable in streams with flows up to 10 m³ s⁻¹ with steep gradients and low background EC levels (Moore 2005). For the sake of citizen scientist repeatability, we chose to only investigate the slug [4] method, because of the added complexity of measuring the flow rate of the salt solution for the continuous rate method.

30 Streamflow (Q; m³ s⁻¹) was solved for using Eq. (2) (Rantz 1982; Moore 2005):

Eq. (2) $\quad Q = \dfrac{V}{k \sum_{i=1}^{n} (EC(t) - EC_{BG}) \Delta t}$ [5]

Number: 1  Author: reviewer  Subject: Replace  Date: 2018-09-14 11:30:40

is

Revision made.

Number: 2  Author: reviewer  Subject: Note  Date: 2018-09-14 11:32:41

Mention the difference between surface flow velocity and velocity at dept and flow velocity neat the banks and the middle of the stream. Mention also the difficulty that the float can get stuck.

Revisions made: "An additional challenge with floats is that they can get stuck in eddies, pools, or overhanging vegetation."

Number: 3  Author: reviewer  Subject: Note  Date: 2018-09-14 11:35:10

mention also how EC values are than translated into dischage!

Number: 4  Author: reviewer  Subject: Note  Date: 2018-09-14 11:35:52

Mention limitations of the salt dilution method

Number: 5  Author: reviewer  Subject: Note  Date: 2018-09-14 11:43:00

It is more common to express the equation in terms of concentrations mot EC!

Revision 3 made: "An approximation of the integral of EC as a function of time is combined with the volume of tracer and a calibration constant (Eq. 2) to determine discharge."

Revision 4 made: "Some limitations of the salt dilution method include: (1) inadequate vertical and horizontal mixing of the tracer in the stream, (2) trapping of the tracer in slow moving pools of the stream, and (3) incomplete dilution of salt within the stream water prior to injection. The first two limitations can be addressed with proper site selection (i.e. well mixed reach with little slow-moving bank storage), while incomplete dilution can be avoided by proper training of the personnel performing the measurement."

Revision 5: since we are including k in the equation per Moore 2005, we thought it would be more consistent to show the equation with EC.

ⓒ ⓘ

where V is the total volume of tracer introduced into the stream ($m^3$), k is the calibration constant in centimeters per microsiemens (cm $\mu S^{-1}$), n is the number of measurements taken during the breakthrough curve (unitless), EC(t) is the EC at time t ($\mu S\ cm^{-1}$), $EC_{BG}$ is the background EC ($\mu S\ cm^{-1}$), and $\Delta t$ is the change in time between EC measurements (s).

We performed the following steps when making a salt dilution measurement:

1. Sele
stream reach with turbulence to facilitate vertical and horizontal mixing
2. Determined upstream point for introducing the salt solution and a downstream point for measuring EC
   a. A rule of thumb in the literature is to separate these locations roughly 25 stream widths apart (Day 1977; Butterworth et al. 2000; Moore 2005)
3. Estimated flow rate visually by estimated width, average depth, and average velocity
4. Prepared salt solution based on the following guidelines (adapted from Moore 2005)
   a. 10000 ml of stream water for every 1 $m^3\ s^{-1}$ of estimated streamflow
   b. 1667 g of salt for every 1 $m^3\ s^{-1}$ of estimated streamflow
   c. Thoroughly mix salt and water until all salt is dissolved
   d. Following these guidelines ensured a homogenous salt solution with 1 to 6 salt to water ratio by mass
5.  (Moore 2004b) to determine calibration constant (k) relating changes in EC values in micro Siemens per centimeter ($\mu S\ cm^{-1}$) in the stream to relative concentration of introduced salt solution (RC)
   a. Made diluted secondary solution by mixing 500 ml of stream water and 5 ml of salt solution
   b. Measured background stream water EC ($EC_{BG}$)
   c. Added known volume (typically 1 or 2 milliliters (ml)) of secondary solution to 500 ml of stream water in dilution cylinder
   d. Measured new dilution cylinder EC
   e. Repeated steps 5.c and 5.d until the full range of expected EC values were observed
   f. Calculated RC for each measurement point
   g. Plotted EC on the horizontal axis and RC on the vertical axis
   h. Performed linear regression
   i. Obtained k from the slope of the linear regression
6. Dumped salt solution at upstream location
7. Measured EC at downstream location during salinity breakthrough until EC returns to $EC_{BG}$
   a. Recorded a video of the EC meter screen at the downstream location and later digitized the values using the time from the video and the EC values from the meter
8. Solved for streamflow (Q) with Eq. (2)

**Number: 1  Author: reviewer  Subject: Note  Date: 2018-09-14 11:43:54**

I would use present tense for the method description

Revision made to state the steps in the present tense.

**Number: 2  Author: reviewer  Subject: Replace  Date: 2018-09-14 11:49:16**

Establish the calibration curve relating EC values to actual salt concentrations (Moore 2004b) ...

Revision made: "Establish the calibration curve relating EC values to actual salt concentrations (Moore 2004b) to determine calibration constant (k) relating changes in EC values in micro Siemens per centimeter ($\mu$S cm$^{-1}$) in the stream to relative concentration of introduced salt solution (RC)"

[Figure]

**2.1.3 Bernoulli run-up**
[Figure]

Similar to the float method, Bernoulli run-up (or Bernoulli) is based on the velocity-area principle. Total streamflow (Q; m³ s⁻¹) was calculated with Eq. (3):

5    Eq. (3)    $Q = \sum_{i=1}^{n} VB_i * d1_i$

where $VB_i$ is velocity from Bernoulli run-up (m s⁻¹), $d1_i$ is depth (m), and $w_i$ is width (m) of each sub-section (i = 1 to n). Area for each sub-section is the product of the width and the depth in the middle of each sub-section. Velocity for each sub-section ($VB_i$) was determined by measuring the "run-up" or change in water level on a thin meter stick from when the stick

10   was inserted parallel and then perpendicular to the direction of flow. The basic principle is that "run-up" on a flat plate inserted perpendicular to flow is proportional to velocity based on the solution to Bernoulli's equation. Velocity ($VB_i$; m s⁻¹) was calculated from Bernoulli's principle with Eq. (4):

Eq. (4)    $VB_i = \sqrt{2g * (d2_i - d1_i)}$

where g is the gravitational constant (m s⁻²) and $d2_i$ and $d1_i$ are the water depths (m) when the flat plate was perpendicular and parallel to the direction of flow, respectively.

Bernoulli run-up measurements involved the following steps:

1. Selected constricted stream with elevated velocity to increase the difference between d1i and d2i
2. Divided cross section into several sub-sections (n, typically between 5 and 20)
3. For each section, measured and recorded
   a. The depth with a flat plate held perpendicular to flow (d2i or the "Run-up" depth)
25      b. The depth with a flat plate held parallel to flow (d1i or the actual water depth)
   c. The width of the sub-section
4. Solved for streamflow (Q) with Eq. (3) and Eq. (4)

**2.2   Reference flow**
[Figure]

30   To evaluate the different citizen science flow measurement methods, a reference (or actual) flow for each site was needed. We used a SonTek FlowTracker acoustic Doppler velocimeter (ADV) to determine reference flows. The United States Geological Survey (USGS) mid-section method was used, following guidelines from USGS Water Supply Paper 2175

**Number: 1  Author: reviewer  Subject: Note  Date: 2018-09-14 12:23:15**

I am not familiar with this method and also the USBR (2001) includes two short paragraphs about this method. Do you use a standardized device for that? What size is the plat you are submerging into the water.

In any case the method is an alternative for estimating/measuring the flow velocity, the rest is the area-flow-velocity method

As far as we know, there is no standardized device for this. The dimensions of the meter stick that we have are now included: "(dimensions: 1 meter long, by 34 mm wide, by 1.5 mm thick)"

**Number: 2  Author: reviewer  Subject: Note  Date: 2018-09-14 12:13:06**

please give citation

USBR 2001 citation included.

**Number: 3  Author: reviewer  Subject: Note  Date: 2018-09-14 11:51:20**

why d1i and not just di?

$d_{1i}$ is defined in Eq. 4 as the water depth with the flat plate is parallel to the direction of flow.

**Number: 4  Author: reviewer  Subject: Note  Date: 2018-09-14 12:32:15**

First you introduce to the various methods used in the sudy! That's fine. Now I would suggest to explain the experimental setup, how you tested the applicability of the methods. (paragrahs below would fit here well)

IMPORTANT: Early in your method section (or as a separate section) describe what type of streams we are looking at in your study. Width, depths, closssection, steepnes, roughness, lamnar, turbulent, sediment transport, etc. This is important to assess whether these methods are appropriate to measure discharge or not!

Section 2.2 provides the details of the experiment and Section 2.1.1 provides information on the types of streams studied.

**Number: 5  Author: reviewer  Subject: Note  Date: 2018-09-14 12:29:13**

I assume this device results ina surface flow velocity. What about your cross section? How detailed did you determine it?

Response 5: Rantz 1982 describes the USGS mid-section method in detail. The FlowTracker ADV measures water velocity within a sample volume depending on the depth that the meter is set in the water column. We have added the following language to specify the range of sub-sections used. Full details of the reference flow measurements are included in the supplementary materials.
"Stream depths were shallow enough that a single vertical 0.6 depth velocity measurement (i.e. 40% up from the channel bottom) was used to measure average velocity for each sub-section. Depending on the total width of the channel, the number of sub-sections ranged from 11 to 30."

[Figure]

(cc) (i)

(Rantz 1982), along with instrument specific recommendations from SonTek's FlowTracker manual (SonTek 2009). The FlowTracker ADV has a stated velocity measurement accuracy of within one percent (SonTek 2009). Flow measurement errors, calculated with an International Standards Organization (ISO) approach built into the FlowTracker software, are typically in the range of 3 to 10 %. Reference flow errors in this study are discuss in Section 4.5. A compilation of the

5 measurement reports generated by the FlowTracker ADV are included as supplementary material.

**2.3 Flow measurement method evaluation and analysis**

We first summarized flow measurement method evaluation results in map and tabular form (Fig. 1; Table 1). Measurement ID can be used to link data between the map and table. We used scatter plots to compare reference flow (x-axis) to the three

10 flow measurement methods evaluated (y-axis) to visualize and interpret results from each method. We fitted these points with a linear regression forced through the origin. To understand relative (normalized) errors, we calculated percent differences in relation to reference flow for each method. To better understand possible explanations for observed variability in our results, we performed a correlation analysis. For each method, we performed a Pearson's r correlation analysis (Lee Rodgers and Nicewander 1988) between the absolute value of percent difference in flow and (1) reference flow, (2) average

15 velocity, (3) $EC_{BG}$, and (4) slope. Slope values were developed using elevations from the Google Earth Digital Elevation Model (DEM) obtained
 along the centreline of the stream alignment both 100 meters upstream and downstream of each measurement point (retrieved July 2nd, 2018). While using DEM data for slope calculations is clearly inferior to performing topographic surveys in the field, this was not possible due to lack of equipment and time; therefore, these slope data are the best available numbers.

**2.4 Salt dilution calibration coefficient (k) analysis**

Arguably, one of the most complicated portions of a salt dilution measurement is performing the dilution test to determine the calibration coefficient k (Moore 2004b). To determine if the dilution test needs to be repeated for each citizen science measurement, we analyzed all k values determined during this study. In addition to the mean, range, and standard deviation,

25 we performed a Pearson's r correlation analysis (Lee Rodgers and Nicewander 1988) to see if k showed statistically significant trends with latitude, longitude, elevation, and $EC_{BG}$.

**2.5 2.5. CS Flow campaign - pilot testing of salt dilution method**

Based on the initial results from this study, S4W-Nepal developed an Open Data Kit (ODK; Anokwa et al. 2009) form for

30 citizen scientists to perform salt dilution measurements. The general workflow was (1) selected an appropriate measurement reach with good mixing and minimal bank storage, (2) performed a simplified float measurement (i.e. only a 3 or 4 depths

**Number: 1  Author: reviewer  Subject: Note  Date: 2018-09-14 12:35:54**

Please mention the spatial resolution of the DEM. How many cells where considered in the 100 up and 100 downstream section to calculate slope?

Auto-level surveys were performed at all 15 Phase 2 sites which are described in Section 2.2.2 now.  DEM no longer used.

**Number: 2  Author: reviewer  Subject: Note  Date: 2018-09-14 12:38:19**

Do you mean that this calibration is most difficult for citizens to do acuratly?

Why are yo then trying to correlate it with latitude and longitude, elevation and ECBG? Please explain better!

Yes, our experience was that k is difficult to measure in the field, but factors affecting k aren't the focus of this study, so it has been removed.

**Number: 3  Author: reviewer  Subject: Note  Date: 2018-09-14 12:39:13**

Do not use abbrevations in a headline and please defien CS before!

This is now first defined in Section 2.2.2 before using the short form CS.

**Number: 4  Author: reviewer  Subject: Note  Date: 2018-09-14 12:43:35**

This is about the experimental setup and would fit better at an earlier spot in the method section.

Material about developing the ODK forms has been moved to Section 2.2.2 of methods.

**Number: 5  Author: reviewer  Subject: Note  Date: 2018-09-14 13:21:35**

What is S4W-Nepal! Please introduce earlier so the reader knows it! I assume iit is a certain project name!

Indeed, S4W-Nepal is a citizen science project.  Section 1.5 has been added to the introduction to provide additional context.

**Number: 6  Author: reviewer  Subject: Note  Date: 2018-09-14 12:41:57**

You have describe the method beforehand. If you procedure is the same as applied by Anokwa et al., 2009 do not repeat it but refer to your description.

The Anokwa et al. 2009 reference is a generic reference describing Open Data Kit (ODK) in general.  We have taken ODK and made a specific form for our data collection processes for Phases 2 and 3.  This is now clarified in Section 2.2.2.

[Figure]

and velocities), (3) used the float flow estimate and $EC_{BG}$ to provide citizen scientists recommended salt/water dose, (4) used pre-weight packets of salt (e.g. 10 g, 20 g, 50 g, 100 g, etc.) to prepare tracer solution, (5) added tracer solution to stream and recorded video of EC breakthrough curve, (6) submitted form to ODK Aggregate server, (6) digitized breakthrough curve (i.e. time and EC) in shared Google Sheet salt dilution flow calculator.

During S4W-Nepal's Citizen Science Flow (CS Flow) campaign (15th to 21st of April 2018; Fig. 6), student volunteers from Khwopa College of Engineering were recruited, trained, divided into groups by sub-watershed, and sent to the field to perform salt dilution flow measurements. In the second week, student volunteers used a salt dilution Google Sheet flow calculator to digitize collected measurement data and compute flow (see supplementary material for Excel version).

10 Students analyzed data (third week) and finally presented oral and written summaries of their quality-controlled results. S4W-Nepal currently leverages the enthusiasm and schedule breaks in the academic calendar of young researchers to perform campaigns to improve our pre and post monsoon understanding of stone spouts (Nepali: dhunge dhara), land use, and now streamflow.

15 To analyze the generated streamflow data, we developed a scatter plot between flow estimates from the simplified float method (used to calculated salt dosage) and the salt dilution flow results. At locations were S4W-Nepal takes regular FlowTracker measurements, we compared the most recent S4W-Nepal observation(s) to CS Flow salt dilution measurements. Because salt dilution measurements were performed during the pre-monsoon period when precipitation is minimal, hydrographs are relatively steady with gradual recession over time as the South Asian Monsoon approaches.

20 Therefore, we did not expect differences in time (e.g. plus or minus one month roughly) between the two measurements to greatly impact the resulting comparisons.

**3 Results**

**3.1 Summary of evaluation measurements**

We performed sets of evaluation measurements at 20 sites within the Kathmandu Valley, Nepal (Fig. 1) at elevations ranging

25 from 1313 to 1905 meters above mean sea level. Flows evaluated ranged from 0.006 to 0.240 $m^3$ $s^{-1}$ (Table 1; sorted in ascending order by reference flow). Percent differences averaged 7.9, 8.2, and 25.7 % and standard deviations (std dev) of 29.1, 17.2, and 61.9 % were observed for float, salt dilution, and Bernoulli methods, respectively (Table 1). Field notes from Bernoulli flow measurements for two measurements (Msmt IDs 17041903 and 17031102) were destroyed by water damage, so flow and percent difference data were not available. Plots of EC and change in EC as a function of time for all 20 salt

30 dilution measurements are shown in Fig. 2. Additional data for evaluation measurements is included as supplementary material.

**Number: 1  Author: reviewer  Subject: Note  Date: 2018-09-14 13:22:14**

You need to indroduce to these watershed. How many, were, how big what average flow during the campaign etc. Refer to Fig. 1!!

These 10 sub-watersheds of the Kathmandu Valley are labeled and shown on Figure 6, along with locations of measurements.

**Number: 2  Author: reviewer  Subject: Note  Date: 2018-09-14 12:46:05**

how long were they trained, was it a theoretical explanation or a hands-on training actually applying the methods themselves under supervision?

Additional details regarding training (theoretical and in the field) have been added to Sections 2.2.2 and 2.2.3.

**Number: 3  Author: reviewer  Subject: Note  Date: 2018-09-14 12:47:06**

Start with this sentence!

This sentence has been moved to Section 1.5 in the introduction where details about S4W-Nepal are provided.

**Number: 4  Author: reviewer  Subject: Note  Date: 2018-09-14 12:48:32**

Explain more: Where are these stations. Are these the same as the students performed the salt dilution tests. Mark these stations in a map!

Because of the difference in time between CS Flow group and verification measurements, this material has been removed.

**Number: 5  Author: reviewer  Subject: Note  Date: 2018-09-14 15:54:23**

One month (!!) is really long! You need to prove your assumption with statistical data. E.g. mean daily runoff at the gauging stations for each month in the pre-monsoon period and a statistical test, showing that the mean daily discharge is not statistically significantly different!

If you do not have the data to prove this a) consider to skip this part or b) take a dataset of a different gauging station of the region (e.g. operated by a local authority or so)  and prove it based on this data.

We agree that this was a primary challenge of our initial dataset, and trust that the new data addresses these concerns.

**Number: 6  Author: reviewer  Subject: Insert Text  Date: 2018-09-14 13:25:11**

... a xx km2 Catchment xx km NW of yyy-city!

A more detailed description of the Kathmandu Valley has been provided in the first 2 sentences of Section 3.1.

**Number: 7  Author: reviewer  Subject: Insert Text  Date: 2018-09-14 13:26:02**

Reference flow ...

Edit made accordingly.

**Number: 8  Author: reviewer  Subject: Note  Date: 2018-09-14 13:29:54**

percent difference to what? I think you men mean percent difference between the float method, the salt dilution method and the Bernoulli method relative the the doppler measurements?

The following language has been added: "Absolute errors with respect to reference flows averaged..."

**Number: 9  Author: reviewer  Subject: Note  Date: 2018-09-14 13:30:38**

exclude them form you data set and from the map

Since only data for the Bernoulli method were lost, we didn't think it was necessary to remove reference flows, float, and salt dilution data.

[Figure]

[Figure]

**Figure 1: Map figure showing the topography (green to tan to white color gradation) of the Kathmandu Valley from a Shuttle Research Telemetry Mission (SRTM 2000) Digital Elevation Model (DEM), resulting stream network (Davids et al. 2018), and locations of flow measurements (msmts). Measurement points are labelled with measurement ID (msmt_id). Names of the ten historically perennial tributaries are shown.**
* * *
📝 Number: 1  Author: reviewer  Subject: Note  Date: 2018-09-14 13:31:46

first order streams ?

The legend on Figures 2, 3, and 6 has been simplified.
* * *
📝 Number: 2  Author: reviewer  Subject: Note  Date: 2018-09-14 13:37:11

the labels partly cover the points.

Even if you (internally used these long digit labels, consider to re-lable them by 1 to 20.

Labels have been edited in favor of 1 to 20 as suggested for Phase 1.  Overlapping labels have been corrected in Figure 2.
* * *
📝 Number: 3  Author: reviewer  Subject: Note  Date: 2018-09-14 13:33:16

do not split the legend for DEM
do not use uneven classbreaks for you r colours!
Msmts is explained in the text but not intuitive.

The legend has been edit to have even DEM breaks show without a split, and "Phase X Site" has been used in favor of "Msmts."

[Figure]

[Figure]

**Table 1: Tabular summary of measurement comparison data. Records sorted in ascending order by reference flow (Q Reference). Latitude and longitude in reference to the WGS84 datum. All flow values shown are shown in $m^3$ $s^{-1}$ rounded to the thousandth place. Percent differences calculated using Q Reference (FlowTracker) as the actual flow. Data summarized at the bottom with average, minimum (min), maximum (max), and standard deviation (std dev). Note that measurement ID (Msmt ID) is comprised of two digits for year, month, date, and measurement number starting at 01 each day.**

| Msmt ID | Latitude | Longitude | Elev-ation (m) | Q Reference (m3 s-1) | Q Float (m3 s-1) | Q Salt (m3 s-1) | Q Bernoulli (m3 s-1) | % Diff-erence Float | % Diff-erence Salt | % Diff-erence Bernoulli |
|---|---|---|---|---|---|---|---|---|---|---|
| 17030202 | 27.78065 | 85.42426 | 1649 | 0.006 | 0.007 | 0.006 | 0.009 | 15.6 | -12.5 | 37.5 |
| 17041802 | 27.78158 | 85.42385 | 1659 | 0.007 | 0.008 | 0.007 | 0.010 | 15.9 | 7.2 | 44.9 |
| 17031001 | 27.79649 | 85.42177 | 1905 | 0.011 | 0.008 | 0.012 | 0.009 | -28.4 | 11 | -19.3 |
| 17042401 | 27.70026 | 85.22077 | 1406 | 0.017 | 0.019 | 0.019 | 0.018 | 11.2 | 11.2 | 4.7 |
| 17032201 | 27.57487 | 85.31314 | 1482 | 0.018 | 0.020 | 0.024 | 0.019 | 12.4 | 37.3 | 5.1 |
| 17041901 | 27.77164 | 85.42657 | 1609 | 0.019 | 0.028 | 0.027 | 0.022 | 48.4 | 47.3 | 16.7 |
| 17033001 | 27.78691 | 85.32589 | 1364 | 0.021 | 0.026 | 0.030 | 0.048 | 27.2 | 43.2 | 132 |
| 17042402 | 27.69620 | 85.23142 | 1382 | 0.023 | 0.010 | 0.025 | 0.006 | -59.2 | 5.2 | -73 |
| 17041903 | 27.75406 | 85.42170 | 1355 | 0.034 | 0.051 | 0.033 | | 51.6 | -0.9 | |
| 17041902 | 27.77154 | 85.42680 | 1609 | 0.041 | 0.041 | 0.047 | 0.063 | 0.5 | 14.6 | 53.2 |
| 17030101 | 27.78483 | 85.44480 | 1877 | 0.104 | 0.111 | 0.088 | 0.102 | 6.9 | -15.9 | -2.6 |
| 17032203 | 27.57542 | 85.31268 | 1477 | 0.111 | 0.106 | 0.120 | 0.116 | -4.3 | 8 | 5.1 |
| 17032202 | 27.57410 | 85.31277 | 1481 | 0.117 | 0.081 | 0.126 | 0.102 | -30.7 | 8 | -13.2 |
| 17033002 | 27.78627 | 85.32583 | 1356 | 0.153 | 0.208 | 0.144 | 0.470 | 36.5 | -5.6 | 207.9 |
| 17030201 | 27.78156 | 85.42383 | 1659 | 0.155 | 0.248 | 0.176 | 0.161 | 59.3 | 13.1 | 3.5 |
| 17041803 | 27.78168 | 85.42373 | 1663 | 0.156 | 0.140 | 0.142 | 0.210 | -10.4 | -8.9 | 34.4 |
| 17031002 | 27.77932 | 85.42496 | 1653 | 0.159 | 0.183 | 0.155 | 0.228 | 15.2 | -2.8 | 43.4 |
| 17031101 | 27.78505 | 85.44473 | 1877 | 0.208 | 0.221 | 0.207 | 0.150 | 6.5 | -0.6 | -27.8 |
| 17031102 | 27.77514 | 85.43867 | 1806 | 0.230 | 0.188 | 0.219 | | -18 | -4.8 | |
| 17042002 | 27.71106 | 85.35432 | 1313 | 0.240 | 0.246 | 0.264 | 0.264 | 2.7 | 10.2 | 10.1 |
| | | average -> | 1579 | 0.092 | 0.098 | 0.094 | 0.112 | 7.9 | 8.2 | 25.7 |
| | | min -> | 1313 | 0.006 | 0.007 | 0.006 | 0.006 | -59.2 | -15.9 | -73.0 |
| | | max -> | 1905 | 0.240 | 0.248 | 0.264 | 0.470 | 59.3 | 47.3 | 207.9 |
| | | std dev -> | 190 | 0.081 | 0.089 | 0.081 | 0.122 | 29.1 | 17.2 | 61.9 |

**Number: 1   Author: reviewer   Subject: Note   Date: 2018-09-14 13:42:16**

Empty cells indicate lost data records

This language has been included.

**Number: 2   Author: reviewer   Subject: Note   Date: 2018-09-14 13:40:12**

... but 01, 02 03 are not replicates at the same location! Make that clear!

Consider to re-label the msmts from 1 to 20 and give the date in a separate colum.

The sites in this table (now Table 3) have been relabeled 1 to 20 with a new column for date.

[Figure]

[Figure]

[Figure]

**Figure 2: Plots of EC (µS cm⁻¹; blue trace) and change in EC (µS cm⁻¹; green trace) as a function of time (s) for the 20 salt dilution evaluation measurements. Measurement ID (Msmt ID; Table 1) shown at the top right of each subplot (i.e. a through t).**

**3.2 Flow and calibration coefficient (k) results**

**3.2.1 Flow scatter plots**

Scatter plots between reference and observed flows with linear regressions forced through the origin had slopes of 1.05, 1.01, and 1.26 for float, salt dilution, and Bernoulli methods, respectively (Fig. 3). A slope of one represents zero systematic bias, whereas values over one represent positive bias, and values less than one represent negative bias. Therefore, for all the methods evaluated we observed different degrees of positive bias. $R$-squared values were 0.90, 0.98, and 0.61 for float, salt dilution, and Bernoulli methods, respectively (Fig. 3). R-squared values represent the goodness of fit between the regression and the observed data; values closer to one represent a better fit. This can also be seen by the observations for salt dilution plotting closest to the regression line, whereas float and Bernoulli points in general plot farther away from the regression line.

**Number: 1  Author: reviewer  Subject: Note  Date: 2018-09-14 13:46:16**

What was your intention to show this rather big Figure?

The intention was to show the typical nature of the breakthrough curves, but we have included this instead as a supplement.

**Number: 2  Author: reviewer  Subject: Note  Date: 2018-09-14 13:49:30**

better say: "an over- or under estimation of the estimated discharge relative to the reference discharge.

We have included this instead as supplementary material.

**Number: 3  Author: reviewer  Subject: Replace  Date: 2018-09-14 13:48:26**

>1

This section has been removed.

**Number: 4  Author: reviewer  Subject: Note  Date: 2018-09-14 13:52:51**

Bias would be systematic behaviour of all datapoints shifted up or down. Your regression line indicates a very slight overestimation of the estimated discharge to the reference.

This section has been removed.

[Figure]

[Figure]

[Figure]

**Figure 3: Scatter plots between reference flow and observed flow for (a) float, (b) salt dilution, and (c) Bernoulli. Note there is one Bernoulli measurement point (17033002) that is outside of the plot space shown (fixed from 0.0 to 0.3 for consistency). Linear regressions and r-squared values shown on the bottom right of each sub-plot.**

**3.2.2    Flow error correlations**

We found statistically significant correlations (n = 20, p = 0.1 … 0.378) between the absolute value of percent error for float and average velocity (Avg Vel; sub-plot 2; r = -0.48) and salt dilution percent error and reference flow (Q Ref; sub-plot 5; r = -0. … (Fig. 4). In both cases, the correlation coefficient was negative, indicating an inverse relationship between the variables. No statistically significant correlations were observed between the remaining pairs of variables.

**Number: 1  Author: reviewer  Subject: Note  Date: 2018-09-14 14:44:28**

Good you tested the significance! But I suggest to also mention that the functional relation between the valiabels is not clear (not linear, not exponetial).

This section has been removed.

**Number: 2  Author: reviewer  Subject: Note  Date: 2018-09-14 13:55:17**

Typically alpah would be chosen to 0.05. Please argue in the text why you use 0.1

This section has been removed.

**Number: 3  Author: reviewer  Subject: Note  Date: 2018-09-14 14:46:21**

Attention: Use the Spearman Rank correlation and approbriate non-parametric statistical test for your type of data. The assumptions for Pearson Correlation are not fulfilled

This section has been removed.

**Number: 4  Author: reviewer  Subject: Note  Date: 2018-09-14 13:55:49**

also give p value

This section has been removed.

**Number: 5  Author: reviewer  Subject: Note  Date: 2018-09-14 13:55:55**

also give p value

This section has been removed.

[Figure]

[Figure]

**Figure 4: Scatter plots between reference flow (Q Ref; m³ s⁻¹), average water velocity (Avg Vel; m s⁻¹), slope, and background EC (EC BG; µS cm⁻¹) and absolute value (Abs) of percent errors for float, salt dilution, and Bernoulli. Pearson r values shown on the upper right of each subplot (i.e. a through l).**

**3.2.3    Salt dilution calibration coefficient (k) results**

The mean calibration coefficient (k) from measurements performed in the field was $2.81 \times 10^{-6} \pm 2.66 \times 10^{-7}$ (95 % confidence interval; n = 10, min = $2.57 \times 10^{-6}$, max = $3.05 \times 10^{-6}$, std dev = $1.33 \times 10^{-7}$). We used mean k to compute salt dilution flows for the remaining 10 measurements. We found statistically significant correlations (n = 10, p = 0.1, r ≥ 0.549)

10  between the calibration coefficient (k) and Longitude (r = 0.60) and Elevation (r = 0.61; Fig. 5). In both cases, the correlation coefficient was positive, indicating a direct relationship between the variables. No statistically significant correlations were observed between the remaining pairs of variables.

Number: 1  Author: reviewer  Subject: Note  Date: 2018-09-14 13:56:39
please also show p values for the relations
This section has been removed.

Number: 2  Author: reviewer  Subject: Note  Date: 2018-09-14 14:42:41
For this type of dataset you need to used the Spearman Rank correlation coefficient and the associated non-parametric statistical test!
This section has been removed.

Number: 3  Author: reviewer  Subject: Insert Text  Date: 2018-09-14 16:41:11
 th the reference measurements taken by the doppler radar.
This section has been removed.

Number: 4  Author: reviewer  Subject: Note  Date: 2018-09-14 14:51:15
what are "remaining measuremetns"

Stricktly k is site and time specific and needs to be determined for each site and at least once a day! k is highly sensitive for the outcome of the discharge. Using an average value is not advisable!

If you were forced to do so, clearly state the reasons in the text and indicate the affected values. Also discuss the error you introduced by this procedure.

Number: 5  Author: reviewer  Subject: Note  Date: 2018-09-14 14:51:37
see my comment on alpha above
This section has been removed.

Number: 6  Author: reviewer  Subject: Note  Date: 2018-09-14 14:55:05
please better describe and argeu why k should be a function of latitude and longitude or elevation. I do not understand the physical based rational. Make clear why you think this is no spurious correlation.

This section has been removed.

Response 4: Because the idea for the salt dilution measurements ultimately is to have citizen scientists take them, we think the processes of determining k needs to be simplified. That is why we used an average k values for all phases of the analysis. The following paragraph has been added to explain the errors introduced by using an average k. Based on Figure 4, the errors introduced by using an average k are much lower than total errors:
"Based on the 10 measured k values in Phase 1, using an average k for all salt dilution measurements caused the largest percent difference for site 7 (8.6 % increase) followed by site 19 (7.6 % decrease). For Phase 2, using average k values for all salt dilution measurements caused the largest percent difference for site D6 (13.7 % decrease) followed by site D3 (12.6 % increase). Because of the sensitivity of salt dilution measurements to k (Eq. 2), additional work should focus on improving understanding of the variables affecting k. Specifically, spatial and temporal variability in k due to changes in stream water chemistry should be investigated prior to applying the salt dilution methodology described in this paper in other areas."

[Figure]

(cc) BY

[Figure]

**Figure 5: Scatter plots between (a) Latitude in degrees, (b) Longitude in degrees, (c) Elevation in meters above mean sea level (m), and (d) background EC in µS cm⁻¹ and the salt dilution calibration coefficient (k).   Pearson's r values shown on the upper right of each sub-plot.**

**3.3    3.3. CS Flow campaign results**

From the 15th to the 21st of April 2018, 20 students from Khwopa College of Engineering in Bhaktapur, Nepal joined S4W-Nepal's CS Flow campaign.   After four hours of training (i.e. two hours classroom and two hours in the field), the student volunteers performed 14 salt dilution streamflow measurements in the 10 sub-watsheds of the Kathmandu Valley (Fig. 6).

10    Observed flows ranged from 0.0004 to 0.425 m³ s⁻¹ (a summary of the measurement data is included as supplementary material).

**Number: 1  Author: reviewer  Subject: Note  Date: 2018-09-14 14:57:56**

This is good information! Copy it to the method section where you describe the experimental setup!

This has been moved to Section 2.2.3 of the methods.

**Number: 2  Author: reviewer  Subject: Note  Date: 2018-09-14 15:00:00**

It is nit fully crear: The data you presented in Tabl. 1 are done by the authors?

Table 2 and Section 2.2 now make it clear who performed what measurements.  In this case, the Phase 3 measurements were performed by CS Flow groups comprised on student volunteers.

[Figure]

**Figure 6: CS Flow Campaign measurement locations (n = 145) within the Kathmandu Valley. Circular symbol colors are graduated by observed flow rate, categorized by quartile. Larger flows (i.e. darker symbols) were observed on the mainstems (i.e. wider blue lines) of the 10 tributaries of the Bagmati River in the Kathmandu Valley.**

Scatter plots between flow estimates from the simplified float method (used to calculated salt dosage) and the salt dilution flow results show that systematic differences increase as the observed flow rate decreases (Fig. 7).

**Number: 1  Author: reviewer  Subject: Note  Date: 2018-09-14 15:01:53**

consider using even class breaks to GIS pre-determined ones.

Histograms (Figures 6.c and 6.d are now used to show flows instead of gradation on the QGIS maps.

**Number: 2  Author: reviewer  Subject: Note  Date: 2018-09-14 15:02:18**

Good that you state the categorization method!

Thanks!

**Number: 3  Author: reviewer  Subject: Note  Date: 2018-09-14 15:39:51**

to better distinguish between "observed flow" measured with the doppler radar (dataset described before) I would call this here differently. You compare two different estimation methods with each other.

This section has been removed.

[Figure]

Figure 7: Scatter plots between salt dilution measurements on the horizontal axis and simplified float estimates on the vertical axis. All data are shown on subplot (a), flows below 0.05 m³ s⁻¹ are included on subplot (b), and flows below 0.01 m³ s⁻¹ are shown on subplot (c). A linear fit with corresponding r-squared values are shown. Note that the vertical axis scales for subplots (b) and (c) are fixed at three times the horizontal axis scale to ensure that all data are visible.

We identified five locations where S4W-Nepal had performed FlowTracker measurements that could be used as reference flows within roughly one month (plus or minus) of the CS Flow salt dilution measurements (Table 2). Comparable flows ranged from 0.012 and 0.111 m³ s⁻¹. The average error between CS Flow salt dilution and S4W-Nepal FlowTracker measurements was -6.3 %, with a standard deviation of 11.5 %. Linear regression forced through the origin between reference flows and CS Flow measurements had a slope of 0.90 with an r-squared value of 0.97.

**Number: 1  Author: reviewer  Subject: Note  Date: 2018-09-14 16:18:18**

here you call it simplified float flow but in the method you call it float flow. Please be consistent.

The difference between float and "simplified" float is the number of stations, and this is clarified in Section 2.2.3 of the methods.

**Number: 2  Author: reviewer  Subject: Note  Date: 2018-09-14 15:34:47**

please include p-value for all three plots

This figure has been removed.

**Number: 3  Author: reviewer  Subject: Note  Date: 2018-09-14 15:35:25**

what is meant with "fixed at three times"?

This figure has been removed.

**Number: 4  Author: reviewer  Subject: Note  Date: 2018-09-14 15:45:28**

see my comment in the method section on the validity of this comparison with data +/- one months before/after the CS-campain.

We agree that this was a primary challenge of our initial dataset, and trust that the new data addresses these concerns.

(cc) ⓘ BY

[Figure]

**Table 2:** *Comparison between CS Flow salt dilution and S4W-Nepal FlowTracker measurements. Five measurements were identified for evaluation. In one case (i.e. CS Flow Msmt Date 4/16/2018 10:03), a linear interpolation between two S4W-Nepal measurements (i.e. 3/15/2018 7:15 and 5/23/2018 14:23) was made because measurements for both March and May were available. Data summarized at the bottom with average, minimum (min), maximum (max), and standard deviation (std dev).*

[Figure]

| S4W-Nepal SiteID | CS Flow Msmt Date | S4W-Nepal Msmt Date | CS Flow Salt Dilution Q (m3 s-1) | S4W-Nepal FlowTracker Reference Q (m3 s-1) | % Difference |
|---|---|---|---|---|---|
| DB02 | 4/16/2018 10:03 | 3/15/2018 7:15 and 5/23/2018 14:23 | 0.0 | 0.0492 | 7.5% |
| BM02 | 4/16/2018 13:22 | 3/15/2018 9:49 | 0.0169 | 0.0172 | -1.7% |
| NA02 | 4/18/2018 13:28 | 3/31/2018 4:39 | 0.0940 | 0.1110 | -15.3% |
| BA01 | 4/20/2018 11:30 | 3/30/2018 9:24 | 0.0090 | 0.0118 | -23.7% |
| NK03 | 4/18/2018 13:57 | 5/16/2018 12:22 | 0.0461 | 0.0454 | 1.5% |
| | | average -> | 0.0438 | 0.0469 | -6.3% |
| | | min -> | 0.0090 | 0.0118 | -23.7% |
| | | max -> | 0.0940 | 0.1110 | 7.5% |
| | | std dev -> | 0.0301 | 0.0353 | 11.5% |

[Figure]

[Figure]

**4    Discussion**

**4.1    Preferred measurement method**

Based on 20 flow measurements performed in this study, we concluded that the salt dilution method will (1) provide the
10    most accurate streamflow data (at least for the range of flows observed), and (2) will be the easiest method for citizen scientists to repeat in the field with limited amounts of training and equipment (see Section 4.4 on citizen scientist repeatability).

While all flow measurement methods evaluated had positive bias, salt dilution showed the closest agreement to the
15    reference flow with an average over estimation of only one percent (based on the linear regression), followed by float and Bernoulli at 5 and 25 %, respectively (Fig. 3).  The standard deviation for errors was 17 % for salt dilution, and 29 and 62 % for float and Bernoulli, respectively.  Additionally, r-squared values indicated that salt dilution had the least amount of variance from the trend line (i.e. closest to one).

20    Only three salt dilution measurements (i.e. 17032201, 17041901, and 17033001) had percent differences larger than 20 %, and these were all positively biased in relatively small streams (flows between 0.018 and 0.021 m$^3$ s$^{-1}$).  While we can't be certain, we suggest that these errors may be due hyporheic exchanges that removed some salt solution from the measurement

**Number: 1  Author: reviewer  Subject: Note  Date: 2018-09-14 15:47:58**

You need to give prove that this is a valid procedure!

This section has been removed.

**Number: 2  Author: reviewer  Subject: Note  Date: 2018-09-14 16:19:43**

Is this valeu the average of both measurements? Better sate both values individually to be consistent with all other values.

This section has been removed.

**Number: 3  Author: reviewer  Subject: Note  Date: 2018-09-14 15:49:19**

Please indicate these sites in the maps!

This section has been removed.

**Number: 4  Author: reviewer  Subject: Note  Date: 2018-09-14 15:51:38**

Why not show the agreement between the salt dilution and the Bernoulli method. If the Bernoulli method has not been part of the CS-campain you need to better describe this in the method section. Right now  one would assume all three methods were used in the SC-campain.

This section has been removed.

**Number: 5  Author: reviewer  Subject: Note  Date: 2018-09-14 16:29:51**

Why not also discuss the other methods and give a suggestion at the end?

**Number: 6  Author: reviewer  Subject: Note  Date: 2018-09-14 16:20:38**

see my comment on "bias" above

This has been noted and uses of bias have been reviewed.

**Number: 7  Author: reviewer  Subject: Note  Date: 2018-09-14 16:21:47**

This reads more like results. Consider to shift this information to the result section.

These types of descriptions have been moved to Section 3 results.

**Number: 8  Author: reviewer  Subject: Note  Date: 2018-09-14 16:24:16**

Give the standard deviation, coefficient of variation or variance to say something about variance.

This analysis has been removed.

[Figure]

reach before lateral and vertical mixing could fully occur. In other words, it is possible that some of the salt solution became "underflow" shortly after the injection point and did not return to the surface stream prior to the EC measurement location. As observed, this "removal" of salt solution would lead to a systematic overestimation of flow. If these three measurements are removed, the mean and standard deviation for salt dilution method percent differences become 2 and 9 %, respectively.

5   These percent differences fall within the expected range of uncertainty presented in the literature for salt dilution gauging (Day 1976; USBR 2001; Moore 2004a; Herschy 2014). Excluding these three errors, and assuming errors are normally distributed, we expect that salt dilution measurements will be within roughly ±18 % (95 % confidence interval).

**4.2    Measurement error correlation**
[Figure]

10  In Fig. 4, three "outlier" percent errors for salt dilution measurements were seen in the middle row of sub-plots (5 through 8) as clusters of three points towards the top of each sub-plot. After removing these points, the Pearson's r value decreased to 0.26, and the correlation became statistically insignificant (n = 20, p = 0.1, r > 0.378). Therefore, we are cautious to conclude that error in salt dilution measurements decreases as the amount of streamflow increases. Errors in salt dilution measurements appeared to be uncorrelated with the other variables evaluated (e.g. average velocity, slope, and $EC_{BG}$).

The other observed statistically significant correlation was an inverse relationship between average velocity and error in float measurements. Our experience in the field validated that slow moving (and shallow) float velocity measurements were difficult to perform. The combination of turbulence and boundary layer impacts from the streambed and the overlying air mass often made floating objects on the surface travel in non-linear paths, adding uncertainty to distance and time

20  measurements. Challenges with applying the float method in shallow depths was supported USBR (2001) and Escurra (2004), who showed that uncertainty in surface velocity coefficients (i.e. the ratio of surface velocity to actual mean velocity of the underlying water column) increased as depth decreased, especially below 0.3 m.

**4.3    Salt dilution calibration coefficient (k)**

25  Moore (2005) suggests that k depends on (1) the ratio of salt and water in the tracer solution and (2) the chemical composition of the stream water. To minimize variability in k due to changes in salt concentration, a fixed ratio of salt to water (e.g. 1 to 6 by mass) should be consistently used to prepare tracer solutions (as it was during this investigation). Significant correlations observed between k and longitude and elevation may be due to changes in water chemistry that co-vary with these independent variables. Measurements performed in the northeastern portion (i.e. higher longitude) of the

30  Kathmandu Valley were higher in altitude. Geology in the north of the Kathmandu Valley is a mixture of weathered igneous and metamorphic parent material (e.g. gneiss, phyllite, schist, etc.). Geology surrounding measurements in the southwest of the Kathmandu Valley is dominated by sedimentary and slightly metamorphosed deposits of sand, silt, and clay (Shrestha et

**Number: 1    Author: reviewer    Subject: Note    Date: 2018-09-14 16:38:34**

This section should also mention potential differences in errors when the three different methods are applied by you as experts and by citizens or at least refer to the section you have on CS.

We agree that the difference in errors between experts and CS is critical. This is now one of the main points of section 4.2.

**Number: 2    Author: reviewer    Subject: Note    Date: 2018-09-14 16:44:58**

Please also mention the error associated with the Doppler radar as a function of discharge (information provided by the selling company) Also the Dopper radar has it's error!

Indeed! This is discussed in Section 2.1.2 now.

**Number: 3    Author: reviewer    Subject: Note    Date: 2018-09-14 16:32:43**

please use the a), b), c), ...to refer to the panels you take about to be clear.

This section has been removed.

**Number: 4    Author: reviewer    Subject: Insert Text    Date: 2018-09-14 16:42:43**

as a function of discharge

This section has been removed.

**Number: 5    Author: reviewer    Subject: Note    Date: 2018-09-14 16:33:55**

I think this is interesting to mention in the result section! As it changes the outcome of your analysis!

This section has been removed.

**Number: 6    Author: reviewer    Subject: Note    Date: 2018-09-14 16:47:36**

Not clear what you are referring to. also Fig 4? But the it is the Absolute percent error of the salt dilution measurement and the doppler radar-based dischage. Please also sy Fig 4 xy.

This section has been removed.

**Number: 7    Author: reviewer    Subject: Note    Date: 2018-09-14 16:36:45**

Did you do a float velocity test  only once? Typically one can do several estimates and average the values to compensate for the effects you mention. Maybe mention waht you did in detail in the method section.

Section 1.3 explains the float measurements, including the typical number of sub-sections (usually between 5 and 20).

**Number: 8    Author: reviewer    Subject: Note    Date: 2018-09-14 16:50:08**

In the first sentence you state the two factors influencing k. Lat and Long are most likely not relevant and meaningful influencing factors, just spurious correlations.

The k correlation analysis has been removed.

[Figure]

al. 2012). These differences in geology could impact water chemistry through water-rock interactions (Lasaga 1984) and ultimately impact k. Additional work should focus on improving our understanding of the variables affecting k.
[Figure]
 Specifically, spatial variability in k due to changes in stream water chemistry should be investigated prior to applying the salt dilution methodology described in this paper into other areas.

**4.4 Citizen scientist repeatability**

In this context, repeatability refers to the overall likelihood that the measurement method can be successfully repeated by citizen scientists. Along these lines, there were several practical observations in the field worth briefly discussing. While difficult to quantitatively evaluate, we offer the following qualitative observations regarding the selection of a preferred citizen science streamflow measurement method.

- **Required training** - Float and salt dilution require similar amount of training, which from our experience we estimate to be roughly four hours, involving both classroom and field time. The amount of training is strongly dependent on the background of the volunteers. Bernoulli requires additional training for how to minimize vertical movement of the metal measurement plate.

- **Cost of equipment** - All methods require a SmartPhone, measuring scale, and measuring tape. Additionally, salt dilution requires an inexpensive EC meter (e.g. $15 HoneForest Water Quality Tester), a graduated cylinder, and a bucket.

- **Number of citizen scientists required** - Teams of at least two citizen scientists are recommended for all methods; teams of three were preferred in our experience.
[Figure]

- **Data recording requirements** - For float and Bernoulli, depth, width, and velocity (including distance and time) data needs to be recorded at multiple locations. Salt dilution only requires some basic data entry and a video of the breakthrough curve.

- **Complexity of procedure** - Float and Bernoulli require detailed transects of the stream. Bernoulli is extremely sensitive to vertical movements in the metal measurement plate. Bernoulli is always very difficult for low velocities.

- **Enjoyability of measurement** - We found that citizen scientists generally enjoyed watching the salt dilution breakthrough curves and found them less repetitive than the tasks associated with float and Bernoulli methods. Bernoulli measurements can be frustrating when trying to keep the metal measurement plate from moving vertically, especially when there is a soft streambed.

- **Safety** - Both float and Bernoulli measurements require citizen scientists to wade through the stream. At certain flow rates this clearly poses a safety risk, especially for people who cannot swim. A clear benefit of the salt dilution

**Number: 1  Author: reviewer  Subject: Note  Date: 2018-09-14 16:52:30**

Different geology effects the background concentration, for sure, but should not effect k.
Please reconsider the content of this paragraph.

Response 1: We have removed this paragraph from the revised manuscript.

**Number: 2  Author: reviewer  Subject: Note  Date: 2018-09-14 17:09:13**

I think this section is useful for the reader.
For you chosen title I would encourage you to check your data and see if you can use it for additional research questions concerning the Citizen Science aspect.
E.g. the quality of the measurements could be a function of the person/or group that performed the calibration. Comparing your dataset (n=20) wiht the CS dataset (n=145) you could maby see if experts do "better" than "citizes" or not.

Response 2: We believe the new experimental design addresses these issues.

**Number: 3  Author: reviewer  Subject: Note  Date: 2018-09-14 17:01:26**

This is an important information you need to mention in the method section. You actually did not have 20 students but 7 teams of 3 persons?

Response 3: This has all been clarified in the methods section now!

[Figure]

method is that everything but the simplified float estimate can be performed from the stream bank. Note that the results from the simplified float method are only used to determine the salt dosing. If the salt dosing doesn't provide a large enough change in EC to be clearly observed, the measurement can be repeated with a higher flow estimate and corresponding increase in tracer solution.

**4.5 Uncertainty in reference flows**

[Figure]

Uncertainty in measurements of reference flow (i.e. actual flow) affected uncertainty in our evaluation of the three flow measurement methods. Based on an ISO discharge uncertainty calculation within the SonTek FlowTracker software, the uncertainties in flow ranged from 2.5 to 6.0 %, with a mean of 3.7 %. Based on the literature (Rantz 1982; Harmel 2006; Herschy 2014), these uncertainties in reference flows are towards the lower end of the expected range for field measurements of streamflow. Therefore, we do not think that any systematic biases or uncertainties in our data change the results of this paper.

**4.6 Reynolds number**

[Figure]

Turbulent mixing of flow is an important aspect of salt dilution flow measurements. Reynolds number is typically used as a quantitative measure of turbulence in fluid flow. In addition to density and viscosity, fluid velocity and a characteristic length are required for calculating Reynolds number. Many of our measurements were performed in mountainous headwater streams with high slopes. To collect the most accurate reference flow measurements, however, we selected the lowest
 gradient stretch of the stream. The cross sections used for reference flow measurements were typically in the widest and deepest reaches of the stream to ensure the most laminar flow lines for accurate velocity and area measurements. Using velocity and characteristic length data from these reference flow measurement locations to calculate Reynolds number would not have been representative of the actual average Reynolds number of the stream reaches used for the salt dilution measurements. This is because selected stream reaches for salt dilution included steep gradients with lots of mixing, which were often either entirely upstream or downstream of reference flow locations. Therefore, we did not include Reynolds number in the correlation analysis for salt dilution.

**4.7 Flow measurement methods not evaluated**
[Figure]

We initially considered including the slope area method (USBR 2001) based on the Manning's equation for evaluation. The concept was to use a long clear flexible tube of a known length (e.g. 20 m) to measure the slope of the water surface using the principle of a water level. The tube was completely submerged and filled with water. The upstream end of the tube remained submerged and the entrance was held perpendicular to flow to ensure that only the pressure head of the stream was

**Number: 1  Author: reviewer  Subject: Note  Date: 2018-09-14 17:34:13**

Ah, here you mention the acuracy of the doppler radar device! Good!

I assume the errror is depending on the flow velocity and has a lower and upper rage of proper application given by the producer. Pease mention this here.

Response 1: The accuracy of the FlowTracker ADV is now provided in the methods section.

**Number: 2  Author: reviewer  Subject: Note  Date: 2018-09-14 17:45:13**

maybe union with the section before

Response 2: The correlation analyses originally included have been removed so this material is no longer relevant.

**Number: 3  Author: reviewer  Subject: Note  Date: 2018-09-14 17:35:45**

give range and mean slope you used in brackets

Where this step and pool river bads?

Response 3: The types of streams is clarified in methods now.

**Number: 4  Author: reviewer  Subject: Note  Date: 2018-09-14 17:45:50**

move to the other methods that I think would all better fit in the introduction section.

Response 4: Good idea.  This is done in the revised manuscript.

[Figure]

sensed at the tube inlet. The tube was stretched out longitudinally along the stream reach, and the downstream end of the tube was exposed to the atmosphere. The difference in water levels inside the downstream end of the tube and the stream water level immediately outside of the tube
measured. This change in head was divided by the total length of the tube to determine the slope of the water surface.

Within the first few days of field work, we concluded that this method was not suitable for piloting at the types of sites we were investigating. Because we were particularly interested in high gradient headwater streams, the primary problem was finding a stretch of stream long enough that was flowing at normal depth, without backwater and drops in the water surface caused by sudden changes in channel geometry (both longitudinally and latitudinally). An additional challenge of this method is that uncertainty in flow measurements are linearly proportional to uncertainty in estimations of the roughness coefficient (n), and n is difficult to estimate visually, especially for citizen scientists. Therefore, Manning's method was not included in this investigation. Despite our experience, we suggest that in certain settings with long straight reaches of uniform flow, Manning's may still be an appropriate citizen science flow measurement method. To simplify observations of the change in head, the Manning's method would also benefit from the installation of upstream and downstream staff gauges survey to a common datum. However, this would make the method difficult and costly to implement, and therefore less scalable.

**4.8    CS Flow campaign**

The CS Flow Campaign provided us with a unique opportunity to evaluate the preferred salt dilution citizen science streamflow measurement method at a larger scale. In addition to the valuable streamflow data that will help us characterize the hydrological situation in the Kathmandu Valley with greater precis we also learned many practical lessons about how to apply citizen science based streamflow generation methods at a broader scale. Unfortunately, there was no systematic way to evaluate the accuracy of all the measurements performed. However, at the five locations were S4W-Nepal FlowTracker measurements were available, the resulting errors ($\mu$ = -6.7 %, std dev = 11.5 %; Table 2) where comparable to our initial evaluation data ($\mu$ = 8.2 %, std dev = 17.2 %; Table 1).

Linear regression forced through the origin between reference flows and salt dilution measurements had slopes of 0.90 and 1.01 with an r-squared values of 0.97 and 0.98 for CS Flow measurements and our initial evaluation measurements, respectively. Goodness of fit was similar, but while evaluation measurements had a slight positive bias (1 %), CS Flow measurements had a larger negative bias (-10 %). One possible explanation for this is that the three CS Flow comparisons that had negative percent differences (i.e. BM02, NA02, and BA01; Table 2) all used reference flows performed prior to CS Flow measurements. Since hydrographs during this season are gradually receding prior to the onset of the South Asian Monsoon, it is possible that the actual flow of the streams decreased between reference flow and CS Flow observations.

**Number: 1  Author: reviewer  Subject: Note  Date: 2018-09-14 17:40:27**

I like your method! Still the section is too long for a method you finally decided to be not practical and did not use in this paper! You could shorten the text and should mention the Manning-Strickler Method as alternative (that you finally not chose) in the Introduction toughest with all other methods.

This has been removed and the Manning-Strickler method is included in the introduction section now.

**Number: 2  Author: reviewer  Subject: Note  Date: 2018-09-14 17:47:08**

Ah, you need to stat in the method section that the CS campaign (n + 145) only was on applying salt dilution not all other methods.

This has now been clarified in the methods section.

**Number: 3  Author: reviewer  Subject: Note  Date: 2018-09-14 18:01:28**

You dataset is good and includes a lot of potential. However the campaign needed to be repeated to learn more about the hydrology of the area. You could discuss if this is feasible or not.

We now have pre and post monsoon data and lots of ideas about the feasibility of repeating these measurements.

**Number: 4  Author: reviewer  Subject: Note  Date: 2018-09-14 18:09:54**

see my concern about the reference measurements +/- 1 month befoer/after the CS campain.

These comparisons have now been removed.

**Number: 5  Author: reviewer  Subject: Note  Date: 2018-09-14 18:11:11**

If this is the case than you definitely cannot take the dropper radar based discharge as reference for you CS campain!

This was a challenge of the initial dataset that we believe has been addressed with the new "Phase 2" data.

[Figure]

As flows decreased, we observed a progressively increasing positive bias between simplified float estimates and salt dilution measurements (Fig. 7). This finding is congruent with previous efforts to characterize the dynamic relationship between channel depth and surface velocity coefficients (USBR 2001; Escurra 2004). Average stream depths were often on the order

5 of a few centimeters for the headwater catchments observed. Surface velocity coefficients provided by USBR (2001) range from 0.66 to 0.80 with increasing depths, but are held constant at 0.66 for depths less than 0.3 meters. Our results indicate that for flows less than 0.01 $m^3$ $s^{-1}$ a surface velocity coefficient of 0.5 would be more appropriate. The strength of the relationship between salt dilution and float streamflows also deteriorates as flows decrease (i.e. r-squared equals 0.83, 0.69, and 0.50 for plots 1, 2, and 3, respectively). This suggests that surface velocity coefficients are highly variable at low flow

10 rates and correspondingly shallow depths.

**5 Summary and future work**

Our aim in this paper was to (1) evaluate possible citizen science streamflow measurement methods, (2) select a preferred approach, and (3) pilot test the selected method in a real-world setting. We evaluated three different approaches (i.e. float,

15 salt dilution, and Bernoulli run-up) by performing 20 side by side comparison measurements in headwater catchments of the Kathmandu Valley. We used USGS mid-section discharge measurements from a SonTek FlowTracker acoustic Doppler velocimeter as reference flows. Evaluated flows ranged from 0.006 to 0.240 $m^3$ $s^{-1}$. Linear regressions forced through the origin for scatter plots with reference flows had slopes of 1.05, 1.01, and 1.26 with r-squared values of 0.90, 0.98, and 0.61, for float, salt dilution, and Bernoulli run-up methods, respectively. The salt dilution method was selected as the preferred

20 approach based on its favourable quantitative results compared to the other methods, and other qualitative factors concerning citizen science repeatability. The approach was then pilot tested in a CS Flow Campaign, which involved 20 volunteers performing 145 measurements, ranging from 0.0004 to 0.425 $m^3$ $s^{-1}$, distributed among the 10 headwater catchments of the Kathmandu Valley. While there was no way to evaluate the accuracy of all 145 measurements, five of the measurements were performed in locations where USGS mid-section method discharge measurements had been performed. For these five

25 locations, a linear regression forced through the origin between reference flows and CS Flow measurements had a slope of 0.90 with an r-squared value of 0.97.

Motivated by these promising results, future work should further evaluate the feasibility of applying citizen science based salt dilution streamflow measurements to larger areas of Nepal and beyond. Issues of how to motivate citizen scientists and

30 young researchers (i.e. all science and engineering minded students from primary through graduate school ages) to participate in citizen science streamflow measurement efforts should receive additional attention, especially in the relatively

**Number: 1  Author: reviewer  Subject: Note  Date: 2018-09-14 18:13:23**

If your flow was in the order of a few cm than standard friction coefficients/correction factors need to be adapted.

The methods section now clarifies that a constant correction factor of 0.8 was used for float measurements.

**Number: 2  Author: reviewer  Subject: Note  Date: 2018-09-14 18:13:53**

in terms of what?

The methods section now clarifies that a constant correction factor of 0.8 was used for float measurements.

**Number: 3  Author: reviewer  Subject: Note  Date: 2018-09-14 18:16:07**

please mention this relevant information in the method section!! Age, education level etc.)

Details about the volunteers' majors, age, and gender have been added to the methods section.

[Figure]

[Figure]

unexplored context of citizen science in Asia. Finally, the assumption of a constant calibration coefficient (k) should be evaluated over a larger sample size covering a broader range of geological and water quality conditions.

**6    Data availability**

5    The data used in this paper are provided as supplementary material.

**7    Author contribution**

Jeffrey C. Davids had the initial idea for this investigation and designed the experiments in collaboration with Martine M. Rutten, Wessel David van Oyen, and Nick van de Giesen. Field work was performed by Jeffrey C. Davids, Anusha Pandey, Nischal Devkota, Wessel David van Oyen, and Rajaram Prajapati. Jeffrey C. Davids prepared the manuscript with valuable

10    contributions from all co-authors.

**8    Competing interests**

The authors declare that they have no conflict of interest.

**9    Acknowledgements**

This work was supported by the Swedish International Development Agency under Grant number 2016-05801; and by

15    SmartPhones4Water (S4W). We appreciate the dedicated efforts of Annette van Loosen, Bhumika Thapa, Sunil Duwal, volunteers from Khwopa College of Engineering, Anurag Gyawali, Anu Grace Rai, Sanam Tamang, Eliyah Moktan, Kristi Davids, and the rest of the S4W-Nepal team. We would also like to thank Dr. Ram Devi Tachamo Shah, Dr. Deep Narayan Shah, Dr. Narendra Man Shakya, and Dr. Steve Lyon for their supervision and support of this work. Finally, a special thanks to SonTek for their donation of a FlowTracker acoustic Doppler velocimeter that was used for the reference flow

20    measurements discussed in this paper and many more to come.

This research was performed in the context of a larger citizen science project called SmartPhones4Water or S4W (Davids et al. 2017; Davids et al. 2018; www.SmartPhones4Water.org). S4W focuses on leveraging citizen science, mobile technology, and young researchers to improve lives by strengthening our understanding and management of water. S4W's first pilot

25    project, S4W-Nepal, initially concentrated on the Kathmandu Valley, and is now expanding into other regions of the country and beyond. All of S4W's efforts, including the research herein, have a focus on simple field data collection methods that

No Comments.

[Figure]

[Figure]

can be standardized and scaled so that young researchers and citizen scientists can help fill data gaps in other data scarce regions.

10   technology, and thus make a significant contribution towards closing the streamflow data gap.

**1.2    Simple streamflow measurement methods considered**

Streamflow measurement techniques suggested in the United States Bureau of Reclamation Water Measurement Manual (USBR 2001) that seemed potentially applicable for citizen scientists included: deflection velocity meters consisting of shaped vanes projecting into the flow along with a method to measure deflection; the Manning-Strickler slope area method
15   whereby the slope of the water surface in a uniform reach is measured and combined with the Manning formula; and pitot tubes for measuring velocity heads. The float, current meter, and and salt dilution methods described by several authors also seemed applicable (British Standards Institute 1964; Rantz 1982; Fleming and Henkel 2001; Escurra 2004; Moore 2004a, 2004b, and 2005; Herschy 2014). Finally, Wilm and Storey (1944) and Church and Kellerhals (1970) introduced the velocity head rod, or what we later refer to as the Bernoulli run-up  method(or just Bernoulli) method, involving

[revised manuscript text omitted]

Salt dilution method streamflow measurements involve the following steps:

1. Select stream reach with turbulence to facilitate vertical and horizontal mixing
2. Determine upstream point for introducing the salt solution and a downstream point for measuring EC
   a. A rule of thumb in the literature is to separate these locations roughly 25 stream widths apart (Day 1977; Butterworth et al. 2000; Moore 2005)
3. Estimate flow either performing a "simplified float measurement (i.e. only a few sub-sections)" or by visually estimating width, average depth, and average velocity
4. Prepare salt solution based on the following guidelines (adapted from Moore 2005)
   a. 10000 ml of stream water for every 1 m$^3$ s$^{-1}$ of estimated streamflow
   b. 1667 g of salt for every 1 m$^3$ s$^{-1}$ of estimated streamflow
   c. Thoroughly mix salt and water until all salt is dissolved
   d. Following these guidelines ensure a homogenous salt solution with 1 to 6 salt to water ratio by mass
5. Establish the calibration curve relating EC values to actual salt concentrations (Moore 2004b) to determine calibration constant (k) relating changes in EC values in micro Siemens per centimeter ($\mu$S cm$^{-1}$) in the stream to relative concentration of introduced salt solution (RC) (See Sect. 2.1.3 for details)
6. Dump salt solution at upstream location
7. Measure EC at downstream location during salinity breakthrough until EC returns to EC$_{BG}$

a.   Recorded a video of the EC meter screen at the downstream location and later digitized the values using
         the time from the video and the EC values from the meter
8.   Solve for streamflow (Q) with Eq. (2)

**1.3.3   Bernoulli run-up method**

5   Similar to the float method, Bernoulli run-up (or Bernoulli; USBR 2001) is based on the velocity-area principle.  Total
streamflow (Q; L s$^{-1}$) is calculated with Eq. (3):

$$Q = 1000 * \sum_{i=1}^{n} VB_i * d1_i * w_i \hspace{2cm} (3)$$

10   where 1000 is a conversion factor from m$^3$ s$^{-1}$ to L s$^{-1}$, VBi is velocity from Bernoulli run-up (m s$^{-1}$), d1$_i$ is depth (m), and w$_i$
is width (m) of each sub-section (i = 1 to n).  Area for each sub-section is the product of the width and the depth in the
middle of each sub-section.  Velocity for each sub-section (VB$_i$) was determined by measuring the "run-up" or change in
water level on a thin meter stick (or "flat plate;" dimensions: 1 meter long, by 34 mm wide, by 1.5 mm thick used in this
study) from when the flat plate was inserted parallel and then perpendicular to the direction of flow.  The basic principle is
15   that "run-up" on a flat plate inserted perpendicular to flow is proportional to velocity based on the solution to Bernoulli's
equation.  Velocity (VB$_i$; m s$^{-1}$) was calculated from Bernoulli's principle with Eq. (4):

$$VB_i = \sqrt{2g * (d2_i - d1_i)} \hspace{2cm} (4)$$

20   where g is the gravitational constant (m s$^{-2}$) and d2$_i$ and d1$_i$ are the water depths (m) when the flat plate was perpendicular
and parallel to the direction of flow, respectively.

Bernoulli method streamflow measurements involve the following steps:

25   1.   Select constricted stream section with elevated velocity to increase the difference between d1i and d2i
2.   Divide cross section into several sub-sections (n, typically between 5 and 20)
3.   For each section, measure and record
      a.   The depth with a flat plate held perpendicular to flow (d2i or the "Run-up" depth)
      b.   The depth with a flat plate held parallel to flow (d1i or the actual water depth)
30      c.   The width of the sub-section
4.   Solve for streamflow (Q) with Eq. (3) and Eq. (4)

**1.4    Research questions**

Our  aims in this paper were to (1) perform an initial  evaluation of these three potential simple streamflow measurement methods, (2) evaluate the same three methods with actual citizen scientists, and  (3) apply the selected method  at a larger scale.  Our research questions were:

- *Which simple streamflow measurement method provides the most accurate results when performed by "experts?"*
- *Which simple streamflow measurement method provides the most accurate results when performed by citizen scientists?*
- *What are citizen scientists' perceptions of the required training, cost, accuracy, etc. of the evaluated simple streamflow measurement methods?*
- *Can citizen scientists apply the selected streamflow measurement method at a larger scale?*

**1.5    SmartPhones4Water**

This research was performed in the context of a larger citizen science project called SmartPhones4Water or S4W (Davids et al. 2017; Davids et al. 2018; www.SmartPhones4Water.org).  S4W focuses on leveraging citizen science, mobile technology, and young researchers to improve lives by strengthening our understanding and management of water.  S4W's first pilot project, S4W-Nepal, initially concentrated on the Kathmandu Valley, and is now expanding into other regions of the country and beyond.  All of S4W's efforts, including the research herein, have a focus on simple field data collection methods that can be standardized and scaled so that young researchers and citizen scientists can help fill data gaps in other data scarce regions.  S4W-Nepal currently leverages the enthusiasm and schedule breaks in the academic calendar of student citizen scientists to perform campaigns to improve our pre and post monsoon understanding of stone spouts (Nepali: dhunge dhara), land use, and now streamflow.

It should be noted that during the work documented by this paper, the use of "citizen scientist" is restricted to only student citizen scientists, which is a narrow (but important) range of potential citizen scientists.  Our aim was to partner with student "citizen scientists" first to develop and evaluate streamflow measurement methodologies.  Once methodologies are refined in coordination with students, it is our goal to partner with community members and students in the rural hills of Nepal to improve the availability of quantitative stream and spring flow data.

**2    Materials and methods**

**2.1    General**

**2.1.1    Types of streams evaluated**

Streams evaluated during this investigation (Phases 1, 2, and 3) were a mixture of pool and drop and run stream types, with combinations of turbulent and laminar flow . Streamflows ranged from 0.4 to 1804 L s$^{-1}$. Stream widths and average depths ranged from 0.1 to 6.0 m and 0.0040 and 0.97 m, respectively. Streambed materials ranged from cobles, gravels, and sands in the upper portions of watershed to sands, silts, and sometimes man-made concrete streambeds and side retaining walls in the lower portions. During pre-monsoon, sediment loads were generally low, while during post-monsoon, increased water velocities led to increased sediment loads (both suspended and bed). Slopes (based on Phase 2 data) ranged from 0.020 to 0.148 m m$^{-1}$. Additional details about the measurement sites are provided in Tables 3 and 4.

**2.1.2    Reference flows**

To evaluate different simple citizen science flow measurement methods, reference (or actual) flows for each site were needed. We used a SonTek FlowTracker acoustic Doppler velocimeter (ADV) to determine reference flows. The United States Geological Survey (USGS) mid-section method was used, following guidelines from USGS Water Supply Paper 2175 (Rantz 1982), along with instrument specific recommendations from SonTek's FlowTracker manual (SonTek 2009). Stream depths were shallow enough that a single vertical 0.6 depth velocity measurement (i.e. 40% up from the channel bottom) was used to measure average velocity for each sub-section. Depending on the total width of the channel, the number of sub-sections ranged from 8 to 30. The FlowTracker ADV has a stated velocity measurement accuracy of within one percent (SonTek 2009). Based on an ISO discharge uncertainty calculation within the SonTek FlowTracker software, the uncertainties in reference flows for Phase 1 and 2 ranged from 2.5 to 8.2 %, with a mean of 4.2 %. Based on the literature (Rantz 1982; Harmel 2006; Herschy 2014), these uncertainties in reference flows are towards the lower end of the expected range for field measurements of streamflow. Therefore, we do not think that any systematic biases or uncertainties in our data change the results of this paper. A compilation of the measurement reports generated by the FlowTracker ADV, including summaries of measurement uncertainty, are included as supplementary material.

**2.1.3    Salt dilution calibration coefficient (k)**

Our experience was that the most complicated portion of a salt dilution measurement was performing the dilution test to determine the calibration coefficient k. The calibration coefficient k relates changes in EC values in micro Siemens per centimeter (μS cm$^{-1}$) in the stream to relative concentrations of introduced salt solution (RC). During Phases 1 and 2, we determined k using a calibrated GHM 3431 [GHM-Greisinger] EC meter with the following steps (based on Moore 2004b):

1. Make diluted secondary solution by mixing 500 ml of stream water and 5 ml of salt solution

2.  Measure background stream water EC (EC$_{BG}$)

3.  Add known volume (typically 1 or 2 milliliters (ml)) of secondary solution to 500 ml of stream water in dilution cylinder

4.  Measure new dilution cylinder EC

5.  Repeat steps 5.e3 and 5.d4 until the full range of expected EC values wereare observed

6.  Calculate RC for each measurement point

7.  Plot EC on the horizontal axis and RC on the vertical axis

8.  Perform linear regression

9.  Obtain k from the slope of the linear regression

[revised manuscript text omitted]

**2.12.2.1  Citizen science streamflow measurement methods evaluatedInitial evaluation (Phase 1)**

The procedures for each of the three citizen science streamflow measurement methods evaluated are described in the following sections.

5  ### 2.1.1   Float

The float method is based on the velocity area principle.  Total streamflow (Q) in cubic meters per second (m$^3$ s$^{-1}$) was calculated with Eq. (1):

Eq. (1)   $Q = \sum_{i=1}^{n} C * VF_i * d_i * w_i$

where C is a unitless coefficient to account for the fact that surface velocity is typically higher than average velocity (typically in the range of 0.66 to 0.80 depending on depth; USBR 2001), $VF_i$ is surface velocity from float in meters per second (m s$^{-1}$), $d_i$ is depth (m), and $w_i$ is width (m) of each sub-section (i = 1 to n, where n is the number of stations). Surface velocity for each sub-section was determined by measuring the amount of time it takes for a floating object to move

15  a certain distance.  For floats we used sticks found on site.  Sticks are widely available (i.e. easiest for citizen scientists), generally float (except for the densest varieties of wood), and depending on their density are between 40 and 80% submerged, which minimizes wind effects.

Float measurements involved the following steps:

1.        Selected stream reach with straight and uniform flow

2.      Divided cross section into several sub-sections (n, typically between 5 and 20)

3.      For each section, measured and recorded

a.      The depth in the middle of the sub-section

b.      The width of the sub-section

5   c.      The time it takes a floating object to move a known distance downstream (typically 1 or 2 m) in the middle of the sub-section

4.      Solved for streamflow (Q) with Eq. (1)

**2.1.2**   Salt dilution

There are two basic types of salt dilution flow measurements: slug (previously known as instantaneous) and continuous rate

10   (Moore 2004a).  Salt dilution measurements are based on the principle of the conservation of mass.  In the case of the slug method, a single known volume of high concentration salt solution is introduced to a stream and the electrical conductivity (EC) is measured over time at a location sufficiently downstream to allow good mixing (Moore 2005).  In contrast, continuous rate salt dilution method involves introducing a known flow rate of salt solution into a stream (Moore 2004b). Slug method salt dilution measurements are broadly applicable in streams with flows up to 10 $m^3 \cdot s^{-1}$ with steep gradients and

15   low background EC levels (Moore 2005).  For the sake of citizen scientist repeatability, we chose to only investigate the slug method, because of the added complexity of measuring the flow rate of the salt solution for the continuous rate method.

Streamflow (Q; $m^3 \cdot s^{-1}$) was solved for using Eq. (2) (Rantz 1982; Moore 2005):

20   Eq. (2)   $Q = \dfrac{V}{k \sum_{i=1}^{n}(EC(t) - EC_{BG}) \Delta t}$

where V is the total volume of tracer introduced into the stream ($m^3$), k is the calibration constant in centimeters per microsiemens (cm $\mu S^{-1}$), n is the number of measurements taken during the breakthrough curve (unitless), EC(t) is the EC at time t ($\mu S \, cm^{-1}$), $EC_{BG}$ is the background EC ($\mu S \, cm^{-1}$), and $\Delta t$ is the change in time between EC measurements (s).

We performed the following steps when making a salt dilution measurement:

1.      Selected stream reach with turbulence to facilitate vertical and horizontal mixing

2.      Determined upstream point for introducing the salt solution and a downstream point for measuring EC

30   a.      A rule of thumb in the literature is to separate these locations roughly 25 stream widths apart (Day 1977; Butterworth et al. 2000; Moore 2005)

3.      Estimated flow rate visually by estimated width, average depth, and average velocity

4.      Prepared salt solution based on the following guidelines (adapted from Moore 2005)

a.        10000 ml of stream water for every 1 m$^3$s$^{-1}$ of estimated streamflow

b.        1667 g of salt for every 1 m$^3$s$^{-1}$ of estimated streamflow

c.        Thoroughly mix salt and water until all salt is dissolved

d.        Following these guidelines ensured a homogenous salt solution with 1 to 6 salt to water ratio by mass

5. 5.        Performed dilution test (Moore 2004b) to determine calibration constant (k) relating changes in EC values in micro Siemens per centimeter (µS cm$^{-1}$) in the stream to relative concentration of introduced salt solution (RC)

a.        Made diluted secondary solution by mixing 500 ml of stream water and 5 ml of salt solution

b.        Measured background stream water EC (EC$_{BG}$)

c.        Added known volume (typically 1 or 2 milliliters (ml)) of secondary solution to 500 ml of stream water in dilution

10 cylinder

d.        Measured new dilution cylinder EC

e.        Repeated steps 5.c and 5.d until the full range of expected EC values were observed

f.        Calculated RC for each measurement point

g.        Plotted EC on the horizontal axis and RC on the vertical axis

15 h.        Performed linear regression

i.        Obtained k from the slope of the linear regression

6.        Dumped salt solution at upstream location

7.        Measured EC at downstream location during salinity breakthrough until EC returns to EC$_{BG}$

a.        Recorded a video of the EC meter screen at the downstream location and later digitized the values using the time

20 from the video and the EC values from the meter

8.        Solved for streamflow (Q) with Eq. (2)

**2.1.3**        Bernoulli run up

Similar to the float method, Bernoulli run up (or Bernoulli) is based on the velocity-area principle. Total streamflow (Q; m$^3$ s$^{-1}$) was calculated with Eq. (3):

Eq. (3)    $Q = \sum_{i=1}^{n} VB_i * d1_i * w_i$

where VB$_i$ is velocity from Bernoulli run up (m s$^{-1}$), d1$_i$ is depth (m), and w$_i$ is width (m) of each sub-section (i = 1 to n). Area for each sub-section is the product of the width and the depth in the middle of each sub-section. Velocity for each sub-

30 section (VB$_i$) was determined by measuring the "run-up" or change in water level on a thin meter stick from when the stick was inserted parallel and then perpendicular to the direction of flow. The basic principle is that "run-up" on a flat plate inserted perpendicular to flow is proportional to velocity based on the solution to Bernoulli's equation. Velocity (VB$_i$; m s$^{-1}$) was calculated from Bernoulli's principle with Eq. (4):

Eq. (4)  $VB_i = \sqrt{2g * (d2_i - d1_i)}$

where g is the gravitational constant (m s$^{-2}$) and d2$_i$ and d1$_i$ are the water depths (m) when the flat plate was perpendicular and parallel to the direction of flow, respectively.

Bernoulli run up measurements involved the following steps:

1.      Selected constricted stream with elevated velocity to increase the difference between d1i and d2i
2.      Divided cross section into several sub-sections (n, typically between 5 and 20)
3.      For each section, measured and recorded
a.      The depth with a flat plate held perpendicular to flow (d2i or the "Run-up" depth)
b.      The depth with a flat plate held parallel to flow (d1i or the actual water depth)
c.      The width of the sub-section
4.      Solved for streamflow (Q) with Eq. (3) and Eq. (4)

**2.2**      Reference flow

To evaluate the different citizen science flow measurement methods, a reference (or actual) flow for each site was needed. We used a SonTek FlowTracker acoustic Doppler velocimeter (ADV) to determine reference flows. The United States Geological Survey (USGS) mid section method was used, following guidelines from USGS Water Supply Paper 2175 (Rantz 1982), along with instrument specific recommendations from SonTek's FlowTracker manual (SonTek 2009). The FlowTracker ADV has a stated velocity measurement accuracy of within one percent (SonTek 2009). Flow measurement errors, calculated with an International Standards Organization (ISO) approach built into the FlowTracker software, are typically in the range of 3 to 10 %. Reference flow errors in this study are discuss in Section 4.5. A compilation of the measurement reports generated by the FlowTracker ADV are included as supplementary material.

**2.3**      Flow measurement method evaluation and analysis

To perform an initial evaluation of the selected methods, we (the authors) performed measurements at 20 sites in March and April of 2017 in headwater catchments of the Kathmandu Valley (Fig. 2). Sites were chosen to represent a typical range of stream types, slopes, and flow rates. At each site, we performed float, salt dilution, and Bernoulli measurements, in addition to reference flow measurements with the FlowTracker ADV per the descriptions in Sect. 1.3 and 2.1.2, respectively. All Phase 1 salt dilution EC measurements were taken with a calibrated GHM 3431 [GHM-Greisinger] EC meter.

At each site, measurements were performed consecutively, and took roughly one to two hours to perform, depending on the size of the stream and the resulting number of sub-sections for float, Bernoulli, and reference flow measurements.

Measurements were performed during steady state conditions in the stream; if runoff generating precipitation occurred during measurements at a site, the measurements were stopped, and repeated after streamflows stabilized at pre-event levels. As previously described, salt dilution calibration coefficient k was determined at 10 of the 20 sites. Field notes for float, salt dilution, and Bernoulli were taken manually and later digitized into a spreadsheet (included in supplementary materials).

5    Results from Phase 1 are summarized in map and tabular form (Fig. 2; Table 3).We first summarized flow measurement method evaluation results in map and tabular form (Fig. 1; Table 1). Measurement ID can be used to link data between the map and table. We used scatter plots to compare reference flow (x-axis) to the three flow measurement methods evaluated (y-axis) to visualize and interpret results from each method. We fitted these points with a linear regression forced through the origin. To understand relative (normalized) errors, we calculated percent differences in relation to reference flow for

10   each method. Averages of absolute value percent differences (absolute errors), average errors (bias), and standard deviations of errors were used as metrics to compare results among methods and between Phase 1 and 2. To better understand possible explanations for observed variability in our results, we performed a correlation analysis. For each method, we performed a Pearson's r correlation analysis (Lee Rodgers and Nicewander 1988) between the absolute value of percent difference in flow and (1) reference flow, (2) average velocity, (3) $EC_{BG}$, and (4) slope. Slope values were developed using elevations

15   from the Google Earth Digital Elevation Model (DEM) obtained along the centreline of the stream alignment both 100 meters upstream and downstream of each measurement point (retrieved July 2nd, 2018). While using DEM data for slope calculations is clearly inferior to performing topographic surveys in the field, this was not possible due to lack of equipment and time; therefore, these slope data are the best available numbers.

20   **2.4    Salt dilution calibration coefficient (k) analysis**

Arguably, one of the most complicated portions of a salt dilution measurement is performing the dilution test to determine the calibration coefficient k (Moore 2004b). To determine if the dilution test needs to be repeated for each citizen science measurement, we analyzed all k values determined during this study. In addition to the mean, range, and standard deviation, we performed a Pearson's r correlation analysis (Lee Rodgers and Nicewander 1988) to see if k showed statistically

25   significant trends with latitude, longitude, elevation, and $EC_{BG}$.

**2.2.2    Citizen scientist evaluation (Phase 2)**

To evaluate the same three streamflow measurement methods with actual citizen scientists, we recruited 37 student volunteers from Khwopa College of Engineering in Bhaktapur, Nepal for our Citizen Science Flow (CS Flow) evaluation. 10 CS Flow evaluation groups of either three or four members were formed. Citizen scientists were allsecond and third-

30   year civil engineering Bachelors' students ranging in age from 21 to 25; 132 were female and 245 were male. Phase 2 started on 17 September (2018) with a four-hour theoretical training on the float, salt dilution, and Bernoulli streamflow

measurement methods per Sect. 1.3.  The theoretical training also introduced citizen scientists to Open Data Kit (ODK; Anokwa et al. 2009) in general, and the specific streamflow measurement workflow described below.

Based on our initial experiences and results from Phase 1 we developed an  ODK form to facilitate the collection of float, salt dilution, Bernoulli, and reference streamflow measurement data.  After installing ODK on an Android smartphone, and downloading the necessary form from S4W-Nepal's ODK Aggregate server on the Google Cloud App Engine, the general workflow was as follows:

1.  Launch ODK and select Stream Flow (v1.1) form
2.  Record measurement date and time and GPS coordinates
3.  Select flow measurement methods to perform (i.e. float, salt dilution, and Bernoulli)
    a.  Note that the "expert" group also selected FlowTracker for reference flow
4.  Record float data (e.g. distance, time, depth, width) per Sect. 1.3.1
5.  Record Bernoulli data (e.g. depth1, depth2, width) per Sect. 1.3.3
6.  Use float flow measurement results to determine recommended salt dose per Sect. 1.3.2
7.  Record GPS and take pictures of  salt injection  locations
8.  Enter actual amount of salt used based on possible combinations of pre-weight packets of salt (e.g. 10 g, 20 g, 50 g, 100 g, 500 g, etc.)
9.  Based on actual amount of salt used, the app calculates the amount of stream water needed to prepare the tracer solution
10. Prepare tracer solution using pre-weighed salt packets and graduated measuring cylinders
11. Record GPS and take pictures of  EC measurement locations
12. Add tracer solution to stream and record video of EC breakthrough curve
    a.  Note that all Phase 2 salt dilution EC breakthrough curve measurements were performed with inexpensive [HoneForest] meters.
13. Submit form to ODK Aggregate server

Training was continued on 18 September with a two-hour field demonstration session in the Dhobi watershed located in the north of the Kathmandu Valley (Fig. 3).  During this field training, we worked with three to four groups at a time, and together performed float, salt dilution, and Bernoulli measurements at site D3.

Following the field training, a Google My Map with 15 measurement sites was provided to the citizen scientists: seven in the Dhobi and eight in the Nakkhu watersheds (Fig. 3).  Sites were chosen to represent a typical range of stream types, slopes,

and flow rates found within the headwater catchments of the Kathmandu Valley, and to minimize travel time between locations. Groups were strictly instructed to not discuss details regarding the selection of measurement reaches or the results of the streamflow measurements with other groups. For the remainder of 18 September and all of 19 September, the 10 CS Flow groups rotated between the seven sites in the Dhobi watershed. To ensure that measurements could be compared with each other, four S4W-Nepal interns travelled between sites to verify that CS Flow groups performed measurements on the same streams in the same general locations. All eight measurements on the Nakkhu watershed were performed in similar fashion on 20 September.

Using the same schedule of the CS Flow groups, the "expert" group (authors) visited the same 15 sites. At each site, in addition to performing float, salt dilution, and Bernoulli measurements, the "expert" group performed (1) reference flow measurements per Sect. 2.1.2, (2) salt dilution calibration coefficient k dilution measurements per Sect. 2.1.3, and (3) an auto-level survey to determine average stream slope. At each site, auto-level surveys included topographical surveys of stream water surface elevations with a AT-B4 24X Auto-Level [Topcon] at five locations including: 10 times and 5 times the stream width upstream of the reference flow measurement site (reference site), at the reference site, and 5 and 10 times the stream width downstream of the reference site. For each site, stream slope was taken as the average of the four slopes computed from the five water surface elevations measured.

All CS Flow and "expert" measurements were conducted under steady state conditions. Based on two S4W-Nepal citizen scientists' precipitation measurements (official government records aren't available until the subsequent year) nearby the Dhobi sites (i.e. roughly 3 km to the west and east), no measurable precipitation occurred during 18 and 19 September. Water level measurements from a staff gauge installed at site D3 taken at the beginning and end of 18 and 19 September confirmed that water levels (and therefore flows) remained steady. On 20 September, 7 mm of precipitation was recorded by an S4W-Nepal citizen scientist in Tikabhairab which is roughly 1 km north of the eight measurement sites in the Nakkhu watershed. Based on field observations of the "expert" group, rain didn't start until 15:30 LT, and all CS Flow group measurements were completed before 15:30 LT. Three "expert" measurement sites were completed after 15:30 LT, but most rain was concentrated downstream (to the north) of these sites (i.e. N1, N2, and N3). Based on water level measurements performed at the beginning, middle, and end of measurements at these sites, no changes in water levels (and therefore flows) were observed. We also don't see any systematic impacts to the resulting comparison data for these sites (Table 4 and Fig. 4).

Once ODK forms from all 15 sites were finalized and submitted to the ODK Aggregate server, CS Flow and "expert" groups digitized breakthrough curves (i.e. time and EC) from EC videos in shared Google Sheet salt dilution flow calculators. Digitizations for all measurements were then reviewed for accuracy and completeness by the authors.

(1) selected an appropriate measurement reach with good mixing and minimal bank storage, (2) performed a simplified float measurement (i.e. only a 3 or 4 depths and velocities), (3) used the float flow estimate and ECBG to provide citizen scientists recommended salt/water dose, (4) used pre weight packets of salt (e.g. 10 g, 20 g, 50 g, 100 g, etc.) to prepare tracer solution, (5) added tracer solution to stream and recorded video of EC breakthrough curve, (6) submitted form to ODK Aggregate server, (6) digitized breakthrough curve (i.e. time and EC) in shared Google Sheet salt dilution flow calculator.

After the completion of Phase 2 field work, a Google Form survey was completed by 33 of the Phase 2 citizen scientists. The purpose of the survey was to evaluate citizen scientists' perceptions of the three simple streamflow measurement methods. The survey questions forced participants to rank each method from 1 to 3. Questions were worded so that in all cases, a rank of 1 was most favourable and 3 was least favourable. The following survey questions were included:

- Q1 - Required training for each method (1 least and 3 most)
- Q2 - Cost of equipment for each method (1 least and 3 most)
- Q3 - Number of citizen scientists required for each method (1 least and 3 most)
- Q4 - Data recording requirements for each method (1 least and 3 most)
- Q5 - Complexity of procedure for each method (1 least and 3 most)
- Q6 - Enjoyability of measurement method (1 most enjoyable and 3 least enjoyable)
- Q7 - Safety of each method (1 safest and 3 least safe)
- Q8 - Accuracy of each method (1 most accurate and 3 least accurate)

A tabular summary of the 15 Phase 2 measurement locations was developed (Table 4). To understand relative (normalized) errors, we calculated percent differences in relation to reference flow for each method. Averages of absolute value percent differences (absolute errors), average errors (bias), and standard deviations of errors were used as metrics to compare results among methods and between Phase 1 and 2. Box plots showing the distribution of CS Flow group measurement errors along with "expert" measurement errors for each method were developed (Fig. 4). To visualize the results of the citizen scientists' perception survey, a stacked horizontal bar plot grouped by streamflow measurement methods was developed (Fig. 5).

**2.52.2.3 2.5. Citizen scientist application (Phase 3)**

From 15 to 21  April (2018; pre-monsoon) and 21 to 25  September (2018; post-monsoon), 25 and 37 second and third-year engineering Bachelors' student citizen scientists, respectively, from Khwopa College of Engineering in Bhaktapur, Nepal joined S4W-Nepal's Citizen Science Flow (CS Flow) campaign. Citizen scientists formed

float flow estimate and $EC_{BG}$ to provide citizen scientists recommended salt/water dose, (4) used pre-weight packets of salt (e.g. 10 g, 20 g, 50 g, 100 g, etc.) to prepare tracer solution, (5) added tracer solution to stream and recorded video of EC breakthrough curve, (6) submitted form to ODK Aggregate server, (6) digitized breakthrough curve (i.e. time and EC) in shared Google Sheet salt dilution flow calculator.

During S4W Nepal's Citizen Science Flow (CS Flow) campaign (15th to 21st of April 2018; Fig. 6), student volunteers from Khwopa College of Engineering 8 pre-monsoon and 10 post-monsoon CS Flow groups of three or four people each, respectively. Ages of pre-monsoon citizen scientists ranged from 21 to 25; 7 were female and 18 were male (post-monsoon group composition is described in Sect. 2.2.2). were recruited, trained, divided into groups by sub-watershed, and sent to the field to perform salt dilution flow measurements.

Post-monsoon Phase 3 measurements were performed by the same 10 CS Flow groups that performed Phase 2 citizen scientist evaluations. Therefore, additional training for these groups was not necessary. Training for pre-monsoon CS Flow groups included a four-hour theoretical training on 15 April about the float and salt dilution streamflow measurement methods per Sect. 1.3. The theoretical training also introduced citizen scientists to ODK Android data collection application. For both pre and post-monsoon Phase 3 measurements, the workflow was similar to that described in Sect. 2.2.2, with the exceptions of (1) skipping step 5 (the collection of Bernoulli data), and (2) only performing a "simplified" float measurement (step 4) involving only two or three sub-sections in order to have a flow estimate for calculating the recommended salt dose (step 6). Training was continued on the afternoon of 15 April with a two-hour field demonstration session in the Hanumante watershed located in the southwestern portion of the Kathmandu Valley (Fig. 6). During this field training, we worked with four groups at a time, and together performed "simplified" float and Bernoulli measurements at two sites.

After training was completed, citizen scientists were sent to the field to perform streamflow measurements as described above in all 10 headwater catchments of the Kathmandu Valley (Fig. 6). Note that aAll Phase 3 salt dilution EC breakthrough curve measurements were performed with inexpensive [HoneForest] meters. Once ODK forms from all Phase 3 measurements were finalized and submitted to the ODK Aggregate server, CS Flow groups digitized breakthrough curves (i.e. time and EC) from EC videos in shared Google Sheet salt dilution flow calculators. Digitizations for all measurements were then reviewed for accuracy and completeness by the authors. In the second week, student volunteers used a salt dilution Google Sheet flow calculator to digitize collected measurement data and compute flow (see supplementary material for Excel version). While not included in this paper, it is important to note that Students students analyzed the collected flow data (third week) and finally presented oral and written summaries of their quality-controlled results to their faculty and peers andat Khwopa College of Engineering. S4W Nepal currently leverages the enthusiasm and schedule breaks in the academic calendar of young researchers to perform campaigns to improve our pre and post-monsoon understanding of stone spouts (Nepali: dhunge dhara), land use, and now streamflow.

While subsequent work will highlight the knowledge about spring and streamflows gained from these data, the purpose herein is more a proof of concept showing that the salt dilution method can be successfully applied at a larger scale. As such, a simple map figure is used to show the spatial distribution of pre and post-monsoon measurements. The three streamflow gauging stations within the Kathmandu Valley (only one in a headwater catchment) operated by the  official government agency responsible for streamflow measurements (i.e. the Department of Hydrology and Meteorology or DHM) are also included. Additionally, histograms of flow and EC for pre and post-monsoon are also shown. While measurements in pre and post-monsoon were not all taken in the same location, histograms can still be used to see seasonal changes in distributions.

**3    Results**

       The following results section is organized into the same three primary sub-sections included in the methodology (Sect. 2.2): initial evaluation (Phase 1), citizen scientist evaluation (Phase 2), and citizen scientist flow application (Phase 3).

**3.1    Initial evaluation results (Phase 1)**

[revised manuscript text omitted]

Table 1: Tabular summary of measurement comparison data. Records sorted in ascending order by reference flow (Q Reference). Latitude and longitude in reference to the WGS84 datum. All flow values shown are shown in $m^3 s^{-1}$ rounded to the thousandth place. Percent differences calculated using Q Reference (FlowTracker) as the actual flow. Data summarized at the bottom with average, minimum (min), maximum (max), and standard deviation (std dev). Note that measurement ID (Msmt ID) is comprised of two digits for year, month, date, and measurement number starting at 01 each day.

| Msmt ID | Latitude | Longitude | Elev-ation (m) | Q Reference (m3 s-1) | Q Float (m3 s-1) | Q Salt (m3 s-1) | Q Bernoulli (m3 s-1) | % Difference Float | % Difference Salt | % Difference Bernoulli |
|---|---|---|---|---|---|---|---|---|---|---|
| 17030202 | 27.78065 | 85.42426 | 1649 | 0.006 | 0.007 | 0.006 | 0.009 | 15.6 | -12.5 | 37.5 |
| 17041802 | 27.78158 | 85.42385 | 1659 | 0.007 | 0.008 | 0.007 | 0.010 | 15.9 | 7.2 | 44.9 |
| 17031001 | 27.79649 | 85.42177 | 1905 | 0.011 | 0.008 | 0.012 | 0.009 | -28.4 | 11 | -19.3 |
| 17042401 | 27.70026 | 85.22077 | 1406 | 0.017 | 0.019 | 0.019 | 0.018 | 11.2 | 11.2 | 4.7 |
| 17032201 | 27.57487 | 85.31314 | 1482 | 0.018 | 0.020 | 0.024 | 0.019 | 12.4 | 37.3 | 5.1 |
| 17041901 | 27.77164 | 85.42657 | 1609 | 0.019 | 0.028 | 0.027 | 0.022 | 48.4 | 47.3 | 16.7 |
| 17033001 | 27.78691 | 85.32589 | 1364 | 0.021 | 0.026 | 0.030 | 0.048 | 27.2 | 43.2 | 132 |
| 17042402 | 27.69620 | 85.23142 | 1382 | 0.023 | 0.010 | 0.025 | 0.006 | -59.2 | 5.2 | -73 |
| 17041903 | 27.75406 | 85.42170 | 1355 | 0.034 | 0.051 | 0.033 | | 51.6 | -0.9 | |
| 17041902 | 27.77154 | 85.42680 | 1609 | 0.041 | 0.041 | 0.047 | 0.063 | 0.5 | 14.6 | 53.2 |
| 17030101 | 27.78483 | 85.44480 | 1877 | 0.104 | 0.111 | 0.088 | 0.102 | 6.9 | -15.9 | -2.6 |
| 17032203 | 27.57542 | 85.31268 | 1477 | 0.111 | 0.106 | 0.120 | 0.116 | -4.3 | 8 | 5.1 |
| 17032202 | 27.57410 | 85.31277 | 1481 | 0.117 | 0.081 | 0.126 | 0.102 | -30.7 | 8 | -13.2 |
| 17033002 | 27.78627 | 85.32583 | 1356 | 0.153 | 0.208 | 0.144 | 0.470 | 36.5 | -5.6 | 207.9 |
| 17030201 | 27.78156 | 85.42383 | 1659 | 0.155 | 0.248 | 0.176 | 0.161 | 59.3 | 13.1 | 3.5 |
| 17041803 | 27.78168 | 85.42373 | 1663 | 0.156 | 0.140 | 0.142 | 0.210 | -10.4 | -8.9 | 34.4 |
| 17031002 | 27.77932 | 85.42496 | 1653 | 0.159 | 0.183 | 0.155 | 0.228 | 15.2 | -2.8 | 43.4 |
| 17031101 | 27.78505 | 85.44473 | 1877 | 0.208 | 0.221 | 0.207 | 0.150 | 6.5 | -0.6 | -27.8 |
| 17031102 | 27.77514 | 85.43867 | 1806 | 0.230 | 0.188 | 0.219 | | -18 | -4.8 | |
| 17042002 | 27.71106 | 85.35432 | 1313 | 0.240 | 0.246 | 0.264 | 0.264 | 2.7 | 10.2 | 10.1 |
| | | Average -> | 1579 | 0.092 | 0.098 | 0.094 | 0.112 | 7.9 | 8.2 | 25.7 |
| | | min -> | 1313 | 0.006 | 0.007 | 0.006 | 0.006 | -59.2 | -15.9 | -73.0 |
| | | max -> | 1905 | 0.240 | 0.248 | 0.264 | 0.470 | 59.3 | 47.3 | 207.9 |
| | | std dev -> | 190 | 0.081 | 0.089 | 0.081 | 0.122 | 29.1 | 17.2 | 61.9 |

[Figure]

Figure 2: Plots of EC (µS cm$^{-1}$; blue trace) and change in EC (µS cm$^{-1}$; green trace) as a function of time (s) for the 20 salt dilution evaluation measurements. Measurement ID (Msmt ID; Table 1) shown at the top right of each subplot (i.e. a through t).

**3.2 Flow and calibration coefficient (k) results**

**3.2.1 Flow scatter plots**

Scatter plots between reference and observed flows with linear regressions forced through the origin had slopes of 1.05, 1.01, and 1.26 for float, salt dilution, and Bernoulli methods, respectively (Fig. 3). A slope of one represents zero systematic bias, whereas values over one represent positive bias, and values less than one represent negative bias. Therefore, for all the methods evaluated we observed different degrees of positive bias. R squared values were 0.90, 0.98, and 0.61 for float, salt dilution, and Bernoulli methods, respectively (Fig. 3). R squared values represent the goodness of fit between the regression and the observed data; values closer to one represent a better fit. This can also be seen by the observations for salt dilution plotting closest to the regression line, whereas float and Bernoulli points in general plot farther away from the regression line.

[Figure]

Figure 3: Scatter plots between reference flow and observed flow for (a) float, (b) salt dilution, and (c) Bernoulli. Note there is one Bernoulli measurement point (17033002) that is outside of the plot space shown (fixed from 0.0 to 0.3 for consistency). Linear regressions and r squared values shown on the bottom right of each sub-plot.

**3.2.2    Flow error correlations**

We found statistically significant correlations (n = 20, p = 0.1, r > 0.378) between the absolute value of percent error for float and average velocity (Avg Vel; sub-plot b; r = -0.48) and salt dilution percent error and reference flow (Q Ref; sub-plot c; r = -0.44) (Fig. 4). In both cases, the correlation coefficient was negative, indicating an inverse relationship between the variables. No statistically significant correlations were observed between the remaining pairs of variables.

[Figure]

Figure 4: Scatter plots between reference flow (Q Ref; $m^3 s^{-1}$), average water velocity (Avg Vel; $m s^{-1}$), slope, and background EC (EC BG; $\mu S cm^{-1}$) and absolute value (Abs) of percent errors for float, salt dilution, and Bernoulli. Pearson's r values shown on the upper right of each subplot (i.e. a through l).

**3.2.3   Salt dilution calibration coefficient (k) results**

The mean calibration coefficient (k) from measurements performed in the field was $2.81 \times 10^{-6} \pm 2.66 \times 10^{-7}$ (95 % confidence interval; n = 10, min = $2.57 \times 10^{-6}$, max = $3.05 \times 10^{-6}$, std dev = $1.33 \times 10^{-7}$). We used mean k to compute salt dilution flows for the remaining 10 measurements. We found statistically significant correlations (n = 10, p = 0.1, r > 0.549) between the calibration coefficient (k) and Longitude (r = 0.60) and Elevation (r = 0.61; Fig. 5). In both cases, the correlation coefficient was positive, indicating a direct relationship between the variables. No statistically significant correlations were observed between the remaining pairs of variables.

[Figure]

Figure 5: Scatter plots between (a) Latitude in degrees, (b) Longitude in degrees, (c) Elevation in meters above mean sea level (m), and (d) background EC in µS cm$^{-1}$ and the salt dilution calibration coefficient (k). Pearson's r values shown on the upper right of each sub-plot.

**3.2 Citizen scientist evaluation results (Phase 2)**

[revised manuscript text omitted]

sub-watersheds of the Kathmandu Valley (Fig. 6). Observed flows ranged from 0.0004 to 0.425 m$^3$ s$^{-1}$ (a summary of the measurement data is included as supplementary material).

The three locations in the Kathmandu Valley that the Nepal Department of Hydrology and Meteorology (DHM) measures either water levels or flows (gauges) are included on Fig. 6.a and 6.b to illustrate the difference in spatial resolutions between the two datasets. Note that only one of the three DHM gauging stations is in a headwater catchment (i.e. Bagmati). Histograms of flow (Fig 6.c and 6.d) and EC (Fig. 6.e and 6.f) show the increase in flows and the decrease in EC from pre to post-monsoon.

[Figure]

[Figure]

**Figure 666: CS Flow**  **campaign measurement locations (n = 131 pre-monsoon; n = 133 post-monsoon** **) within the Kathmandu Valley for (a) pre and (b) post-monsoon. Histograms show distributions of measured flows in L s$^{-1}$ ((c) and (d)) and EC in µS cm$^{-1}$ ((e) and (f)). Bins are set to 20 units wide for both flow and EC. Three flow measurements for the post-monsoon (d) that were above 1000 L s$^{-1}$ are not shown: 1059, 1287, and 1804. Three Department of Hydrology and Meteorology (DHM) gauging stations shown as yellow triangles.**

Scatter plots between flow estimates from the simplified float method (used to calculated salt dosage) and the salt dilution flow results show that systematic differences increase as the observed flow rate decreases (Fig. 7).

[Figure]

Figure 7: Scatter plots between salt dilution measurements on the horizontal axis and simplified float estimates on the vertical axis. All data are shown on subplot (a), flows below 0.05 $m^3 \cdot s^{-1}$ are included on subplot (b), and flows below 0.01 $m^3 \cdot s^{-1}$ are shown on subplot (c). A linear fit with corresponding r-squared values are shown. Note that the vertical axis scales for subplots (b) and (c) are fixed at three times the horizontal axis scale to ensure that all data are visible.

We identified five locations where S4W-Nepal had performed FlowTracker measurements that could be used as reference flows within roughly one month (plus or minus) of the CS Flow salt dilution measurements (Table 2). Comparable flows ranged from 0.012 and 0.111 $m^3 \cdot s^{-1}$. The average error between CS Flow salt dilution and S4W Nepal FlowTracker measurements was -6.3 %, with a standard deviation of 11.5 %. Linear regression forced through the origin between reference flows and CS Flow measurements had a slope of 0.90 with an r-squared value of 0.97.

*Table 2: Comparison between CS Flow salt dilution and S4W-Nepal FlowTracker measurements. Five measurements were identified for evaluation. In one case (i.e. CS Flow Msmt Date 4/16/2018 10:03), a linear interpolation between two S4W-Nepal measurements (i.e. 3/15/2018 7:15 and 5/23/2018 14:23) was made because measurements for both March and May were available. Data summarized at the bottom with average, minimum (min), maximum (max), and standard deviation (std dev).*

| S4W-Nepal SiteID | CS Flow Msmt Date | S4W-Nepal Msmt Date | CS Flow Salt Dilution Q (m3 s-1) | S4W-Nepal FlowTracker Reference Q (m3 s-1) | % Difference |
|---|---|---|---|---|---|
| DB02 | 4/16/2018 10:03 | 3/15/2018 7:15 and 5/23/2018 14:23 | 0.0529 | 0.0492 | 7.5% |
| BM02 | 4/16/2018 13:22 | 3/15/2018 9:49 | 0.0169 | 0.0172 | -1.7% |
| NA02 | 4/18/2018 13:28 | 3/31/2018 4:39 | 0.0940 | 0.1110 | -15.3% |
| BA01 | 4/20/2018 11:30 | 3/30/2018 9:24 | 0.0090 | 0.0118 | -23.7% |
| NK03 | 4/18/2018 13:57 | 5/16/2018 12:22 | 0.0461 | 0.0454 | 1.5% |
| | | average -> | 0.0438 | 0.0469 | -6.3% |
| | | min -> | 0.0090 | 0.0118 | -23.7% |
| | | max -> | 0.0940 | 0.1110 | 7.5% |
| | | std dev -> | 0.0301 | 0.0353 | 11.5% |

**4 Discussion**

4 —

**4.1 Preferred measurement method Initial evaluation discussion (Phase 1)**

10 Our first research question was: *Which simple streamflow measurement method provides the most accurate results when performed by "experts?"* Based on Phase 1 "expert" measurements, we found that salt dilution had the lowest absolute error (i.e. 15 %), compared to float and Bernoulli methods (i.e. 23 and 37 %, respectively; Table 3).

The largest salt dilution errors occurred for reference flows of 21 L s$^{-1}$ or less, while float and Bernoulli errors appeared to be
15 more evenly distributed through the range of observed flows. Because salt dilution measurements of low flows require less salt and water, it is possible that larger relative measurement errors caused while measuring these small quantities led to larger overall measurement errors. However, this is not substantiated in Phase 2 results, so additional research is required in this area. Based on 20 flow measurements performed in this study, we concluded that the salt dilution method will (1) provide the most accurate streamflow data (at least for the range of flows observed), and (2) will be the easiest method for
20 citizen scientists to repeat in the field with limited amounts of training and equipment (see Section 4.4 on citizen scientist repeatability).

Our experience in the field was that float velocity measurements in slow moving and shallow areas were difficult to perform. The combination of turbulence and boundary layer impacts from the streambed and the overlying air mass often made floating objects on the surface travel in non-linear paths, adding uncertainty to distance and time measurements. In the literature, challenges with applying the float method in shallow depths is supported by USBR (2001) and Escurra (2004), who showed that uncertainty in surface velocity coefficients (i.e. the ratio of surface velocity to actual mean velocity of the underlying water column; C from Eq. (1)) increased as depth decreased, especially below 0.3 m.

While all flow measurement methods evaluated had positive biases, salt dilution showed the closest agreement to the reference flow with an average over estimation of only one percent (based on the linear regression), followed by float and Bernoulli at 5 and 25 %, respectively (Fig. 3). The standard deviation for errors was 17 % for salt dilution, and 29 and 62 % for float and Bernoulli, respectively. Additionally, r squared values indicated that salt dilution had the least amount of variance from the trend line (i.e. closest to one).

Only three salt dilution measurements (i.e. 17032201, 17041901, and 17033001) had percent differences larger than 20 %, and these were all positively biased in relatively small streams (flows between 0.018 and 0.021 $m^3 \cdot s^{-1}$). While we can't be certain, we suggest that these errors may be due hyporheic exchanges that removed some salt solution from the measurement reach before lateral and vertical mixing could fully occur. In other words, it is possible that some of the salt solution became "underflow" shortly after the injection point and did not return to the surface stream prior to the EC measurement location. As observed, this "removal" of salt solution would lead to a systematic overestimation of flow. If these three measurements are removed, the mean and standard deviation for salt dilution method percent differences become 2 and 9 %, respectively. These percent differences fall within the expected range of uncertainty presented in the literature for salt dilution gauging (Day 1976; USBR 2001; Moore 2004a; Herschy 2014). Excluding these three errors, and assuming errors are normally distributed, we expect that salt dilution measurements will be within roughly ±18 % (95 % confidence interval). 
[revised manuscript text omitted]

**4.2 Measurement error correlation**

In Fig. 4, three "outlier" percent errors for salt dilution measurements were seen in the middle row of sub-plots (5 through 8) as clusters of three points towards the top of each sub-plot. After removing these points, the Pearson's r value decreased to 0.26, and the correlation became statistically insignificant ($n = 20$, $p = 0.1$, $r > 0.378$). Therefore, we are cautious to conclude that error in salt dilution measurements decreases as the amount of streamflow increases. Errors in salt dilution measurements appeared to be uncorrelated with the other variables evaluated (e.g. average velocity, slope, and $EC_{BG}$).

The other observed statistically significant correlation was an inverse relationship between average velocity and error in float measurements. Our experience in the field validated that slow moving (and shallow) float velocity measurements were difficult to perform. The combination of turbulence and boundary layer impacts from the streambed and the overlying air mass often made floating objects on the surface travel in non linear paths, adding uncertainty to distance and time measurements. Challenges with applying the float method in shallow depths was supported USBR (2001) and Escurra (2004), who showed that uncertainty in surface velocity coefficients (i.e. the ratio of surface velocity to actual mean velocity of the underlying water column) increased as depth decreased, especially below 0.3 m.

**4.3    Salt dilution calibration coefficient (k)**

Moore (2005) suggests that k depends on (1) the ratio of salt and water in the tracer solution and (2) the chemical composition of the stream water. To minimize variability in k due to changes in salt concentration, a fixed ratio of salt to water (e.g. 1 to 6 by mass) should be consistently used to prepare tracer solutions (as it was during this investigation). Significant correlations observed between k and longitude and elevation may be due to changes in water chemistry that co-vary with these independent variables. Measurements performed in the northeastern portion (i.e. higher longitude) of the Kathmandu Valley were higher in altitude. Geology in the north of the Kathmandu Valley is a mixture of weathered igneous and metamorphic parent material (e.g. gneiss, phyllite, schist, etc.). Geology surrounding measurements in the southwest of the Kathmandu Valley is dominated by sedimentary and slightly metamorphosed deposits of sand, silt, and clay (Shrestha et al. 2012). These differences in geology could impact water chemistry through water rock interactions (Lasaga 1984) and ultimately impact k. Additional work should focus on improving our understanding of the variables affecting k. Specifically, spatial variability in k due to changes in stream water chemistry should be investigated prior to applying the salt dilution methodology described in this paper into other areas.

**4.4    Citizen scientist repeatability**

In this context, repeatability refers to the overall likelihood that the measurement method can be successfully repeated by citizen scientists. Along these lines, there were several practical observations in the field worth briefly discussing. While difficult to quantitatively evaluate, we offer the following qualitative observations regarding the selection of a preferred citizen science streamflow measurement method.

- **Required training** — Float and salt dilution require similar amount of training, which from our experience we estimate to be roughly four hours, involving both classroom and field time. The amount of training is strongly dependent on the background of the volunteers. Bernoulli requires additional training for how to minimize vertical movement of the metal measurement plate.

5 - **Cost of equipment** — All methods require a SmartPhone, measuring scale, and measuring tape. Additionally, salt dilution requires an inexpensive EC meter (e.g. $15 HoneForest Water Quality Tester), a graduated cylinder, and a bucket.

- **Number of citizen scientists required** - Teams of at least two citizen scientists are recommended for all methods; teams of three were preferred in our experience.

- **Data recording requirements** — For float and Bernoulli, depth, width, and velocity (including distance and time)

10 data needs to be recorded at multiple locations. Salt dilution only requires some basic data entry and a video of the breakthrough curve.

- **Complexity of procedure** — Float and Bernoulli require detailed transects of the stream. Bernoulli is extremely sensitive to vertical movements in the metal measurement plate. Bernoulli is always very difficult for low velocities.

- **Enjoyability of measurement** — We found that citizen scientists generally enjoyed watching the salt dilution

15 breakthrough curves and found them less repetitive than the tasks associated with float and Bernoulli methods. Bernoulli measurements can be frustrating when trying to keep the metal measurement plate from moving vertically, especially when there is a soft streambed.

- **Safety** — Both float and Bernoulli measurements require citizen scientists to wade through the stream. At certain flow rates this clearly poses a safety risk, especially for people who cannot swim. A clear benefit of the salt dilution method

20 is that everything but the simplified float estimate can be performed from the stream bank. Note that the results from the simplified float method are only used to determine the salt dosing. If the salt dosing doesn't provide a large enough change in EC to be clearly observed, the measurement can be repeated with a higher flow estimate and corresponding increase in tracer solution.

25 **4.5** Uncertainty in reference flows

Uncertainty in measurements of reference flow (i.e. actual flow) affected uncertainty in our evaluation of the three flow measurement methods. Based on an ISO discharge uncertainty calculation within the SonTek FlowTracker software, the uncertainties in flow ranged from 2.5 to 6.0 %, with a mean of 3.7 %. Based on the literature (Rantz 1982; Harmel 2006; Herschy 2014), these uncertainties in reference flows are towards the lower end of the expected range for field measurements

30 of streamflow. Therefore, we do not think that any systematic biases or uncertainties in our data change the results of this paper.

**4.6** Reynolds number

Turbulent mixing of flow is an important aspect of salt dilution flow measurements. Reynolds number is typically used as a quantitative measure of turbulence in fluid flow. In addition to density and viscosity, fluid velocity and a characteristic length are required for calculating Reynolds number. Many of our measurements were performed in mountainous headwater streams with high slopes. To collect the most accurate reference flow measurements, however, we selected the lowest gradient stretch of the stream. The cross sections used for reference flow measurements were typically in the widest and deepest reaches of the stream to ensure the most laminar flow lines for accurate velocity and area measurements. Using velocity and characteristic length data from these reference flow measurement locations to calculate Reynolds number would not have been representative of the actual average Reynolds number of the stream reaches used for the salt dilution measurements. This is because selected stream reaches for salt dilution included steep gradients with lots of mixing, which were often either entirely upstream or downstream of reference flow locations. Therefore, we did not include Reynolds number in the correlation analysis for salt dilution.

**4.7** Flow measurement methods not evaluated

We initially considered including the slope area method (USBR 2001) based on the Manning's equation for evaluation. The concept was to use a long clear flexible tube of a known length (e.g. 20 m) to measure the slope of the water surface using the principle of a water level. The tube was completely submerged and filled with water. The upstream end of the tube remained submerged and the entrance was held perpendicular to flow to ensure that only the pressure head of the stream was sensed at the tube inlet. The tube was stretched out longitudinally along the stream reach, and the downstream end of the tube was exposed to the atmosphere. The difference in water levels inside the downstream end of the tube and the stream water level immediately outside of the tube was measured. This change in head was divided by the total length of the tube to determine the slope of the water surface.

Within the first few days of field work, we concluded that this method was not suitable for piloting at the types of sites we were investigating. Because we were particularly interested in high gradient headwater streams, the primary problem was finding a stretch of stream long enough that was flowing at normal depth, without backwater and drops in the water surface caused by sudden changes in channel geometry (both longitudinally and latitudinally). An additional challenge of this method is that uncertainty in flow measurements are linearly proportional to uncertainty in estimations of the roughness coefficient (n), and n is difficult to estimate visually, especially for citizen scientists. Therefore, Manning's method was not included in this investigation. Despite our experience, we suggest that in certain settings with long straight reaches of uniform flow, Manning's may still be an appropriate citizen science flow measurement method. To simplify observations of the change in head, the Manning's method would also benefit from the installation of upstream and downstream staff gauges survey to a common datum. However, this would make the method difficult and costly to implement, and therefore less scalable.

**4.8**

~~The CS Flow Campaign provided us with a unique opportunity to evaluate the preferred salt dilution citizen science streamflow measurement method at a larger scale. In addition to the valuable streamflow data that will help us characterize the hydrological situation in the Kathmandu Valley with greater precision, we also learned many practical lessons about how to apply citizen science based streamflow generation methods at a broader scale. Unfortunately, there was no systematic way to evaluate the accuracy of all the measurements performed. However, at the five locations were S4W Nepal FlowTracker measurements were available, the resulting errors ($\mu = 6.7$ %, std dev = 11.5 %; Table 2) where comparable to our initial evaluation data ($\mu = 8.2$ %, std dev = 17.2 %; Table 1).~~

~~Linear regression forced through the origin between reference flows and salt dilution measurements had slopes of 0.90 and 1.01 with an r-squared values of 0.97 and 0.98 for CS Flow measurements and our initial evaluation measurements, respectively. Goodness of fit was similar, but while evaluation measurements had a slight positive bias (1 %), CS Flow measurements had a larger negative bias (10 %). One possible explanation for this is that the three CS Flow comparisons that had negative percent differences (i.e. BM02, NA02, and BA01; Table 2) all used reference flows performed prior to CS Flow measurements. Since hydrographs during this season are gradually receding prior to the onset of the South Asian Monsoon, it is possible that the actual flow of the streams decreased between reference flow and CS Flow observations.~~

~~As flows decreased, we observed a progressively increasing positive bias between simplified float estimates and salt dilution measurements (Fig. 7). This finding is congruent with previous efforts to characterize the dynamic relationship between channel depth and surface velocity coefficients (USBR 2001; Escurra 2004). Average stream depths were often on the order of a few centimeters for the headwater catchments observed. Surface velocity coefficients provided by USBR (2001) range from 0.66 to 0.80 with increasing depths, but are held constant at 0.66 for depths less than 0.3 meters. Our results indicate that for flows less than 0.01 m³·s⁻¹ a surface velocity coefficient of 0.5 would be more appropriate. The strength of the relationship between salt dilution and float streamflows also deteriorates as flows decrease (i.e. r squared equals 0.83, 0.69, and 0.50 for plots 1, 2, and 3, respectively). This suggests that surface velocity coefficients are highly variable at low flow rates and correspondingly shallow depths.~~

**5 Summary and future work**

Of the simple streamflow measurement methods evaluated in this paper, salt dilution provides the most accurate streamflow measurements for both "experts" and citizen scientists alike. Our aims in this paper were to (1) perform an initial evaluation of these three potential simple streamflow measurement methods (Phase 1), (2) evaluate the same three methods with actual citizen scientists (Phase 2), and (3) apply the selected approach at a larger scale (Phase 3).

evaluate possible citizen science streamflow measurement methods, (2) select a preferred approach, and (3) pilot test the selected method in a real world setting. We evaluated three different In both Phases 1 and 2, salt dilution method resulted in the lowest absolute errors and biases (approaches (i.e. float, salt dilution, and Bernoulli run up)Table 5) compared to float and Bernoulli methods.

**Table 5: Summary of average absolute errors (Avg Abs Error), average biases (Avg Bias), and error standard deviations (Std Dev Error) for Phase 1 and 2 measurements. All values shown as percentages rounded to the nearest integer.**

| Phase | Performed by | Metric | Float Method | Salt Dilution Method | Bernoulli Method |
|---|---|---|---|---|---|
| 1 | Authors | Avg Abs Error (%) | 23 | 15 | 37 |
| | | Avg Bias (Avg Error (%)) | 8 | 6 | 26 |
| | | Std Dev Error (%) | 29 | 19 | 62 |
| 2 | "Expert" (Authors) | Avg Abs Error (%) | 41 | 21 | 43 |
| | | Avg Bias (Avg Error (%)) | 41 | 19 | 40 |
| | | Std Dev Error (%) | 34 | 26 | 51 |
| 2 | CS Flow Groups | Avg Abs Error (%) | 63 | 28 | 131 |
| | | Avg Bias (Avg Error (%)) | 52 | 7 | 127 |
| | | Std Dev Error (%) | 82 | 36 | 225 |

During Phase 1, we (the authors) performed 20 comparison measurements including float, salt dilution, and Bernoulli methods in headwater catchments of the Kathmandu Valley during March and April of 2017. For reference flows, we performed USGS mid-section method discharge measurements with a SonTek FlowTracker acoustic Doppler velocimeter (ADV). Reference flows ranged from 6.4 to 240 L s$^{-1}$. Absolute errors averaged 23, 15, and 37 %, biases were 8, 6, and 26 %, and error standard deviations were 29, 19, and 62 % for float, salt dilution, and Bernoulli methods, respectively.

During Phase 2, we partnered with 37 citizen scientists (second and third-year Bachelors' student volunteers from Khwopa College of Engineering) to evaluate the same three measurement methods in a citizen science flow (CS Flow) evaluation. In September 2018, CS Flow groups and an "expert" group (authors) performed measurements at 15 sites (seven in the Dhobi and eight in the Nakkhu watersheds). Reference flows, measured with a FlowTracker ADV, ranged from 4.2 to 896 L s$^{-1}$. "Expert" absolute errors averaged 41, 21, and 43 %, while for CS Flow groups they averaged 63, 28, and 131 % for float, salt dilution, and Bernoulli methods, respectively. While there was an increase in absolute error from the "expert" to the CS Flow groups (i.e. 21 to 28 %), salt dilution had the smallest incremental difference of the three methods. "Expert" biases averaged 41, 19, and 40 %, while for CS Flow groups they averaged 52, 7, and 127 % for float, salt dilution, and Bernoulli methods, respectively. Average bias for the salt dilution method was lower for CS Flow groups than for the "expert" (7 compared to 19 %), while for float and Bernoulli methods biases were higher. "Expert" error standard deviations were 34,

26, and 51 %, while for CS Flow groups they were 82, 36, and 225 % for float, salt dilution, and Bernoulli methods, respectively.  Based on these results, we selected salt dilution as the preferred simple streamflow measurement method.

Finally, during Phase 3, we performed larger scale pilot testing of the salt dilution method in week-long pre and post-monsoon (April and September 2018) CS Flow campaigns involving 25 and 37 citizen scientists, respectively.  Observed flows (n = 131 pre-monsoon; n = 133 post-monsoon) were distributed among the 10 headwater catchments of the Kathmandu Valley and ranged from 0.4 to 425 L s$^{-1}$ (pre) and 1.51 to 1804 L s$^{-1}$ (post).  Histograms of flow and EC showed the increase in flows and the decrease in EC from pre to post-monsoon.  The Department of Hydrology and Meteorology in Nepal operates three gauging stations in the Kathmandu Valley, so these additional data should add important spatial and temporal resolution to the distribution of streamflow in the Kathmandu Valley.  During salt dilution measurements, background EC of streams and springs is also measured, which provides information about water quality.

 by performing 20 side by side comparison measurements in headwater catchments of the Kathmandu Valley.  We used USGS mid section discharge measurements from a SonTek FlowTracker acoustic Doppler velocimeter as reference flows.  Evaluated flows ranged from 0.006 to 0.240 m$^3$-s$^{-1}$.  Linear regressions forced through the origin for scatter plots with reference flows had slopes of 1.05, 1.01, and 1.26 with r squared values of 0.90, 0.98, and 0.61, for float, salt dilution, and Bernoulli run up methods, respectively.  The salt dilution method was selected as the preferred approach based on its favourable quantitative results compared to the other methods, and other qualitative factors concerning citizen science repeatability.  The approach was then pilot tested in a CS Flow Campaign, which involved 20 volunteers performing 145 measurements, ranging from 0.0004 to 0.425 m$^3$-s$^{-1}$, distributed among the 10 headwater catchments of the Kathmandu Valley.  While there was no way to evaluate the accuracy of all 145 measurements, five of the measurements were performed in locations where USGS mid section method discharge measurements had been performed.  For these five locations, a linear regression forced through the origin between reference flows and CS Flow measurements had a slope of 0.90 with an r squared value of 0.97.

Motivated by these promising results, fFuture work should further evaluate the feasibility of applying citizen science based salt dilution streamflow measurements to larger areas of Nepal and beyond.  The information content of additional streamflow data should be explored.  Issues of how to effectively recruit and motivate citizen scientists and young researchers (i.e. all science and engineering minded students from primary through graduate school ages) to participate in citizen science streamflow measurement efforts should receive additional attention, especially in the relatively unexplored context of citizen science in Asia.  Finally, the assumption of a constant calibration coefficient (k) should be evaluated over a larger sample size covering a broader range of geological and water quality conditions.

**6 Data availability**

The data used in this paper are provided as supplementary material.

**7 Author contribution**

Jeffrey C. Davids had the initial idea for this investigation and designed the experiments in collaboration with Martine M. Rutten, Wessel David van Oyen, and Nick van de Giesen. Field work was performed by Jeffrey C. Davids, Anusha Pandey, Nischal Devkota, Wessel David van Oyen, and Rajaram Prajapati. Jeffrey C. Davids prepared the manuscript with valuable contributions from all co-authors.

**8 Competing interests**

The authors declare that they have no conflict of interest.

**9 Acknowledgements**

This work was supported by the Swedish International Development Agency under Grant number 2016-05801; and by SmartPhones4Water (S4W). We appreciate the dedicated efforts of Annette van Loosen, Bhumika Thapa, Sunil Duwal,  citizen scientists from Khwopa College of Engineering, Anurag Gyawali, Anu Grace Rai, Sanam Tamang, Eliyah Moktan, Kristi Davids, and the rest of the S4W-Nepal team. We would also like to thank Dr. Ram Devi Tachamo Shah, Dr. Deep Narayan Shah, Dr. Narendra Man Shakya, and Dr. Steve Lyon for their supervision and support of this work. Finally, a special thanks to SonTek for their donation of a FlowTracker acoustic Doppler velocimeter that was used for the reference flow measurements discussed in this paper and many more to come.

~~This research was performed in the context of a larger citizen science project called SmartPhones4Water or S4W (Davids et al. 2017; Davids et al. 2018; www.SmartPhones4Water.org). S4W focuses on leveraging citizen science, mobile technology, and young researchers to improve lives by strengthening our understanding and management of water. S4W's first pilot project, S4W-Nepal, initially concentrated on the Kathmandu Valley, and is now expanding into other regions of the country and beyond. All of S4W's efforts, including the research herein, have a focus on simple field data collection methods that can be standardized and scaled so that young researchers and citizen scientists can help fill data gaps in other data scarce regions.~~